# Chlorine partitioning near the polar vortex edge observed with ground-based FTIR and satellites at Syowa Station, Antarctica in 2007 and 2011

Hideaki Nakajima[1,2], Isao Murata[2], Yoshihiro Nagahama[1], Hideharu Akiyoshi[1], Kosuke Saeki[2,3], Takeshi Kinase[4], Masanori Takeda[2], Yoshihiro Tomikawa[5,6], Eric Dupuy[1], and Nicholas B. Jones[7]

[1]National Institute for Environmental Studies, Tsukuba, Ibaraki, 305-8506, Japan
[2]Graduate School of Environmental Studies, Tohoku University, Sendai, Miyagi, 980-8572, Japan
[3]now at Weathernews Inc., Chiba, 261-0023, Japan
[4]Meteorological Research Institute, Tsukuba, Ibaraki, 305-0052, Japan
[5]National Institute of Polar Research, Tachikawa, Tokyo, 190-8518, Japan
[6]The Graduate University for Advanced Studies, Tachikawa, Tokyo, 190-8518, Japan
[7]University of Wollongong, Wollongong, New South Wales, 2522, Australia

*Correspondence to*: Hideaki Nakajima (nakajima@nies.go.jp)

**Abstract.**

We retrieved lower stratospheric vertical profiles of $O_3$, $HNO_3$, and $HCl$ from solar spectra taken with a ground-based Fourier-Transform infrared spectrometer (FTIR) installed at Syowa Station, Antarctica (69.0°S, 39.6°E) from March to December 2007 and September to November 2011. This was the first continuous measurements of chlorine species throughout the ozone hole period from the ground in Antarctica. We analyzed temporal variation of these species combined with ClO, HCl, and $HNO_3$ data taken with the Aura/MLS (Microwave Limb Sounder) satellite sensor, and $ClONO_2$ data taken with the Envisat/MIPAS (The Michelson Interferometer for Passive Atmospheric Sounding) satellite sensor at 18 and 22 km over Syowa Station. HCl and $ClONO_2$ decrease occurred at both 18 and 22 km, and soon both HCl and $ClONO_2$ were almost depleted in early winter. When the sun returned to Antarctica in spring, enhancement of ClO and gradual $O_3$ destruction were observed. During the ClO enhanced period, negative correlation between ClO and $ClONO_2$ was observed in the time-series of the data at Syowa Station. This negative correlation was associated with the relative distance between Syowa Station and the edge of the polar vortex. We used MIROC3.2 Chemistry-Climate Model (CCM) results to see the behavior of whole chlorine and related species inside the polar vortex and the boundary region in more detail. From CCM model results, rapid conversion of chlorine reservoir species (HCl and $ClONO_2$) into $Cl_2$, gradual conversion of $Cl_2$ into $Cl_2O_2$, increase of HOCl in winter period, increase of ClO when sunlight became available, and conversion of ClO into HCl, was successfully reproduced. HCl decrease in the winter polar vortex core continued to occur due to both transport of $ClONO_2$ from the subpolar region to higher latitudes, providing a flux of $ClONO_2$ from more sunlit latitudes into the polar vortex, and the heterogeneous reaction with HOCl. Temporal variation of chlorine species over Syowa Station was affected by both heterogeneous chemistries related to Polar Stratospheric Cloud (PSC) occurrence inside the polar vortex, and transport of a $NO_x$-rich airmass from the polar vortex boundary region which can produce additional $ClONO_2$ by reaction of ClO with $NO_2$. The deactivation

pathways from active chlorine into reservoir species (HCl and/or $ClONO_2$) were confirmed to be highly dependent on the availability of ambient $O_3$. At an altitude where most ozone was depleted in Antarctica (18 km), most ClO was converted to HCl. However, at an altitude where there were some $O_3$ available (22 km), additional increase of $ClONO_2$ from pre-winter value can occur, similar to the case in the Arctic.

## 1. Introduction

Discussion of the detection of "recovery" of the Antarctic ozone hole as the result of chlorofluorocarbon (CFC) regulations has been attracting attention. The occurrence of the Antarctic ozone hole is considered to continue at least until the middle of this century. The world's leading Chemistry-Climate Models (CCMs) indicate that the multi-model mean time series of the springtime Antarctic total column ozone will return to 1980 levels shortly after mid-century (about 2060) (WMO, 2019). In fact, the recovery time predicted by CCMs has large uncertainty, and the observed ozone hole magnitude also shows year-to-year variability (e.g., see Figure 4-6 in WMO (2019)). Although Solomon et al. (2016) and de Laat et al. (2017) reported signs of healing in the Antarctic ozone layer only in September, there is no statistically conclusive report on the Antarctic ozone hole recovery (Yang et al., 2008; Kuttippurath et al, 2010; WMO, 2019).

To understand ozone depletion processes in polar regions, understanding of the behavior and partitioning of active chlorines ($ClO_x=Cl+Cl_2+ClO+ClOO+Cl_2O_2$) and chlorine reservoirs (HCl and $ClONO_2$) are crucial. Recently, the importance of $ClONO_2$ was reviewed by von Clarmann and Johansson (2018). Chlorine reservoir is converted to active chlorine that destroys ozone on polar stratospheric clouds (PSCs) and/or cold binary sulphate through heterogeneous reactions:

$$ClONO_2 \text{ (g)} + HCl \text{ (s, l)} \rightarrow Cl_2 \text{ (g)} + HNO_3 \qquad (R1)$$
$$ClONO_2 \text{ (g)} + H_2O \text{ (s, l)} \rightarrow HOCl \text{ (g)} + HNO_3 \qquad (R2)$$

where g, s, and l represent the gas, solid, and liquid phases, respectively (Solomon et al., 1986; Solomon, 1999; Drdla and Müller, 2012, Wegner et al., 2012; Nakajima et al., 2016).

Heterogeneous reactions:

$$N_2O_5 \text{ (g)} + HCl \text{ (s, l)} \rightarrow ClNO_2 \text{ (g)} + HNO_3 \qquad (R3)$$
$$HOCl \text{ (g)} + HCl \text{ (s, l)} \rightarrow Cl_2 \text{ (g)} + H_2O \qquad (R4)$$

are responsible for additional chlorine activation. When solar illumination is available, $Cl_2$, HOCl, and $ClNO_2$ are photolyzed to produce chlorine atoms by reactions:

$$Cl_2 + h\nu \rightarrow Cl + Cl \qquad (R5)$$
$$HOCl + h\nu \rightarrow Cl + OH \qquad (R6)$$
$$ClNO_2 + h\nu \rightarrow Cl + NO_2. \qquad (R7)$$

The yielded chlorine atoms then start to destroy ozone catalytically through reactions (Canty et al., 2016):

$$Cl + O_3 \rightarrow ClO + O_2 \qquad (R8)$$

$$ClO + ClO + M \rightarrow Cl_2O_2 + M \qquad \text{(R9)}$$

$$Cl_2O_2 + hv \rightarrow Cl + ClOO \qquad \text{(R10)}$$

$$ClOO + M \rightarrow Cl + O_2 + M. \qquad \text{(R11)}$$

There are three types of PSCs, i.e., nitric acid trihydrate (NAT), supercooled ternary solution (STS), and ice PSCs. When the stratospheric temperatures get warmer than NAT saturation temperature (about 195 K at 50 hPa) and no PSCs are present, deactivation of chlorine starts to occur. Re-formation of $ClONO_2$ and HCl mainly occurs through reactions (Santee et al., 2008; Grooß et al., 2011; Müller et al., 2018):

$$ClO + NO_2 + M \rightarrow ClONO_2 + M \qquad \text{(R12)}$$

$$Cl + CH_4 \rightarrow HCl + CH_3 \qquad \text{(R13)}$$

$$CH_2O + Cl \rightarrow HCl + CHO. \qquad \text{(R14)}$$

The re-formation of $ClONO_2$ by reaction (R12) from active chlorine is much faster than that of HCl by reactions (R13) and (R14), if enough $NO_x$ are available (Mellqvist et al., 2002; Dufour et al., 2006). But the formation rates of $ClONO_2$ and HCl are also related to ozone concentration. Grooß et al. (1997) showed that HCl increases more rapidly in the Antarctic polar vortex than in the Arctic polar vortex due to lower ozone concentrations in the Antarctic polar vortex. Low ozone reduces the rate of reaction (R8), and then Cl/ClO ratio becomes high. Low ozone also reduces the rate of the following reaction:

$$NO + O_3 \rightarrow NO_2 + O_2. \qquad \text{(R15)}$$

This makes $NO/NO_2$ ratio high and increases Cl/ClO ratio by the following reaction:

$$ClO + NO \rightarrow Cl + NO_2. \qquad \text{(R16)}$$

High Cl/ClO ratio leads to rapid HCl formation by reactions (R13) and (R14), and reduces the formation ratio of $ClONO_2$ by reaction (R12) (Grooß et al., 2011; Müller et al., 2018).

The processes of deactivation of active chlorine are different between typical conditions in the Antarctic and those in the Arctic. In the Antarctic, the temperature cools below the NAT PSC existence threshold (about 195 K at 50 hPa) in the whole area of the polar vortex in all years, and almost complete denitrification and chlorine activation occur (WMO, 2007), followed by severe ozone depletion in spring. In the chlorine reservoir recovery phase, HCl is mainly formed by reaction (R13) due to the lack of ozone (typically less than 0.5 ppmv) by the mechanism described in the previous paragraph (Grooß et al., 2011).

On the other hand, in the Arctic, typically less PSC formation occurs in the polar vortex due to generally higher stratospheric temperatures (~10-15K in average) compared with that of Antarctica. Then only partial denitrification and chlorine activation occur in some years (Manney et al., 2011; WMO, 2014). In this case, some ozone and $NO_2$ are available in the chlorine reservoir recovery phase. Therefore, the $ClONO_2$ amount becomes higher than that of HCl after PSCs have disappeared due to the rapid reaction (R12) (Michelsen et al., 1999; Santee et al., 2003), which results in additional increase of $ClONO_2$ than pre-winter value at the time of chlorine deactivation in spring (von Clarmann et al., 1993; Müller et al., 1994; Oelhaf et al., 1994). In this way, the partitioning of the chlorine reservoir in springtime is related to temperature, PSC amounts, ozone, and $NO_2$ concentrations (Santee et al., 2008; Solomon et al., 2015).

In the polar regions, the ozone and related atmospheric trace gas species have been intensively monitored by several measurement techniques since the discovery of the ozone hole. These measurements consist of direct observations by high-altitude aircrafts (e.g., Anderson et al., 1989; Ko et al., 1989; Tuck et al., 1995; Jaeglé et al., 1997; Bonne et al., 2000), remote sensing observations by satellites (e.g., Müller et al., 1996; Michelsen et al., 1999; Höpfner et al., 2004; Dufour et al., 2006;

Hayashida et al., 2007), remote sensing observations of OClO using UV-visible spectrometer from the ground (Solomon et al., 1987; Kreher et al., 1996), and remote sensing observations of ClO by a microwave spectrometer from the ground (de Zafra et al., 1989). Within these observations, ground-based measurements have the characteristic of high temporal resolution. In addition, the Fourier-Transform infrared spectrometer (FTIR) has the capability of measuring several trace gas species at the same time or in a short time interval (Rinsland et al., 1988). In this paper, we show the results of ground-based FTIR

observations of $O_3$ and other trace gas species at Syowa Station in the Antarctic in 2007 and 2011, combined with satellite measurements of trace gas species from the Microwave Limb Sounder onboard the Aura satellite (Aura/MLS) and Michelson Interferometer for Passive Atmospheric Sounding onboard the European Environmental Satellite (Envisat/MIPAS), to show the temporal variation and partitioning of active chlorine ($ClO_x$) and chlorine reservoirs (HCl, $ClONO_2$) from fall to spring during the ozone hole formation and dissipation period. In order to monitor the appearance of PSCs over Syowa Station, we

used the Cloud-Aerosol Lidar with Orthogonal Polarization (CALIOP) data onboard the Cloud-Aerosol Lidar and Infrared Pathfinder Satellite Observations (CALIPSO) satellite. The methods of FTIR and satellite measurements are described in Section 2. The validation of FTIR measurements is described in Section 3. The results of FTIR and satellite measurements and discussion on the behavior of active and inert chlorine species using the MIROC3.2 chemistry-climate model are described in Section 4.

## 2. Measurements

### 2.1 FTIR measurements

The Japanese Antarctic Syowa Station (69.0°S, 39.6°E) was established in January 1957. Since then, several scientific observations related to meteorology, upper atmospheric physics, glaciology, biology, geology, seismology, etc. have been

performed. The ozone hole was first detected by Dobson spectrometer and ozonesonde measurements from Syowa Station in 1982 (Chubachi, 1984) and by Dobson spectrometer measurement at Halley Bay (Farman et al., 1985). We installed a Bruker IFS-120M high-resolution Fourier-Transform infrared spectrometer (FTIR) in the Observation Hut at Syowa Station in March 2007. This was the third high-resolution FTIR site in Antarctica in operation after U.S.A.'s South Pole Station (90.0°S) (Goldman et al., 1983; Goldman et al., 1987; Murcray et al., 1987), U.S.A.'s McMurdo Station and New Zealand's Arrival

Heights facility at Scott Station (77.8°S, 166.7°E) (Farmer et al., 1987; Murcray et al., 1989; Kreher et al, 1996; Wood et al., 2002; Wood et al., 2004). The IFS-120M FTIR has a wavenumber resolution of 0.0035 cm$^{-1}$, with two liquid nitrogen cooled detectors (InSb and HgCdTe covering the frequency ranges 2000-5000 and 700-1300 cm$^{-1}$, respectively) with six optical filters,

and fed by an external solar tracking system. One measurement takes about 10 minutes. At least six spectra were taken per day covering each filter region. Since Syowa Station is located at a relatively low latitude (69.0°S) compared with McMurdo or Scott Stations (77.8°S), there is an advantage of the short (about one month) polar night period, when we cannot measure atmospheric species using the sun as a light source. Since FTIR measurements at Syowa Station are possible from early spring

(late July), FTIR can measure chemical species during ozone hole development. On the other hand, FTIR observations become possible only after September at McMurdo and Scott Stations. Another advantage of Syowa Station is that it is sometimes located at vortex boundary as well as inside and outside of the polar vortex and this enables us to measure chemical species at different regions of polar chemistry related to the ozone hole. From March to December 2007, we made in total 78 days of FTIR measurements on sunny days. Another 19 days of FTIR measurements were performed from September to November

2011. After a few more measurements were performed in 2016, the FTIR was brought back to Japan in 2017. In Appendix, Table A1 shows the days when FTIR measurements were made at Syowa Station with the information inside/boundary/outside of the polar vortex defined by the method described in Appendix B using ERA-Interim reanalysis data. Strahan et al. (2014) showed the year-to-year variation of $Cl_y$ observed in the lower stratosphere of the Antarctic polar vortex. The $Cl_y$ observed in 2007 (2.88 ppbv) was about +4.3% more, and that observed in 2011 (2.53 ppbv) was about -5.2% less, than the projected $Cl_y$

from Newman et al. (2007) (2.76 ppbv for 2007 and 2.67 ppbv for 2011).

The retrieval of the FTIR spectra was done with the SFIT2 Version 3.92 program (Rinsland et al., 1998; Hase et al., 2004). SFIT2 retrieves a vertical profile of trace gases using an optimal estimation formulation of Rodgers (2000), implemented with a semi-empirical method which was originally developed for microwave measurements (Parrish et al., 1992; Connor et al., 1995). The SFIT2 forward model fully describes the FTIR instrument response, with absorption coefficients calculated using

the algorithm of Norton and Rinsland (1991). The atmosphere is constructed with 47 layers from the ground to 100 km, using the FSCATM (Gallery et al., 1983) program for atmospheric ray-tracing to account for refractive bending. The retrieval parameters for each gas, typical vertical resolution, and typical degrees of freedom for signal (DOFS) are shown in Table 1. Temperature and pressure profiles between 0 and 30 km are taken by the Rawin sonde observations flown from Syowa Station on the same day by the Japanese Meteorological Agency (JMA), while values between 30 and 100 km are taken from the

COSPAR International Reference Atmosphere 1986 (CIRA-86) standard atmosphere profile (Rees et al., 1990).

We retrieved vertical profiles of $O_3$, $HCl$, and $HNO_3$ from the solar spectra. We used monthly averaged ozonesondes profiles (0-30 km) and Improve Limb Atmospheric Spectrometer-II (ILAS-II) (Nakajima, 2006; Nakajima et al., 2006; Sugita et al., 2006) profiles (30-100 km) for the a priori of $O_3$, monthly averaged profiles from ILAS-II for $HNO_3$ and monthly averaged profiles from HALOE (Anderson et al., 2000) for $HCl$. We focus on the altitude range of 15-25 km in this study. Typical

averaging kernels of the SFIT2 retrievals for $O_3$, $HNO_3$, and $HCl$ are shown in Figures 1(a), (b), and (c), respectively.

## 2.2 Satellite measurements

The Earth Observing System (EOS) MLS onboard the Aura satellite was launched on 15 July 2004, to monitor several atmospheric chemical species in the upper troposphere to mesosphere (Waters et al., 2006). The Aura orbit is sun-synchronous

at 705 km altitude with an inclination of 98°, 13:45 ascending (north-going) equator-crossing time, and 98.8-min period. Vertical profiles are measured every ~165 km along the suborbital track, horizontal resolution is ~200-600 km along-track, ~3-10 km across-track, and vertical resolution is ~3-4 km in the lower to middle stratosphere (Froidevaux et al., 2006). ClO, HCl, and $HNO_3$ profiles used in this study were taken from Aura/MLS version 4.2 data (Liversey et al., 2006; Santee et al., 2011; Ziemke et al., 2011; Liversey et al., 2018). Only daytime ClO data was used for the analysis. The daily MLS data within 320 km distance between the measurement location and Syowa Station were selected.

MIPAS is a Fourier transform spectrometer sounding the thermal emission of the earth's atmosphere between 685 and 2410 $cm^{-1}$ (14.6-4.15 μm) in limb geometry (Fischer and Oelhaf, 1996; Fischer et al., 2008). The maximum optical path difference of MIPAS is 20 cm. The field-of-view of the instrument at the tangent points is about 3 km in the vertical and 30 km in the horizontal. In the standard observation mode in one limb-scan, 17 tangent points are observed with nominal altitudes 6, 9, 12, 15, 18, 21, 24, 27, 30, 33, 36, 39, 42, 47, 52, 60, and 68 km. In this mode, about 73 limb scans are recorded per orbit. The measurements of each orbit cover nearly the complete latitude range from about 87°S to 89°N. MIPAS was placed on board Envisat, which was launched on 1 March 2002, and was put into a polar sun-synchronous orbit at an altitude of about 800 km with an inclination of 98.55° (von Clarmann et al., 2003). On its descending node, the satellite crosses the equator at 10:00 local time. Envisat performs 14.3 orbits per day, which results in a good global coverage. $ClONO_2$ profiles which we used in this study were taken from Envisat/MIPAS IMK/IAA version V5R_CLONO2_220 and V5R_CLONO2_222 (Höpfner et al., 2007). The daily MIPAS data within 320 km distance between the measurement location and Syowa Station were selected.

The CALIPSO satellite was launched on 28 April 2006. On the CALIPSO satellite, the CALIOP instrument was on board, to monitor aerosols, clouds, and PSCs (Pitts et al., 2007). CALIOP is a two-wavelength, polarization sensitive lidar that provides high vertical resolution profiles of backscatter coefficient at 532 and 1064 nm, as well as two orthogonal (parallel and perpendicular) polarization components at 532 nm (Winker et al., 2007). In order to monitor the appearance of PSCs over Syowa Station, the daily CALIOP PSC data (Pitts et al., 2007; 2009; 2011) within 320 km distance between the measurement location and Syowa Station were selected.

## 3. Validation of retrieved profiles from FTIR spectra with other measurements

We validated retrieved FTIR profiles of $O_3$ with ozonesondes and Aura/MLS version 3.3 data (Liversey et al., 2013) for 2007 measurements. Also, retrieved FTIR profiles of $HNO_3$ and HCl were validated with Aura/MLS data. We identified the nearest Aura/MLS data from the distance between the Aura/MLS tangent point at 20 km altitude and the point at 20 km altitude for the direction of the sun from Syowa Station at the time of the FTIR measurement. The spatial and temporal collocation criteria used was within 300 km radius and ±6 hours. The ozonesonde and Aura/MLS profiles were interpolated onto a 1 km-grid, then smoothed using a 5 km-wide running mean.

Figures 2(a)-(b) show absolute and relative differences of $O_3$ profiles retrieved from FTIR measurements and those from model 1Z ECC-type ozonesonde measurements, respectively, calculated from 14 coincident measurements from September 5 to December 17, 2007. Typical precision and accuracy of the ECC-type ozone sondes are considered to be ±(3-5)% and ±(4-5)%, respectively (Komhyr, 1986). We define the relative percentage difference D as:

$D (\%) = 100 * (FTIR-sonde) / ((FTIR+sonde)/2).$                     (1)

The mean absolute difference between 15 and 25 km was within -0.02 to 0.40 ppmv. The mean relative difference D between 15 and 25 km was within -10.4 to +24.4%. The average of the mean relative differences D of $O_3$ for the altitude of interest in this study (18-22 km) was +6.1%, with the minimum of -10.4% and the maximum of +19.2%. FTIR data agree with validation data within root mean squares of typical errors in FTIR and validation data at the altitude of interest. Note that

relatively large D values between 16 and 18 km are due to small ozone amount in the ozone hole. Our validation results are quite comparable with the validation study at Izaña Observatory by Schneider et al. (2008).

Figures 2(c)-(d) show absolute and relative differences of $O_3$ profiles retrieved from FTIR measurements and those from Aura/MLS measurements, respectively, calculated from 33 coincident measurements from April 1 to December 20, 2007. The accuracy of MLS $O_3$ data are reported to be 5-8% between 0.5 and 46 hPa (Livesey et al., 2013). The mean absolute difference

between 15 and 25 km was within -0.13 to +0.16 ppmv. The mean relative difference D between 15 and 25 km was within -16.2 to +5.2%. The average of the mean relative differences D for $O_3$ for the altitude of interest in this study (18-22 km) was -5.5%, with the minimum of -16.3% and the maximum of +4.5%. Froidevoux et al. (2008) showed that Aura/MLS is +8% higher than ACE-FTS at 70°S, which may explain the negative bias of FTIR data compared with MLS data.

Figures 2(e)-(f) show absolute and relative differences of $HNO_3$ profiles retrieved by FTIR measurements and those from

Aura/MLS measurements, respectively, calculated from 47 coincident measurements from March 25 to December 20, 2007. The mean absolute difference between 15 and 25 km was within -0.56 to +0.57 ppbv. The mean relative difference D between 15 and 25 km was within -25.5 to +21.9%. The average of the mean relative differences D for $HNO_3$ for the altitude of interest in this study (18-22 km) was +13.2%, with the minimum of +0.2% and the maximum of +21.9%. This positive bias of FTIR data is still within the error bars of FTIR measurements. Livesey et al. (2013) showed that Aura/MLS version 3.3 data has no

bias within errors (~0.6-0.7 ppbv (10-12%) at pressure level of 100-3.2 hPa) compared with other measurements. Livesey et al. (2018) showed no major differences between Aura/MLS version 3.3 and version 4.2 data for $HNO_3$.

Figures 2(g)-(h) show absolute and relative differences of HCl profiles retrieved by FTIR measurements and those from Aura/MLS measurements, respectively, calculated from 50 coincident measurements from March 25 to December 20, 2007. The mean absolute difference between 15 and 25 km was within -0.20 to -0.09 ppbv. The mean relative difference D between

15 and 25 km was within -34.1 to -3.0%. The average of mean relative differences D for HCl for the altitude of interest in this study (18-22 km) is -9.7%, with a minimum of -14.6% and a maximum of -3.0%. This negative bias of FTIR data is still within the error bars of FTIR measurements. Moreover, Livesey et al. (2013) showed that Aura/MLS version 3.3 values are systematically greater than HALOE values by 10-15% with a precision of 0.2-0.6 ppbv (10-30%) in the stratosphere, which

may partly explain the negative bias of FTIR data compared with MLS data. Livesey et al. (2018) showed no major differences between Aura/MLS version 3.3 and version 4.2 data for HCl.

Table 2 summarizes validation results of FTIR profiles compared with ozonesonde or Aura/MLS measurements, and possible Aura/MLS biases from literature.

## 4. Results and discussion

### 4.1 Time series of observed species

Figure 3(a) shows daytime hours at Syowa Station. Polar night ends at Syowa Station on July 14 (day 195). Figures 3(b)-(e) show the time series of temperatures at 18 and 22 km over Syowa Station using ERA-Interim data (Dee et al., 2011) for 2007

and 2011. Approximate saturation temperatures for NAT ($T_{NAT}$) and ice ($T_{ICE}$) calculated by assuming 6 ppbv $HNO_3$ and 4.5 ppmv $H_2O$ are also shown in the figures. The dates when PSCs were observed at Syowa Station identified by the nearest CALIOP data of that day were indicated by asterisks at the bottom of the figures. Over Syowa Station, PSCs were observed when temperature fell ~4K below $T_{NAT}$. PSCs were often observed at 15-25 km from the beginning of July (day 183) to late August (day 241) in 2007, and from late June (day 175) to early September (day 251) in 2011.

PSCs were observed only at 18 km after August, due to the sedimentation of PSCs and downwelling of vortex air in late winter as is seen in Figure 3. Although temperatures above Syowa Station were sometimes below $T_{NAT}$-4K in June and in late September, no PSC was observed during those periods. This may be due to other reasons, such as a different time history of temperature for PSC formation, and/or low $HNO_3$ (denitrification) and/or $H_2O$ concentration (dehydration) which are needed for PSC formation in late winter season (Saitoh et al., 2006).

Figures 4-7 show time series of HCl, $ClONO_2$, ClO, $Cl_y$*, $O_3$, and $HNO_3$ over Syowa Station in 2007 and 2011 at altitudes of 18 and 22 km for all ground-based and satellite-based observations used in this study, respectively. $O_3$ (sonde) is observed with the KC96 ozonesonde for 2007, which is different from the ones that were used for the validation in Section 3, and the ECC-1Z ozonesonde for 2011 by JMA (Smit and Straeter, 2004). HCl and $HNO_3$ observed by Aura/MLS and FTIR are plotted by different symbols. Total inorganic chlorine $Cl_y$* corresponds to the sum of HCl, $ClONO_2$, and $Cl_x$, where active chlorine

species $Cl_x$ is defined as the sum of ClO, Cl, and $2*Cl_2O_2$ (Bonne et al., 2000). It is known that total inorganic chlorine $Cl_y$* has a compact relationship with $N_2O$ (Bonne et al., 2000; Schauffler et al., 2003; Strahan et al., 2014). Inferred total inorganic chlorine $Cl_y$* is calculated from the $N_2O$ value (in ppbv) measured by MLS and by using the empirical polynomial equation derived from the correlation analysis of $Cl_y$ and $N_2O$ from the Photochemistry of Ozone Loss in the Arctic Region in Summer (POLARIS) mission which took place from April to September 1997 (Bonne et al., 2000). In order to compensate for the

temporal trends of $Cl_y$ and $N_2O$ values (2.09, 2.76, and 2.67 ppbv for $Cl_y$ (Strahan et al., 2014), and 313, 321, and 324 ppbv for $N_2O$ (WMO, 2007; 2011; 2014) for the years 1997, 2007, and 2011, respectively), we used values 8 and 11 for constant A, and 140 and 230 for constant B for years 2007 and 2011 in the following equation, respectively:

$$Cl_y*(pptv) = 4.7070*10^{-7}(N_2O-A)^4 - 3.2708*10^{-4}(N_2O-A)^3 + 4.0818*10^{-2}(N_2O-A)^2 - 4.6856(N_2O-A) + 3225-B. \quad (2)$$

A transport barrier of minor constituents at the edge of polar vortex was reported by Lee et al. (2001) and Tilmes et al. (2006). The distribution of minor constituents is quite different among the inside, the boundary region, and outside the polar vortex. In Figures 4-7, the dark shaded area, the light shaded area, and the white area indicate the days when Syowa Station was located outside, in the boundary region, and inside the polar vortex, respectively. In the Antarctic winter, there are often double peaks in the isentropic potential vorticity gradient with respect to equivalent latitude at the 450-600 K level (Tomikawa et al., 2015). The method to determine the three polar regions, i.e., inside the polar vortex, the boundary region, and outside the polar vortex is described in Appendix B. Note that the Syowa Station is often located near the vortex edge and the temporal variations of chemical species observed over Syowa Station reflect the spatial distributions as well as local chemical evolution.

When Syowa Station was located at the boundary region or outside the polar vortex (e.g., day 310-316 in Figure 4, day 192-195 in Figure 5, day 309-316 in Figure 6, day 276-282 in Figure 7), chemical species showed different values compared with the ones inside the polar vortex. The lack of data for ClO and HCl (MLS) from day 195 to day 219, 2007 and $ClONO_2$ from day 170 to day 216, 2007 (Figures 4(a) and 6(a)) is due to unrealistic large error values in Aura/MLS or Envisat/MIPAS data products during these periods.

The altitude of 18 km was selected because it was one of the altitudes where nearly complete ozone loss occurred. The altitude of 22 km, where only about half of the ozone was depleted, was selected to show the difference in the behavior of chemical species from that at 18 km.

The general features of the chemical species observed inside the polar vortex at 18 and 22 km in 2007 and 2011 are summarized as follows: HCl and $ClONO_2$ decreased first, then ClO started to increase in winter, while HCl increases and ClO decreases were synchronized in spring. HCl was almost zero from late June to early September and the day-to-day variations were small over this period. HCl over Syowa Station indicates relatively larger values when it was located outside the polar vortex: For example, early August and the beginning of September at 22 km, 2007 in Figure 6. $HNO_3$ showed large decreases from June to July, and then gradually increased in summer. Day-to-day variations of $HNO_3$ from June to August were large. $O_3$ decreased from July to late September when ClO concentration was increased. ClO was enhanced in August and September and the day-to-day variations were large over this period. $Cl_y*$ gradually increased in the polar vortex from late autumn to spring. The $Cl_y*$ value became larger compared with its mixing ratio outside of the polar vortex in spring.

The following characteristics are evident at 18 km (Figures 4 and 5). $O_3$ gradually decreased from values of 2.5-3 ppmv before winter to values less than one fifth, 0.3-0.5 ppmv, in October. The values of HCl from late June to early September were as small as 0-0.3 ppbv. The recovered values of HCl inside the vortex in spring (October-December) were larger than those before winter and those outside the polar vortex during the same period. $ClONO_2$ inside the vortex kept near zero even after ClO disappeared and did not recover to the level before winter until spring.

At 22 km (Figures 6 and 7), $O_3$ gradually decreased from winter to spring, but the magnitude of the decrease was much smaller than that at 18 km. The values of HCl from late June to early September were 0-1 ppbv, larger than those at 18 km.

The recovered values of HCl in spring were nearly the same as those before winter (around 2.2 ppbv). $ClONO_2$ recovered to larger values than those before winter after ClO disappeared.

As for the temporal increase of $ClONO_2$ in spring during the ClO decreasing phase, we can see a peak of 1.5 ppbv at 18 km in 2011, and at 22 km in both 2007 and 2011 around September 27 (day 270), but we see no temporal increase of $ClONO_2$ at 18 km in 2007.

Figure 7 shows that temporal ClO enhancement and decrease of $O_3$, $ClONO_2$, and $HNO_3$ occurred in early winter (May 30-June 19; day 150-170) at 22 km in 2011. This small ozone depletion event before winter may be due to an airmass movement from the polar night area to a sunlit area at lower latitudes.

## 4.2 Time series of ratios of chlorine species

In order to discuss the temporal variations of the chlorine partitioning, the ratios of observed HCl, $ClONO_2$, and ClO with respect to $Cl_y$* were calculated. Hereafter, we will discuss the ratios of chlorine species only for the cases when Syowa Station was located inside the polar vortex.

Figures 8 and 9 show the time series of the ratios of each chlorine species with respect to $Cl_y$* in 2007 (a) and in 2011 (b) at 18 km and 22 km, respectively. In these plots, HCl data from Aura/MLS were used. Note that light blue in these figures shows either $ClONO_2$ or ClO data was missing on that day, while dark blue shows all three data were available on that day. For both 2007 and 2011 at 18 km (Figure 8), HCl/$Cl_y$* was 0.6-0.8 and $ClONO_2$/$Cl_y$* was 0.2-0.3 before winter (May 10-20; day 130-140). The ratio of HCl to $Cl_y$*was three times larger than that of $ClONO_2$ at that time. ClO/$Cl_y$* increased to ~0.5 during the ClO enhanced period (the period when ClO values were more than 80 % of its maximum value: August 18-September 17; day230-260.). HCl/$Cl_y$* was 0-0.2 and $ClONO_2$/$Cl_y$* was 0-0.6 during this same period. $ClONO_2$ shows negative correlation with ClO, while HCl kept low even when ClO was low during this period. This negative correlation is shown in Figure 10. When ClO was enhanced, the $O_3$ amount gradually decreased, and finally reached <0.5 ppmv (>80% destruction) in October (October 7; day 280) (See Figures 4 and 5). The ratios to $Cl_y$* became 0.9-1.0 for HCl and 0-0.1 for $ClONO_2$ after the recovery in spring (after October 17; day 290), indicating that almost all chlorine reservoir species became HCl via reactions (R13) and/or (R14), due to the lack of $O_3$ and $NO_2$ during this period. The sum ratios (HCl + $ClONO_2$ + ClO)/$Cl_y$* were around 0.5-0.8 at the time of ClO enhanced period. The remaining chlorine is thought to be either $Cl_2O_2$ or HOCl, which will be shown in model simulation result in Section 4.6. The sum ratio (HCl + $ClONO_2$)/$Cl_y$* became close to 1 after the recovery period (after October 7; day 280).

For both 2007 and 2011 at 22 km (Figure 9), HCl/$Cl_y$* was 0.8-0.9 and $ClONO_2$/$Cl_y$* was 0.2-0.3 before winter (April 20-May 20; day 110-140). The ratio of HCl to $Cl_y$* was three to four times larger than that of $ClONO_2$. ClO/$Cl_y$* increased to 0.5-0.7 during the ClO enhanced period (August 8-28; day 220-240 in 2007, August 18-September 7; day 230-250 in 2011). HCl/$Cl_y$* was 0-0.2 and $ClONO_2$/$Cl_y$* was 0-0.6 during this period. $ClONO_2$ shows negative correlation with ClO, while HCl kept low even when ClO was low during this period as in the case at 18 km. The $O_3$ amount gradually decreased during the ClO enhanced period but kept the concentration at more than 1.5 ppmv (less than half destruction) at this altitude (See Figures

6 and 7). When the ClO enhancement ended, increase of both $ClONO_2$ and HCl occurred simultaneously in early spring (September 17-October 7; day 260-280). Then the ratios to $Cl_y$* became 0.6-0.7 for HCl and 0.3-0.4 for $ClONO_2$ in spring (after October 7; day 280). This phenomenon shows that more chlorine deactivation via reaction (R12) occurred towards $ClONO_2$ at 22 km rather than at 18 km. This is attributed to the existence of $O_3$ and $NO_2$ during this period at 22 km, which was different from the case at 18 km. The sum ratios $(HCl + ClONO_2 + ClO)/Cl_y$* was around 0.7-1.2 at the time of ClO enhanced period. The remaining chlorine is thought to be either $Cl_2O_2$ or HOCl. The sum ratio $(HCl + ClONO_2 + ClO)/Cl_y$* became around 1.2 after the recovery period (after September 27; day 270). The reason why the observed sum ratio exceed the calculated $Cl_y$* value might be because the $N_2O$-$Cl_y$ correlation from the one in equation (2) is not applicable at this altitude.

In 2011 at 18 km (Figure 8), another temporal increase of $ClONO_2$ up to a ratio of 0.4 occurred in early spring (around October 2-12; day 275-285) in accordance with HCl increase, then the $ClONO_2$ amount gradually decreased to nearly zero after late October (after October 27; day 300). This temporal increase in $ClONO_2$ could be attributed to the temporal change of the location of Syowa Station with respect to the polar vortex. Although Syowa Station was judged to be inside the polar vortex during July 14-December 16 (day 195-350) by our analysis, the difference between the equivalent latitude over Syowa Station and that at the inner edge became less than 10 degrees at around October 7 (day 280), while it was typically between 15 and 20 degrees on other days. $O_3$ and $HNO_3$ showed higher values around October 7 (day 280) (see Figure 5), indicating that Syowa Station was located close to the boundary region during this period (See Figure A2). Therefore, the temporal increase of $ClONO_2$ in 2011 at 18 km was attributed to spatial variation, not to chemical evolution.

### 4.3 Correlation between ClO and $ClONO_2$

Figure 10 shows the correlation between ClO and $ClONO_2$ during the ClO enhanced period (August 8-September 17; day 220-260) at 18 km in 2007 (a) and 2011 (b), and at 22 km in 2007 (c) and 2011 (d). In this plot, the location of Syowa Station with respect to the polar vortex (inside, the boundary region, and outside the polar vortex) is indicated by different symbols. Note that MLS ClO and MIPAS $ClONO_2$ data were sampled on the same day at the nearest orbit to Syowa Station for both satellites. The maximum differences between these two satellites' observational times and locations are 9.0 hours in time and 587 km in distance. Mean differences are 6.8 hours in time and 270 km in distance, respectively. Solid lines show regression lines obtained by Reduced Major Axis (RMA) regression. Negative correlations of slope about -1.0 between ClO and $ClONO_2$ are seen in all figures.

The negative correlation between ClO and $ClONO_2$ at Syowa Station is explained by the difference in the concentration of ClO, $NO_2$, $ClONO_2$, and $HNO_3$ inside, outside, and at the boundary region of the polar vortex around the station. Outside of the polar vortex, ClO concentration is lower and $NO_2$ concentration is higher than those inside the polar vortex. Inside the polar vortex, $HNO_3$ is taken up by PSCs and removed by the sedimentation of PSCs from the lower stratosphere (denitrification process). The $NO_x$ concentration is low because $HNO_3$ is a reservoir of $NO_x$ through the reactions;

$$NO_2 + OH + M \rightarrow HNO_3 + M, \qquad (R17)$$

$$HNO_3 + h\nu \rightarrow NO_2 + OH, \qquad (R18)$$

and

$$HNO_3 + OH \rightarrow NO_3 + H_2O. \qquad (R19)$$

The $NO_2$ concentration is low and $ClONO_2$ concentration is also low due to the consumption of $ClONO_2$ by heterogeneous reaction (R2) inside the polar vortex. In spring, the ClO amount gets high due to the activation of chlorine species by reactions (R1~R8) inside the polar vortex. At the boundary region, ClO and $NO_2$ concentrations indicate the value between inside and outside of the polar vortex, that is, the ClO concentration is much higher than that outside of the polar vortex and $NO_2$ concentration is much higher than that inside of the polar vortex. Thus, the $ClONO_2$ concentration there is elevated in August-September due to reaction (R12). This causes the negative correlation between ClO and $ClONO_2$ due to the relative distance between Syowa Station and the edge of the polar vortex. When Syowa Station was located deep inside the polar vortex, there was more ClO and less $ClONO_2$. On the contrary when Syowa Station was located near the vortex edge, there was less ClO and more $ClONO_2$. The equivalent latitude (EL) over Syowa Station was calculated as described in Appendix B for each correlation point. The EL at each correlation point is now shown by the color code in Figure 10. It generally shows the tendency, that warm coloured higher equivalent latitude points are located more towards the bottom right-hand side. This is further confirmed by 3-dimensional model simulation as shown later.

## 4.4 Comparison with model results

Figures 11 and 12 show comparisons of daily time series of simulated mixing ratios of ClO, HCl, $ClONO_2$, $Cl_y$, and $O_3$ by the MIROC3.2 Chemistry-Climate Model (CCM) (Akiyoshi et al., 2016) with FTIR, Aura/MLS, and Envisat/MIPAS measurements at 18 km and 22 km, respectively. For a description of the MIROC3.2 CCM, please see Appendix A for detail. In these figures, $Cl_y$ for Aura/MLS in the panels (d) and (i) actually represents the $Cl_y*$ value calculated by equation (2) using the $N_2O$ value measured by Aura/MLS. $Cl_y$ from the MIROC3.2 CCM is the sum of total reactive chlorines, i.e., $Cl_y = Cl + 2*Cl_2 + ClO + 2*Cl_2O_2 + OClO + HCl + HOCl + ClONO_2 + ClNO_2 + BrCl$. Note that we plotted modeled values at 12h UTC (~15h local time of Syowa Station) calculated by the MIROC3.2 CCM in order to compare the daytime measurements of FTIR and satellites. In Figures 11(b), (d), (g), and (i), modeled HCl and $Cl_y$ are systematically smaller by 20-40% compared with FTIR or MLS measurements. The cause of this discrepancy may be partly due to either smaller downward advection and/or faster horizontal mixing of airmass across the subtropical barrier in MIROC3.2 CCM (Akiyoshi et al., 2016). Nevertheless, evolutions of measured ClO and $ClONO_2$ for the period are well simulated by the MIROC3.2 CCM. Modeled $O_3$ were in very good agreement with FTIR and/or MLS measurements throughout the year in both altitudes for both years. Hereafter, the result of MIROC3.2 CCM at 50 hPa (~18 km) is discussed.

## 4.5 Polar distribution of minor species

Figure 13 shows distributions of temperature from the model nudged toward the ERA-Interim data, and simulated mixing ratios of $O_3$, $NO_2$, $HNO_3$, ClO, HCl, and $ClONO_2$ by the MIROC3.2 CCM at 50 hPa for June 24 (day 175), September 2 (day 245), September 6 (day 249), and October 6 (day 279) in 2007. Polar vortex edges defined by the method described in

Appendix B were plotted with white circles. The location of Syowa Station is shown by a white star in each panel. On June 24 (day 175), stratospheric temperatures over Antarctica were already low enough for the onset of heterogeneous chemistry. Consequently, $NO_2$ was converted into $HNO_3$ via reaction (R17), and $HNO_3$ in the polar vortex was condensed onto PSCs. Note that the depleted area of $NO_2$ was greater than that of $HNO_3$. This is due to the occurrence of reaction (R12) that converts ClO and $NO_2$ into $ClONO_2$ at the edge of the polar vortex, which is shown by the enhanced $ClONO_2$ area at the vortex edge in Figure 13. Also, HCl and $ClONO_2$ are depleted in the polar vortex due to the heterogeneous reactions (R1), (R2), (R3), and (R4) on the surface of PSCs and aerosols. Some HCl remains near the core of the polar vortex, because the initial amount of the counter-part of heterogeneous reaction (R1) ($ClONO_2$) was less than that of HCl, as was also shown by CLaMS, SD-WACCM, and TOMCAT/SLIMCAT model simulations by Grooß et al. (2018). The $O_3$ amount was only slightly depleted within the polar vortex on this day.

On September 2 (day 245), amounts of $NO_2$, $HNO_3$, HCl, and $ClONO_2$ all show very depleted values in the polar vortex. The amount of ClO shows some enhanced values inside the polar vortex. Development of ozone depletion was seen in the polar vortex. Note that $ClONO_2$ shows enhanced values around the boundary region of the polar vortex. This might be due to reaction (R12) at this location. On this day (day 245), Syowa Station was located inside the polar vortex close to the vortex edge, where ClO was smaller and $ClONO_2$ was greater than the values deep inside the polar vortex as observed and indicated by upper left circle with cross in Figure 10 (a).

On September 6 (day 249), most features were the same as on September 2, but the shape of the polar vortex was different. Consequently, Syowa Station was located deep inside the polar vortex, where ClO was greater and $ClONO_2$ was smaller than the values around the boundary region of the polar vortex as observed and indicated by lower right circle with cross in Figure 10 (a). Hence, the negative correlation between ClO and $ClONO_2$ seen in Figure 10 was due to variation of the relative distance between Syowa Station and the edge of the polar vortex.

As for HCl, it kept near zero not only on this day (September 6) but also on September 2 when Syowa Station was located inside the polar vortex close to the vortex edge. Therefore, observed day-to-day variations of HCl were small and did not show any correlation with ClO (see Figures 4-7). A possible explanation to keep a near zero HCl value close to the vortex edge is due to a so-called "HCl-null cycles" which was started with fast reaction (R13) proposed by Müller et al. (2018). This cycle is discussed later.

On October 6 (day 279), ClO enhancement has almost disappeared. Inside the polar vortex, $O_3$, $NO_2$, $HNO_3$, and $ClONO_2$ showed very low values. Ozone was almost fully destroyed at this altitude in the polar vortex. However, the amount of HCl increased deep inside the polar vortex. This might be due to the recovery of HCl by reactions (R13) and/or (R14) deep inside the polar vortex, where there was no $O_3$ or $NO_2$ left and reactions (R13) and/or (R14) were favoured compared with reaction (R12). Syowa Station was located deep inside the polar vortex and the simulated and observed amounts of HCl were both more than ten times greater than those of $ClONO_2$ on this day (see Figure 4).

Figure 14 shows distributions of temperature from the model nudged toward the ERA-Interim data, and simulated mixing ratios of $O_3$, $NO_2$, $HNO_3$, ClO, HCl, and $ClONO_2$ by the MIROC3.2 CCM at 50 hPa for July 5 (day 186), August 19 (day 231),

August 21 (day 233), and October 9 (day 282) in 2011. Polar vortex edges and location of Syowa Station were also plotted. On July 5 (day 186), the situation was similar to that of June 24 (day175) in 2007. Note that inner edge of the polar vortex was defined on this day. Syowa Station was located deeper inside the polar vortex on July 5 in 2011 than on June 24 in 2007 and remaining HCl was observed by MLS (see Figure 5).

On August 19 (day 231) and August 21 (day 233), the situations were similar to those of September 2 (day 245) and September 6 (day 249) in 2007, respectively. ClO and $ClONO_2$ correlations on these days are also indicated by circles with crosses in Figure 10 (b).

On October 9 (day 282), the situation was similar to that of October 6 (day 279) in 2007, but Syowa Station was located inside the polar vortex closer to the inner vortex edge than in 2007. The recovery of $ClONO_2$ by reaction (R12) was simulated and observed at Syowa Station besides the recovery of HCl by reaction (R13) (see Figure 8 (b)), because there were some $O_3$ and $NO_2$ near the inner vortex edge (see Figure 5). This shows the phenomena described on the last paragraph in Section 4.2.

## 4.6 Time evolution of chlorine species from CCM and discussion

Three-hourly time series of zonal-mean active chlorine species, $Cl_2O_2$ (b), $Cl_2$ (c), ClO (d), and their sum $(ClO+2*Cl_2O_2+2*Cl_2)$ (a), HOCl (e), and chlorine reservoir species HCl (f) and $ClONO_2$ (g) modeled by MIROC3.2 CCM at 68.4°S, 71.2°S, 76.7°S, and 87.9°S in 2007 are plotted in Figure 15. The dates on which the distribution of each species is shown in Figure 13 are indicated by vertical dotted lines. In Figure 15, it is shown that HCl and $ClONO_2$ rapidly decreased at around May 10 (day 130) at 87.9°S due to heterogeneous reaction (R1), when PSCs started to form in the Antarctic polar vortex (Figures 15(f) and 15(g)). Consequently, $Cl_2$ was formed (Figure 15(c)). Similar chlorine activation was seen at 76.7°S about 5-10 days later than at 87.9°S. The decrease of HCl stopped when the counter-part of the heterogeneous reaction (R1) ($ClONO_2$) was missing at around May 20 (day 140). Continuous loss of HCl occurred from June to July (day 160-200). The possible cause of this loss will be discussed later. Gradual conversion from $Cl_2$ into $Cl_2O_2$ (ClO-dimer) was seen at all latitudes at around May 30-June 9 (day 150-160) (Figures 15(b) and 15(c)) through reactions (R5), (R8), and (R9). At 87.9°S, conversion from $Cl_2$ to $Cl_2O_2$ was slow, due to lack of sunlight which is needed for reaction (R5). Increase of ClO occurred much later in winter (July 9; day 190 or later), because sun light is needed to form ClO by reactions (R5) and (R8) in the polar vortex (Figure 15(d)). Nevertheless, there were some enhancements of ClO in early winter, June 24 (day 175), simulated at the edge of the polar vortex (Figure 13) where there was some sunlight available due to the distortion of the shape of the polar vortex. Increase of ClO occurred from lower latitude (68.4°S) at around July 14 (day 195), towards higher latitude (87.9°S) at around September 12 (day 255) (Figure 15(d)). Diurnal variation of ClO was also seen at latitudes between 68.4°S and 76.7°S. When the stratospheric temperature increased above NAT saturation temperature at around September 27 (day 270) (Figure 3(b)), chlorine activation ended, and ClO was mainly converted into HCl at all latitudes inside the polar vortex (Figures 15(d) and 15(f)). This is because reactions (R13) and/or (R14) occur more frequently than reaction (R12) inside the polar vortex due to the depleted $O_3$ amount there, as was described in Section 1 (Douglass et al., 1995). Increase of HOCl due to heterogeneous reaction (R2) on the surface of PSCs occurred gradually from June at lower latitudes (68.4°S and 71.2°S)

(Figure 15(e)). It also occurred at 76.7°S from July, and at 87.9°S from August. The cause of HOCl increase at 87.9°S from August is not clear at the moment. In Figure 15, the species which decreased at 87.9°S from August was $Cl_2$ (Figure 15 (c)). If sunlight was available, $Cl_2$ was converted into HOCl through reactions (R5), (R8), and the following reaction

$$ClO + HO_2 \rightarrow HOCl + O_2. \qquad (R20)$$

Here, $HO_2$ was needed to yield HOCl. One possibility to yield $HO_2$ in August is either one of "HCl null cycles" C1 or C2 (see Appendix C), or "HCl destruction cycles" C3 or C4 (see Appendix C) which was described in Müller et al. (2018). If the airmass at 87.9°S was located equatorward due to the obliqueness of the polar vortex a few days earlier, then sunlight may have been available and such reactions could yield HOCl at 87.9°S.

    Continuous loss of HCl was seen at 87.9°S between June 9 (day 160) and July 19 (day 200) even after the disappearance of

10 the counterpart of heterogeneous reaction (R1) (Figure 15(f)). The cause of this continuous loss was unknown until recently, where a hypothesis was proposed that includes the effect of decomposition of particulate $HNO_3$ by some processes like ionisation of air molecules caused by galactic cosmic rays during the winter polar vortex or photolysis of dissolved $HNO_3$ in PSC particles (Grooß et al., 2018). However, they concluded that these processes could not explain the HCl discrepancy between their models and observations. We consider that the continuous loss of HCl was caused by the mixing of $ClONO_2$-

15 containing air with the air at the core of polar vortex, due to the excursions of air parcels in and out of sunlight during the winter caused by the distortion of polar vortex, which photochemically resupply $ClONO_2$ and HOCl. Our result was indicated by some sporadic increase in $ClONO_2$ at around June 7 (day 158), June 28 (day 179), and July 8 (day 189) at 76.7°S as shown in Figure 15(g). We confirmed that the shapes of polar vortex were rather distorted on these days. Subsequently, HCl losses were observed at 76.7°S and 87.9°S during these episodes in Figure 15(f). Thus, the continuous loss of HCl at the most polar

latitude (87.9°S) could be due to the gradual mixing of air within the polar vortex during the winter period, when polar vortex was still strong.

    The $ClONO_2$ distribution on June 24, 2007 in Figure 13 shows a peak at around the polar vortex outer edge (equivalent latitude of ~62°S) where there was no HCl. We confirmed that such a peak continued to exist from the middle of June to the end of July, when sunlight was not available in geographical latitude higher than 70°S due to polar night. However, substantial

amount of $ClONO_2$ exists between equivalent latitude of 62°S (polar vortex outer edge) and 70°S, mainly due to the distortion of the polar vortex and availability of $O_3$. At 62°S equivalent latitude, the amount of $ClONO_2$ has a peak around these days, while HCl amounts were low. Around 65-70°S, HCl was almost fully depleted. Therefore, if the airmass which contains some $ClONO_2$ travelled toward poleward due to mixing, it can react with HCl by reaction (R1) and continuously destroy HCl in the core of the polar vortex during polar night in June and July. Another explanation of the loss of HCl is by heterogeneous

reaction (R4) on the surface of PSCs with HOCl. Spiky increases of HOCl at 76.7°S and 87.9°S, and simultaneous decrease of HCl occurred at around July 7 (day 188) and July 20 (day 201) in Figures 15(e) and 15(f). Continuous loss of HCl at the core of the polar vortex in August and September was recently proposed by Müller et al. (2018), that "HCl destruction cycles" C3 and C4 (See Appendix C) are responsible for the decline of HCl in the vortex core. These chemical cycles also require sun light to occur, which may not be available in June and July at the vortex core. Therefore, the distortion of the polar vortex is

also important for that reaction to occur as well. We consider that the reactions of HCl with both $ClONO_2$ and HOCl contribute the continuous loss of HCl during winter in the vortex core.

Recently, Müller et al. (2018) discussed on "race" between chlorine activation and deactivation to maintain enhanced levels of active chlorine during the time period (September and early October) when rapid ozone loss occurs. They proposed a so-called "HCl null-cycles" to keep enhanced chlorine levels. They proposed a mechanism that the formation HCl (R13) is followed by immediate reactivation of HCl by null-cycles C1 and C2 (Müller et al., 2018). Our MIROC3.2 CCM model results support the mechanism by high ClO and HOCl levels in September as shown in Figures 15(d) and 15(e).

## 5. Conclusions

Lower stratospheric vertical profiles of $O_3$, $HNO_3$, and HCl were retrieved using SFIT2 from solar spectra taken with a ground-based FTIR installed at Syowa Station, Antarctica from March to December 2007 and September to November 2011. This was the first continuous measurements of chlorine species throughout the ozone hole period from the ground in Antarctica. Retrieved profiles were validated with Aura/MLS and ozonesonde data. The absolute differences between FTIR and Aura/MLS or ozonesonde measurements were within measurement error bars at the altitudes of interest.

To study the temporal variation of chlorine partitioning and ozone destruction from fall to spring in the Antarctic polar vortex, we analyzed temporal variations of measured minor species by FTIR over Syowa Station combined with satellite measurements of ClO, HCl, $ClONO_2$ and $HNO_3$. When the stratospheric temperature over Syowa Station fell ~4K below NAT saturation temperature, PSCs started to form and heterogeneous reaction between HCl and $ClONO_2$ occurred and both HCl and $ClONO_2$ were almost completely depleted at both 18 km and 22 km in early winter. When the sun came back to the Antarctic in spring, enhancement of ClO and gradual $O_3$ destruction were observed. During the ClO enhanced period, negative correlation between ClO and $ClONO_2$ was observed in the time-series of the data at Syowa Station. This negative correlation is associated with the relative distance between Syowa Station and the edge of the polar vortex.

To see the behavior of whole chlorine and related species inside the polar vortex and the boundary region in more detail, results of MIROC3.2 CCM simulation were analyzed. Direct comparison between CCM results and observations show good day-to-day agreement in general, although some species show systematic differences especially at 18 km. The modeled $O_3$ is in good agreement with FTIR and satellite observations. Rapid conversion of chlorine reservoir species (HCl and $ClONO_2$) into $Cl_2$, gradual conversion of $Cl_2$ into $Cl_2O_2$, increase of HOCl in winter period, increase of ClO when sunlight became available, and conversion of ClO into HCl were successfully reproduced by the CCM. HCl decrease in the winter polar vortex core continued to occur due to both transport of $ClONO_2$ from the polar vortex boundary region to higher latitudes, providing a flux of $ClONO_2$ from more sunlit latitudes into the polar vortex, and the heterogeneous reaction with HOCl. Temporal variation of chlorine species over Syowa Station was affected both by heterogeneous chemistries related to PSC occurrence

inside the polar vortex, and transport of $NO_x$-rich airmass from the polar vortex boundary region, which can produce additional $ClONO_2$ by reaction (R12).

The deactivation pathways from active ClO into reservoir species (HCl and/or $ClONO_2$) were confirmed to be very dependent on the availability of ambient $O_3$. At an altitude (18 km) where most ozone was depleted in the Antarctic, most ClO was converted to HCl. However, at an altitude (22 km) when there was some $O_3$ available, additional increase of $ClONO_2$ than pre-winter value can occur, as in the case in the Arctic, through reactions (R15) and (R12) (Douglass et al., 1995; Grooß et al., 1997).

*Data availability:* The FTIR data presented here can be obtained in electronic form (hdf files) from the following DOIs: doi:10.17595/20190911.001 (FTIR data for 2007), doi:10.17595/20190911.002 (FTIR data for 2011).

The MIROC3.2 CCM outputs are from the REF-C1SD simulation data from the CCMI, which are stored at the CCMI site of BADC at:

http://badc.nerc.ac.uk/browse/badc/wcrp-ccmi/data/CCMI-1/output/NIES.

The MLS data were taken from the following site: http://avdc.gsfc.nasa.gov/index.php?site=2045907950.

The MIPAS data were taken from the following site: http://share.lsdf.kit.edu/imk/asf/sat/mipas-export/Data_by_Target/.

The CALIOP data were taken from the following site: https://www-calipso.larc.nasa.gov/resources/calipso_users_guide/data_summaries/psc/index.php.

*Author contributions.* HN, IM, YN, and MT conceived and worked on the current research project. HN, KS, and TK made FTIR observations at Syowa Station in 2007 and 2011. HN, YN, KS, and NBJ conducted the SFIT2 retrievals. HA conducted MIROC3.2 CCM simulations and ED analysed them. YT performed polar vortex categorization calculation. HN, IM, YN, HA, YT, and NBJ contributed to the interpretation of the results and wrote the paper.

*Competing interests.* The authors declare that they have no competing financial interests.

## Appendix A: MIROC3.2 nudged chemistry–climate model

The chemistry-climate model (CCM) used in this study was MIROC3.2 CCM, which was developed on the basis of version 3.2 of the Model for Interdisciplinary Research on Climate (MIROC3.2) general circulation model (GCM). The MIROC3.2 CCM introduces the stratospheric chemistry module of the old version of the CCM that was used for simulations proposed by the chemistry–climate model validation (CCMVal) and the second round of CCMVal (CCMVal2) (WMO, 2007, 2011; SPARC CCMVal, 2010; Akiyoshi et al., 2009, 2010). The MIROC3.2 CCM is a spectral model with a T42 horizontal

resolution (2.8° × 2.8°) and 34 vertical atmospheric layers above the surface. The top layer is located at approximately 80 km (0.01 hPa). Hybrid sigma–pressure coordinates are used for the vertical coordinate. The horizontal wind velocity and temperature in the CCM were nudged toward the ERA–Interim data (Dee et al., 2011) to simulate global distributions of ozone and other chemical constituents on a daily basis. The transport is calculated by a semi–Lagrangian scheme (Lin and Rood, 1996). The chemical constituents included in this model are $O_x$, $HO_x$, $NO_x$, $ClO_x$, $BrO_x$, hydrocarbons for methane oxidation, heterogeneous reactions on the surface for sulfuric-acid aerosols, supercooled ternary solutions, nitric-acid trihydrate, and ice particles. The CCM contains 13 heterogeneous reactions on multiple aerosol types (Akiyoshi, 2007) as well as gas-phase chemical reactions and photolysis reactions. The surface of the particles for the heterogeneous reactions are calculated from the volume of condensation, assuming number density of the particles and the size distributions. Sedimentation of the particles is considered. The reaction-rate and absorption coefficients are based on JPL–2010 (Sander et al., 2011). Family method is used to calculate gas phase chemical reactions. The time integrations for the families and heterogeneous reactions are performed explicitly. The time step for the chemistry scheme is 6 minutes. A scheme of spherical geometry for radiation transfer was developed (Kurokawa et al., 2005) and used for radiation transfer calculation in the CCM. The photolysis rates of chemical constituents are calculated online, using the radiation flux in the CCM with 32 spectral bins. All the 42 photolysis reactions, 140 gas phase reactions, and 13 heterogeneous reactions are summarized in Table A2. See Akiyoshi et al. (2016) and Supplement of Morgenstern et al. (2017) for more details.

### Appendix B: Determined polar vortex edges

Inner and outer edges of the polar vortex were determined as follows:

1) Equivalent latitudes (EL) (McIntyre and Palmer, 1984; Butchart and Remsberg, 1986) were computed based on isentropic potential vorticity at 450 K and 560 K isentropic surfaces for 18 km and 22 km using the ERA-Interim reanalysis data (Dee et al., 2011), respectively.

2) Inner and outer edges (at least 5º apart from each other) of the polar vortex were defined by local maxima of the isentropic potential vorticity gradient with respect to equivalent latitude only when a tangential wind speed (i.e., mean horizontal wind speed along the isentropic potential vorticity contour; see Eq. (1) of Tomikawa and Sato (2003)) near the vortex edge exceeds a threshold value (i.e., 20 m s$^{-1}$, see Nash et al. (1996) and Tomikawa et al. (2015)).

3) Then, the polar region was divided into three regions; i.e., inside the polar vortex (inside of inner edge), the boundary region (between inner and outer edges), and outside the polar vortex (outside of outer edge) when there were two polar vortex edges. When there was only one edge, the polar region was divided into two regions; i.e., inside the polar vortex and outside the polar vortex.

Figures A1 and A2 show time-equivalent latitude sections of modified potential vorticity (MPV) and its gradient with respect to EL at 450 and 560 K isentropic potential temperature (PT) surfaces in 2007 and 2011, respectively. MPV is a scaled PV to

remove its exponential increase with height (cf., Lait, 1994).  The inner and outer edge(s) are plotted by black dots, while the ELs of Syowa Station on those days are plotted by red dots.  It can be seen that inner edges were first formed at around EL of 70 degrees in April, and outer edges started to form at around EL of 55 degrees in July-August, emerging the boundary region.  Then, those two edges converge into one edge at around EL of 60 degrees in November.  Finally, the polar vortex edge

5    disappeared in December.  Syowa Station was mostly located inside the polar vortex, but sometimes located at the boundary region or outside the polar vortex, depending on different PT levels.

## Appendix C:  "HCl null cycles" C1, C2 and "HCl destruction cycles" C3, C4

In Müller et al. (2018), new chemical cycles C1 and C2 were proposed to maintain high levels of active chlorine in Antarctic

10    spring, referred to as "HCl null cycles", as follows:

$$CH_4 + Cl \rightarrow HCl + CH_3 \qquad\qquad (R13)$$
$$CH_3 + O_2 + M \rightarrow CH_3O_2 + M \qquad\qquad (R21)$$
$$CH_3O_2 + ClO \rightarrow CH_3O + Cl + O_2 \qquad\qquad (R22)$$
$$CH_3O + O_2 \rightarrow HO_2 + CH_2O \qquad\qquad (R23)$$

$$ClO + HO_2 \rightarrow HOCl + O_2 \qquad\qquad (R24)$$
$$HOCl + HCl \rightarrow Cl_2 + H_2O \qquad\qquad (R4)$$
$$Cl_2 + h\nu \rightarrow 2Cl \qquad\qquad (R5)$$
$$Cl + O_3 \rightarrow ClO + O_2 \qquad (2\times) \qquad\qquad (R8)$$
$$\text{Net(C1)} \; CH_4 + 2O_2 \rightarrow CH_2O + H_2O + 2O_2 \qquad\qquad (R25)$$

$$CH_2O + Cl \rightarrow HCl + CHO \qquad\qquad (R26)$$
$$CHO + O_2 \rightarrow CO + HO_2 \qquad\qquad (R27)$$
$$ClO + HO_2 \rightarrow HOCl + O_2 \qquad\qquad (R24)$$
$$HOCl + HCl \rightarrow Cl_2 + H_2O \qquad\qquad (R4)$$

$$Cl_2 + h\nu \rightarrow 2Cl \qquad\qquad (R5)$$
$$Cl + O_3 \rightarrow ClO + O_2 \qquad\qquad (R8)$$
$$\text{Net(C2)} \; CH_2O + O_3 \rightarrow CO + H_2O + O_2 \qquad\qquad (R28)$$

Also, other chemical cycles which are responsible for the decline of HCl during Antarctic August and September, referred

30    to "HCl destruction cycles", are also proposed by Müller et al. (2018) as follows:

$$CH_2O + h\nu \rightarrow CHO + H \qquad\qquad (R29)$$
$$H + O_2 + M \rightarrow HO_2 + M \qquad\qquad (R30)$$

$$CHO + O_2 \rightarrow CO + HO_2 \qquad\qquad (R27)$$

$$ClO + HO_2 \rightarrow HOCl + O_2 \ (2\times) \qquad\qquad (R24)$$

$$HOCl + HCl \rightarrow Cl_2 + H_2O \ (2\times) \qquad\qquad (R4)$$

$$Cl_2 + h\nu \rightarrow 2Cl \qquad (2\times) \qquad\qquad (R5)$$

$$Cl + O_3 \rightarrow ClO + O_2 \ (4\times) \qquad\qquad (R8)$$

Net(C3) $CH_2O + 2HCl + 4O_3 \rightarrow CO + 2ClO + 2H_2O + 4O_2$ (R31)

$$O_3 + h\nu \rightarrow O(^1D) + O_2 \qquad\qquad (R32)$$

$$O(^1D) + H_2O \rightarrow 2OH \qquad\qquad (R33)$$

$$OH + O_3 \rightarrow HO_2 + O_2 \qquad (2\times) \qquad\qquad (R34)$$

$$ClO + HO_2 \rightarrow HOCl + O_2 \ (2\times) \qquad\qquad (R24)$$

$$HOCl + HCl \rightarrow Cl_2 + H_2O \ (2\times) \qquad\qquad (R4)$$

$$Cl_2 + h\nu \rightarrow 2Cl \ (2\times) \qquad\qquad (R5)$$

$$Cl + O_3 \rightarrow ClO + O_2 \qquad (4\times) \qquad\qquad (R8)$$

Net(C4) $2HCl + 7O_3 \rightarrow 2ClO + H_2O + 9O_2$ $\qquad\qquad$ (R35)

**Acknowledgments**

We acknowledge all the members of the 48[th] Japanese Antarctic Research Expedition (JARE-48), JARE-52 for their support in making the FTIR observation at Syowa Station.  Data provision of Aura/MLS and Envisat/MIPAS are much appreciated. We thank Japan Meteorological Agency for providing the ozonesonde and Rawin sonde data at Syowa Station.  Thanks are due to Dr. Yosuke Yamashita for performing the CCM run.  The model computations were performed on NEC-SX9/A(ECO) and NEC SX-ACE computers at the CGER, NIES, supported by the Environment Research and Technology Development

Funds of the Ministry of the Environment (2-1303) and Environment Restoration and Conservation Agency (2-1709).  We thank Dr. Takafumi Sugita for useful discussion and comments.

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

**Tables**

Table 1. Retrieval parameters of SFIT2

| Species | $O_3$ | $HNO_3$ | HCl |
|---|---|---|---|
| Spectroscopy | HITRAN 2008 | HITRAN 2008 | HITRAN 2008 |
| Pressure and temperature profile | Daily sonde (0-30 km) CIRA 86 (30-100 km) | Daily sonde (0-30 km) CIRA 86 (30-100 km) | Daily sonde (0-30 km) CIRA 86 (30-100 km) |
| A priori profiles | Monthly averaged by ozonesonde (0-30 km) & ILAS-II (30-100 km) | Monthly averaged by ILAS-II | Monthly averaged by HALOE |
| Microwindows (cm$^{-1}$) | 1002.578 – 1003.500 1003.900 – 1004.400 1004.578 – 1005.000 | 867.000 – 869.591 872.800 – 874.000 | 2727.730 – 2727.830 2775.700 – 2775.800 2925.800 – 2926.000 |
| Retrieved interfering species | $O_3$ (668), $O_3$ (686), $CO_2$, $H_2O$ | $H_2O$, OCS, $NH_3$, $CO_2$, $C_2H_6$ | $CO_2$, $H_2O$, $O_3$, $NO_2$ |
| Typical retrieval error (%) for 15-25 km | 5 | 15 | 17 |
| Typical vertical resolution (km) | 5 | 7 | 6 |
| Mean degrees of freedom (DOFS) | 4.9 | 2.8 | 2.3 |

**Table 2.  Summary of validation results of FTIR profiles compared with ozonesonde and Aura/MLS measurements, and possible Aura/MLS biases from literatures**

| | Number of coincidences | Root mean squares of official errors* (%) at 18-22 km | D (%) at 18-22 km | Min/Max (%) at 18-22 km | Range of mean absolute differences for 15-25 km (ppmv/ppbv) | Literature values |
|---|---|---|---|---|---|---|
| $O_3$ (sonde) | 14 | 7.1 | +6.1 | -10.4/+19.2 | -0.02~+0.40 | |
| $O_3$ (MLS) | 33 | 9.4 | -5.5 | -16.3/+4.5 | -0.13~+0.16 | Aura/MLS is +8% higher than ACE-FTS at 70°S (Froidevoux et al., 2008) |
| $HNO_3$ | 47 | 19.2 | +13.2 | +0.2/+21.9 | -0.56~+0.57 | Aura/MLS no bias with errors (0.6 ppbv) (Livesey et al., 2011) |
| HCl | 50 | 39.5 | -9.7 | -14.6/-3.0 | -0.20~+0.09 | Aura/MLS > HALOE by 10-15%, precision 0.2-0.6 ppbv (Livesey et al., 2013) |

*Root mean squares of official absolute and relative errors given by each data set.

**Table A1. FTIR observation dates at Syowa Station in 2007 and 2011**

| Month | Dates (2007) | Dates (2011) | Number of days inside the polar vortex (2007/2011) | Number of days in the boundary region of the polar vortex (2007/2011) | Number of days outside the polar vortex (2007/2011) | Number of measurement days (2007/2011) |
|---|---|---|---|---|---|---|
| March | 25 | | 0 / 0 | 0 / 0 | 1 / 0 | 1 / 0 |
| April | 1, 3, 4, 5, 8, 24, 26, 28 | | 0 / 0 | 0 / 0 | 8 / 0 | 8 / 0 |
| May | 8, 9, 10, 13, 14, 15, 20, 21, 22 | | 7 / 0 | 0 / 0 | 2 / 0 | 9 / 0 |
| June | | | 0 / 0 | 0 / 0 | 0 / 0 | 0 / 0 |
| July | 29, 30 | | 0 / 0 | 2 / 0 | 0 / 0 | 2 / 0 |
| August | 1, 8, 9, 10, 24, 25, 26, 28, 29 | | 8 / 0 | 1 / 0 | 0 / 0 | 9 / 0 |
| September | 1, 4, 5, 6, 7, 8, 16, 18, 23, 26, 27, 30 | 25, 29, 30 | 12 / 3 | 0 / 0 | 0 / 0 | 12 / 3 |
| October | 6, 10, 11, 14, 19, 20, 25, 26, 27 | 1, 3, 4, 8, 11, 22, 23, 24, 26 | 9 / 9 | 0 / 0 | 0 / 0 | 9 / 9 |
| November | 2, 3, 5, 6, 7, 8, 9, 10, 11, 16, 17, 18, 19, 21, 27, 29, 30 | 1, 2, 3, 9, 11, 16, 19 | 12 / 7 | 1 / 0 | 4 / 0 | 17 / 7 |
| December | 4, 7, 8, 9, 13, 15, 16, 17, 20, 22, 29 | | 8 / 0 | 0 / 0 | 3 / 0 | 11 / 0 |
| Total | | | 56 / 19 | 4 / 0 | 18 / 0 | 78 / 19 |

**Table A2. Chemical reactions in the MIROC3.2 Chemistry-Climate Model**

| (1) Photolysis reactions | Reaction coefficients (reference) |
|---|---|
| $O_2 + h\nu \rightarrow O(1D) + O$ | JPL-2010 |
| $O_2 + h\nu \rightarrow 2O$ | JPL-2010 |
| $O_3 + h\nu \rightarrow O(1D) + O_2$ | JPL-2010 |
| $O_3 + h\nu \rightarrow O + O_2$ | JPL-2010 |
| $H_2O + h\nu \rightarrow OH + H$ | JPL-2010 |
| $CH_3OOH + h\nu \rightarrow CH_3O + OH$ | JPL-2010 |
| $CH_2O + h\nu \rightarrow CHO + H$ | JPL-2010 |
| $CH_2O + h\nu \rightarrow CO + H_2$ | JPL-2010 |
| $H_2O_2 + h\nu \rightarrow 2OH$ | JPL-2010 |
| $N_2O + h\nu \rightarrow N_2 + O(1D)$ | JPL-2010 |
| $NO + h\nu \rightarrow N + O$ | JPL-2010 |
| $NO_3 + h\nu \rightarrow NO_2 + O$ | JPL-2010 |
| $NO_3 + h\nu \rightarrow NO + O_2$ | JPL-2010 |
| $HNO_3 + h\nu \rightarrow OH + NO_2$ | JPL-2010 |
| $HNO_4 + h\nu \rightarrow OH + NO_3$ | JPL-2010 |
| $HNO_4 + h\nu \rightarrow HO_2 + NO_2$ | JPL-2010 |
| $N_2O_5 + h\nu \rightarrow NO_2 + NO_3$ | JPL-2010 |
| $HCl + h\nu \rightarrow H + Cl$ | JPL-2010 |
| $HOCl + h\nu \rightarrow OH + Cl$ | JPL-2010 |
| $ClONO_2 + h\nu \rightarrow Cl + NO_3$ | JPL-2010 |
| $ClONO_2 + h\nu \rightarrow O + ClONO$ | JPL-2010 |
| $Cl_2O_2 + h\nu \rightarrow ClOO + Cl$ | JPL-2010 |
| $OClO + h\nu \rightarrow O + ClO$ | JPL-2010 |
| $Cl_2 + h\nu \rightarrow 2Cl$ | JPL-2010 |
| $BrONO_2 + h\nu \rightarrow BrO + NO_2$ | JPL-2010 |
| $BrONO_2 + h\nu \rightarrow Br + NO_3$ | JPL-2010 |
| $BrCl + h\nu \rightarrow Br + Cl$ | JPL-2010 |
| $HOBr + h\nu \rightarrow Br + OH$ | JPL-2010 |
| $BrO + h\nu \rightarrow Br + O$ | JPL-2010 |
| $HBr + h\nu \rightarrow Br + H$ | JPL-2010 |
| $Br_2 + h\nu \rightarrow 2Br$ | JPL-2010 |
| $CCl_4 + h\nu \rightarrow 4Cl + products$ | JPL-2010 |
| $CH_3CCl_3 + h\nu \rightarrow 3Cl + products$ | JPL-2010 |
| $CH_3Cl + h\nu \rightarrow Cl + products$ | JPL-2010 |
| $CFCl_3(CFC-11) + h\nu \rightarrow 3Cl + products$ | JPL-2010 |
| $CF_2Cl_2(CFC-12) + h\nu \rightarrow 2Cl + products$ | JPL-2010 |
| $CClF_2CCl_2F(CFC-113) + h\nu \rightarrow 3Cl + products$ | JPL-2010 |
| $CHClF_2(HCFC-22) + h\nu \rightarrow Cl + products$ | JPL-2010 |
| $CHBr_3 + h\nu \rightarrow 3Br + products$ | JPL-2010 |
| $CH_3Br + h\nu \rightarrow Br + products$ | JPL-2010 |
| $CF_2ClBr(Halon-1211) + h\nu \rightarrow Br + Cl + products$ | JPL-2010 |
| $CF_3Br(Halon-1301) + h\nu \rightarrow Br + products$ | JPL-2010 |
| **(2) Gas phase reactions** | **Reaction coefficients (reference)** |

| | |
|---|---|
| $O + O + M \rightarrow O_2 + M$ | Brasseur and Solomon (1986) |
| $O + O_2 + M \rightarrow O_3 + M$ | see JPL-2010 |
| $O + O_3 \rightarrow O_2 + O_2$ | $8.0\times10^{-12}$ exp[-2060/T] |
| $O(1D) + N_2 \rightarrow O + N_2$ | $2.15\times10^{-11}$ exp[110/T] |
| $O(1D) + O_2 \rightarrow O + O_2$ | $3.3\times10^{-11}$ exp[55/T] |
| $H_2O + O(1D) \rightarrow 2OH$ | $1.63\times10^{-10}$ exp[60/T] |
| $H_2 + O(1D) \rightarrow H + OH$ | $1.2\times10^{-10}$ |
| $CH_4 + O(1D) \rightarrow CH_3 + OH$ | $1.75\times10^{-10}$ |
| $CH_4 + OH \rightarrow CH_3 + H_2O$ | see JPL-2010 |
| $CH_3 + O_2 + M \rightarrow CH_3O_2 + M$ | see JPL-2010 |
| $CH_3 + O \rightarrow CH_2O + H$ | $1.1\times10^{-10}$ |
| $CH_3O_2 + CH_3O_2 \rightarrow$ products | $9.5\times10^{-14}$ exp[390/T] |
| $CH_3O_2 + HO_2 \rightarrow CH_3OOH + O_2$ | $4.1\times10^{-13}$ exp[750/T] |
| $OH + CH_3OOH \rightarrow CH_3O_2 + H_2O$ | $3.8\times10^{-12}$ exp[200/T] |
| $CH_3O_2 + NO \rightarrow CH_3O + NO_2$ | $2.8\times10^{-12}$ exp[300/T] |
| $CH_3O + O_2 \rightarrow CH_2O + HO_2$ | $3.9\times10^{-14}$ exp[-900/T] |
| $CH_2O + OH \rightarrow CHO + H_2O$ | $5.5\times10^{-12}$ exp[125/T] |
| $CH_2O + O \rightarrow$ products ($CHO + OH$, $H + HCO_2$) | $3.4\times10^{-11}$ exp[-1600/T] |
| $CHO + O_2 \rightarrow CO + HO_2$ | $5.2\times10^{-12}$ |
| $CO + OH \rightarrow H + CO_2$ | see JPL-2010 |
| $OH + HO_2 \rightarrow H_2O + O_2$ | $4.8\times10^{-11}$ exp[250/T] |
| $H + HO_2 \rightarrow$ products | $8.05\times10^{-11}$ |
| $H + HO_2 \rightarrow H_2O + O$ | $1.6\times10^{-12}$ |
| $O_3 + OH \rightarrow HO_2 + O_2$ | $1.7\times10^{-12}$ exp[-940/T] |
| $O + OH \rightarrow H + O_2$ | $1.8\times10^{-11}$ exp[180/T] |
| $H + O_3 \rightarrow OH + O_2$ | $1.4\times10^{-10}$ exp[-470/T] |
| $H + O_2 + M \rightarrow HO_2 + M$ | see JPL-2010 |
| $HO_2 + O \rightarrow OH + O_2$ | $3.0\times10^{-11}$ exp[200/T] |
| $HO_2 + O_3 \rightarrow OH + 2O_2$ | $1.0\times10^{-14}$ exp[-490/T] |
| $2HO_2 \rightarrow H_2O_2 + O_2$ | see JPL-2010 |
| $H_2O_2 + OH \rightarrow H_2O + HO_2$ | $1.8\times10^{-12}$ |
| $OH + OH \rightarrow H_2O + O$ | $1.8\times10^{-12}$ |
| $H_2 + OH \rightarrow H_2O + H$ | $2.8\times10^{-12}$ exp[-1800/T] |
| $N_2O + O(1D) \rightarrow 2NO$ | $7.25\times10^{-11}$ exp[20/T] |
| $N_2O + O(1D) \rightarrow N_2 + O_2$ | $4.63\times10^{-11}$ exp[20/T] |
| $N + NO \rightarrow N_2 + O$ | $2.1\times10^{-11}$ exp[100/T] |
| $N + O_2 \rightarrow NO + O$ | $1.5\times10^{-11}$ exp[-3600/T] |
| $N + O_3 \rightarrow NO + O_2$ | $1.0\times10^{-16}$ |
| $NO + HO_2 \rightarrow NO_2 + OH$ | $3.3\times10^{-12}$ exp[270/T] |
| $NO + O_3 \rightarrow NO_2 + O_2$ | $3.0\times10^{-12}$ exp[-1500/T] |
| $NO_2 + O \rightarrow NO + O_2$ | $5.1\times10^{-12}$ exp[210/T] |
| $NO_2 + O_3 \rightarrow NO_3 + O_2$ | $1.2\times10^{-13}$ exp[-2450/T] |
| $OH + NO_2 + M \rightarrow HNO_3 + M$ | see JPL-2010 |
| $HNO_3 + OH \rightarrow NO_3 + H_2O$ | see JPL-2010 |
| $HO_2 + NO_2 + M \rightarrow HNO_4 + M$ | see JPL-2010 |
| $HO_2 + NO_2 \rightleftarrows HNO_4$ | $2.1\times10^{-27}$ exp[10900/T] (equilibrium constant) |
| $HNO_4 + OH \rightarrow H_2O + NO_2$ | $1.3\times10^{-12}$ exp[380/T] |

| Reaction | Rate constant |
|---|---|
| $NO_3 + NO_2 + M \rightarrow N_2O_5 + M$ | see JPL-2010 |
| $NO_3 + NO_2 \rightarrow N_2O_5$ | $2.7\times10^{-27}$ exp[11000/T] |
| $Cl + O_3 \rightarrow ClO + O_2$ | $2.3\times10^{-11}$ exp[-200/T] |
| $ClO + O \rightarrow Cl + O_2$ | $2.8\times10^{-11}$ exp[85/T] |
| $ClO + NO \rightarrow Cl + NO_2$ | $6.4\times10^{-12}$ exp[290/T] |
| $Cl + CH_4 \rightarrow CH_3 + HCl$ | $7.3\times10^{-12}$ exp[-1280/T] |
| $Cl + HO_2 \rightarrow O_2 + HCl$ | $1.4\times10^{-11}$ exp[270/T] |
| $HCl + OH \rightarrow H_2O + Cl$ | $1.8\times10^{-12}$ exp[-250/T] |
| $HO_2 + ClO \rightarrow HOCl + O_2$ | $2.6\times10^{-12}$ exp[290/T] |
| $HOCl + OH \rightarrow H_2O + ClO$ | $3.0\times10^{-12}$ exp[-500/T] |
| $ClO + NO_2 + M \rightarrow ClONO_2 + M$ | see JPL-2010 |
| $ClONO_2 + O \rightarrow$ products | $3.6\times10^{-12}$ exp[-840/T] |
| $ClO + ClO + M \rightarrow Cl_2O_2 + M$ | see JPL-2010 |
| $ClO + ClO \rightleftarrows Cl_2O_2$ | $1.72\times10^{-27}$ exp[8649/T] (equilibrium constant) |
| $Cl + O_2 + M \rightarrow ClOO + M$ | see JPL-2010 |
| $Cl + O_2 \rightleftarrows ClOO$ | $6.6\times10^{-25}$ exp[2502/T] (equilibrium constant) |
| $ClO + O_3 \rightarrow OClO + O_2$ | $1.0\times10^{-12}$ exp[-4117/T] |
| $OClO + Cl \rightarrow ClO + ClO$ | $3.4\times10^{-11}$ exp[160/T] |
| $OClO + OH \rightarrow HOCl + O_2$ | $1.4\times10^{-12}$ exp[600/T] |
| $OClO + NO \rightarrow ClO + NO_2$ | $2.5\times10^{-12}$ exp[-600/T] |
| $OClO + O \rightarrow ClO + O_2$ | $2.4\times10^{-12}$ exp[-960/T] |
| $Cl + H_2 \rightarrow H + HCl$ | $3.05\times10^{-11}$ exp[-2270/T] |
| $Cl + H_2O_2 \rightarrow HCl + HO_2$ | $1.1\times10^{-11}$ exp[-980/T] |
| $Cl + CH_2O \rightarrow CHO + HCl$ | $8.1\times10^{-11}$ exp[-30/T] |
| $HCl + O \rightarrow OH + Cl$ | $1.0\times10^{-11}$ exp[-3300/T] |
| $Cl + HO_2 \rightarrow Cl_2 + OH$ | $3.6\times10^{-11}$ exp[-375/T] |
| $Cl + HOCl \rightarrow$ products ($Cl_2 + OH$, $HCl + ClO$) | $3.4\times10^{-12}$ exp[-130/T] |
| $HOCl + O \rightarrow OH + ClO$ | $1.7\times10^{-13}$ |
| $ClO + ClO \rightarrow Cl_2 + O_2$ | $1.0\times10^{-12}$ exp[-1590/T] |
| $ClO + ClO \rightarrow OClO + Cl$ | $3.5\times10^{-13}$ exp[-1370/T] |
| $ClO + ClO \rightarrow ClOO + Cl$ | $3.0\times10^{-11}$ exp[-2450/T] |
| $ClO + OH \rightarrow Cl + HO_2$ | $7.4\times10^{-12}$ exp[270/T] |
| $ClO + OH \rightarrow HCl + O_2$ | $6.0\times10^{-13}$ exp[230/T] |
| $CH_2O + Cl \rightarrow HCl + CHO$ | $8.1\times10^{-11}$ exp[-30/T] |
| $CH_3O_2 + ClO \rightarrow$ products ($ClOO + CH_3O$, $CH_3OCl + O_2$) | $3.3\times10^{-12}$ exp[-115/T] |
| $Cl + NO_2 + M \rightarrow ClNO2 + M$ | see JPL-2010 |
| $OH + ClNO_2 \rightarrow HOCl + NO_2$ | $2.4\times10^{-12}$ exp[-1250/T] |
| $CH_3CCl_3 + OH \rightarrow 3Cl +$ products | $1.64\times10^{-12}$ exp[-1520/T] |
| $CH_3CCl_3 + Cl \rightarrow CH_2CCl_3 + HCl$ | $3.23\times10^{-12}$ exp[-1770/T] |
| $CH_3Cl + OH \rightarrow Cl +$ products | $2.4\times10^{-12}$ exp[-1250/T] |
| $CH_3Cl + Cl \rightarrow CH_2Cl + HCl$ | $2.17\times10^{-11}$ exp[-1130/T] |
| $CCl_4 + O(1D) \rightarrow$ products | $3.3\times10^{-10}$ |
| $CFCl_3 + O(1D) \rightarrow$ products | $2.3\times10^{-10}$ |
| $CF_2Cl_2 + O(1D) \rightarrow$ products | $1.4\times10^{-10}$ |
| $CClF_2CCl_2F + O(1D) \rightarrow$ products | $2.0\times10^{-10}$ |
| $CHClF_2 + O(1D) \rightarrow$ products | $1.0\times10^{-10}$ |
| $Br + O_3 \rightarrow BrO + O_2$ | $1.6\times10^{-11}$ exp[-780/T] |

| | |
|---|---|
| $Br + HO_2 \rightarrow HBr + O_2$ | $4.8 \times 10^{-12} \exp[-310/T]$ |
| $Br + H_2O_2 \rightarrow HBr + HO_2$ | $1.0 \times 10^{-11} \exp[-3000/T]$ |
| $Br + OClO \rightarrow BrO + ClO$ | $2.6 \times 10^{-11} \exp[-1300/T]$ |
| $Br + CH_2O \rightarrow HBr + CHO$ | $1.7 \times 10^{-11} \exp[-800/T]$ |
| $BrO + O \rightarrow Br + O_2$ | $1.9 \times 10^{-11} \exp[230/T]$ |
| $BrO + OH \rightarrow Br + HO_2$ | $1.666 \times 10^{-11} \exp[250/T]$ |
| $BrO + OH \rightarrow HBr + O_2$ | $3.4 \times 10^{-13} \exp[250/T]$ |
| $BrO + HO_2 \rightarrow HOBr + O_2$ | $4.5 \times 10^{-12} \exp[460/T]$ |
| $BrO + NO \rightarrow Br + NO_2$ | $8.8 \times 10^{-12} \exp[260/T]$ |
| $BrO + NO_2 + M \rightarrow BrONO_2 + M$ | see JPL-2010 |
| $BrO + ClO \rightarrow Br + OClO$ | $9.5 \times 10^{-13} \exp[550/T]$ |
| $BrO + ClO \rightarrow Br + ClOO$ | $2.3 \times 10^{-12} \exp[260/T]$ |
| $BrO + ClO \rightarrow BrCl + O_2$ | $4.1 \times 10^{-13} \exp[290/T]$ |
| $HBr + O(1D) \rightarrow$ products (Br + OH, HBr + O, H + BrO) | $1.5 \times 10^{-10}$ |
| $HBr + O \rightarrow OH + Br$ | $5.8 \times 10^{-12} \exp[-1500/T]$ |
| $HBr + OH \rightarrow Br + H_2O$ | $5.5 \times 10^{-12} \exp[200/T]$ |
| $BrO + BrO \rightarrow 2Br + O_2$ | $2.4 \times 10^{-12} \exp[40/T]$ |
| $BrO + BrO \rightarrow Br_2 + O_2$ | $2.8 \times 10^{-14} \exp[860/T]$ |
| $BrO + O_3 \rightarrow Br + 2O_2$ | $1.0 \times 10^{-12} \exp[-3225/T]$ |
| $BrONO_2 + O \rightarrow BrO + NO_3$ | $1.9 \times 10^{-11} \exp[215/T]$ |
| $CHBr_3 + O(1D) \rightarrow$ products | $6.6 \times 10^{-10}$ |
| $CHBr_3 + OH \rightarrow 3Br + $ products | $1.35 \times 10^{-12} \exp[-600/T]$ |
| $CHBr_3 + Cl \rightarrow CBr_3 + HCl$ | $4.85 \times 10^{-12} \exp[-850/T]$ |
| $CH_3Br + O(1D) \rightarrow$ products | $1.8 \times 10^{-10}$ |
| $CH_3Br + OH \rightarrow Br + $ products | $2.35 \times 10^{-12} \exp[-1300/T]$ |
| $CH_3Br + Cl \rightarrow CH_2Br + HCl$ | $1.4 \times 10^{-11} \exp[-1030/T]$ |
| $CH_2Br_2 + O(1D) \rightarrow Br + $ products | $2.7 \times 10^{-10}$ |
| $CH_2Br_2 + OH \rightarrow CHBr_2 + H_2O$ | $2.0 \times 10^{-12} \exp[-840/T]$ |
| $CH_2Br_2 + Cl \rightarrow CHBr_2 + HCl$ | $6.3 \times 10^{-12} \exp[-800/T]$ |
| $CF_2ClBr + O(1D) \rightarrow$ products | $1.5 \times 10^{-10}$ |
| $CF_3Br + O(1D) \rightarrow$ products | $1.0 \times 10^{-10}$ |

| (3) Heterogeneous reactions | Reaction coefficients (reference) |
|---|---|
| $N_2O_5 + H_2O \rightarrow 2HNO_3$ | see JPL-2010 and Akiyoshi (2007) |
| $ClONO_2 + H_2O \rightarrow HOCl + HNO_3$ | see JPL-2010 and Akiyoshi (2007) |
| $ClONO_2 + HCl \rightarrow Cl_2 + HNO_3$ | see JPL-2010 and Akiyoshi (2007) |
| $HOCl + HCl \rightarrow Cl_2 + H_2O$ | see JPL-2010 and Akiyoshi (2007) |
| $BrONO_2 + H_2O \rightarrow HOBr + HNO_3$ | see JPL-2010 and Akiyoshi (2007) |
| $HOBr + HCl \rightarrow BrCl + H_2O$ | see JPL-2010 and Akiyoshi (2007) |
| $BrONO_2 + HCl \rightarrow BrCl + HNO_3$ | see JPL-2010 and Akiyoshi (2007) |
| $ClONO_2 + HBr \rightarrow BrCl + HNO_3$ | see JPL-2010 and Akiyoshi (2007) |
| $BrONO_2 + HBr \rightarrow Br_2 + HNO_3$ | see JPL-2010 and Akiyoshi (2007) |
| $HOCl + HBr \rightarrow BrCl + H_2O$ | see JPL-2010 and Akiyoshi (2007) |
| $HOBr + HBr \rightarrow Br_2 + H_2O$ | see JPL-2010 and Akiyoshi (2007) |
| $N_2O_5 + HBr \rightarrow BrNO_2 + HNO_3$ | see JPL-2010 and Akiyoshi (2007) |
| $N_2O_5 + HCl \rightarrow ClNO_2 + HNO_3$ | see JPL-2010 and Akiyoshi (2007) |

Total:

42 photolysis reactions
140 gas phase reactions
13 heterogeneous reactions

Figures

Figure 1.  Averaging kernel functions of the SFIT2 retrievals for (a) $O_3$, (b) $HNO_3$, and (c) HCl.

Figure 2.  (a) Mean absolute and (b) mean relative differences of $O_3$ profiles retrieved from FTIR measurements minus those from ozonesonde measurements.  (c) Mean absolute and (d) mean relative differences of $O_3$ profiles retrieved from FTIR measurements minus those from Aura/MLS measurements.  (e) Mean absolute and (f) mean relative differences of $HNO_3$ profiles retrieved from FTIR measurements minus those from Aura/MLS measurements.  (g) Mean absolute and (h) mean relative differences of HCl profiles retrieved from FTIR measurements minus those from Aura/MLS measurements.  Horizontal bars indicate the root mean squares of differences at each altitude.  Horizontal dashed bars indicate the altitude range of our focus (15-25 km).

Figure 3.  Time series of (a) daytime hour, temperatures at 18 km in (b) 2007 and (c) 2011, and at 22 km in (d) 2007 and (e) 2011 over Syowa Station using ERA-Interim data.  Approximate saturation temperatures for nitric acid trihydrate ($T_{NAT}$) and ice ($T_{ICE}$) calculated by assuming 6 ppbv $HNO_3$ and 4.5 ppmv $H_2O$ are also plotted in the figures by dotted lines.  Dates when PSCs were observed over Syowa Station are indicated by asterisks on the bottom of the figures.

Figure 4.  Time series of (a) HCl, $ClONO_2$, ClO, $Cl_y$*, (b) $O_3$, and $HNO_3$ mixing ratios at 18 km in 2007 over Syowa Station. $O_3$(FTIR), HCl(FTIR), and $HNO_3$(FTIR) were measured by FTIR at Syowa Station, while HCl(MLS), ClO(MLS), and $HNO_3$(MLS) were measured by Aura/MLS.  $O_3$(sonde) was measured by ozonesonde.  $ClONO_2$ was measured by Envisat/MIPAS.  $Cl_y$* is calculated from Aura/MLS $N_2O$ value.  See text in detail.  The unit of $O_3$ is ppmv and the other gases are ppbv.  The dark shaded area, the light shaded area, and the white area indicate the days when Syowa Station was located outside, in the boundary region, and inside the polar vortex, respectively.

Figure 5.  Same as Figure 4 but in 2011.

Figure 6.  Same as Figure 4 but at 22 km.

Figure 7.  Same as Figure 5 but at 22 km.

Figure 8.  Time series of the ratios of HCl (dark blue or light blue), $ClONO_2$ (yellow), and ClO (red) to total chlorine ($Cl_y$*) over Syowa Station at 18 km in (a) 2007 and in (b) 2011.  Light blue shows either $ClONO_2$ or ClO data was missing on that day, while dark blue shows all three data were available on that day.  Shaded areas are the same as Figure 4.

Figure 9. Same as Figure 8 but at 22 km.

Figure 10. Scatter plot between ClO (Aura/MLS) and ClONO$_2$ (Envisat/MIPAS) mixing ratios between August 8 and September 17 (day 220 – 260) at 18 km and 22 km in 2007 and 2011. Crosses, triangles, and squares represent the data when Syowa Station was located inside the polar vortex, the boundary region, and outside the polar vortex, respectively. Solid lines are regression lines obtained by RMA regression. Color represents the equivalent latitude over Syowa Station on that day. Circles with crosses represent the days which are shown in Figures 13 and 14.

Figure 11. Daily time series of measured and modeled minor species over Syowa Station at 18 km. Black diamonds are data by FTIR, red squares are by Aura/MLS and Envisat/MIPAS, blue triangles are data by MIROC3.2 CCM. (a) is for ClO, (b) is for HCl, (c) is for ClONO$_2$, (d) is for Cly, and (e) is for O$_3$ in 2007. (f) is for ClO, (g) is for HCl, (h) is for ClONO$_2$, (i) is for Cly, and (j) is for O$_3$ in 2011.

Figure 12. Same as Figure 11 but for 22 km.

Figure 13. Polar southern hemispheric plots for ERA-Interim temperature, simulated mixing ratios of O$_3$, NO$_2$, HNO$_3$, ClO, HCl, and ClONO$_2$ by a MIROC3.2 chemistry-climate model (CCM) at 50 hPa for June 24 (day 175), September 2 (day 245), September 6 (day 249), and October 6 (day 279), 2007. Polar vortex (outer) edge defined as the method described in Appendix B at 450 K was plotted by white circle in each panel. The location of Syowa Station was shown by white star in each panel.

Figure 14. Same as Figure 13 but for July 5 (day 186), August 19 (day 231), August 21 (day 233), and October 9 (day 282), 2011. Polar vortex edge on July 5 plotted by dotted while circle indicates that the inner vortex edge was defined on this day.

Figure 15. Three-hourly zonal-mean time series of MIROC3.2 CCM outputs for (a) ClO+2*Cl$_2$O$_2$+2*Cl$_2$, (b) Cl$_2$O$_2$, (c) Cl$_2$, (d) ClO, (e) HOCl, (f) HCl, and (g) ClONO$_2$ during day number 120–300 at 50 hPa in 2007.

Figure A1. Time-equivalent latitude sections of MPV (contours) and its gradient with respect to EL (colors) at (a) 450 K and (b) 560 K isentropic PT surfaces in 2007. Black dots represent the inner and outer edge(s) of the polar vortex. Red dots represent the EL of Syowa Station on each day.

Figure A2. Same as Figure A1 but for the year in 2011.

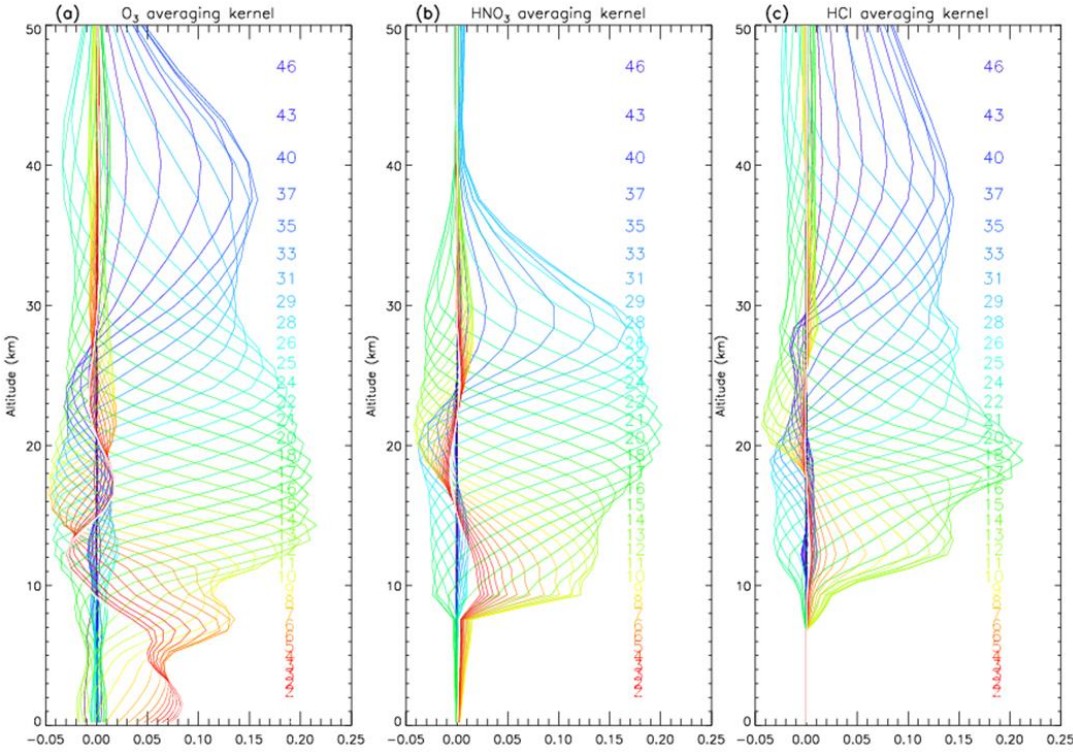

Figure 1. Averaging kernel functions of the SFIT2 retrievals for (a) O₃, (b) HNO₃, and (c) HCl.

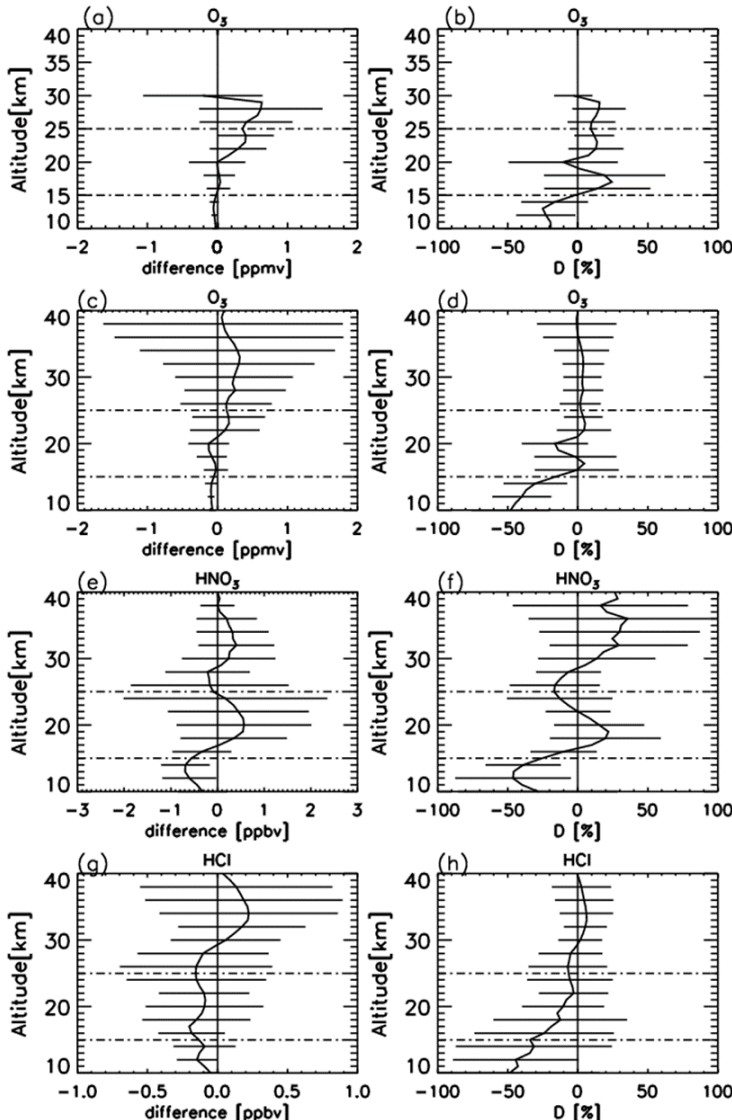

Figure 2. (a) Mean absolute and (b) mean relative differences of $O_3$ profiles retrieved from FTIR measurements minus those from ozonesonde measurements. (c) Mean absolute and (d) mean relative differences of $O_3$ profiles retrieved from FTIR measurements minus those from Aura/MLS measurements. (e) Mean absolute and (f) mean relative differences of $HNO_3$ profiles retrieved from FTIR measurements minus those from Aura/MLS measurements. (g) Mean absolute and (h) mean relative differences of HCl profiles retrieved from FTIR measurements minus those from Aura/MLS measurements. Horizontal bars indicate the root mean squares of differences at each altitude. Horizontal dashed bars indicate the altitude range of our focus (15-25 km).

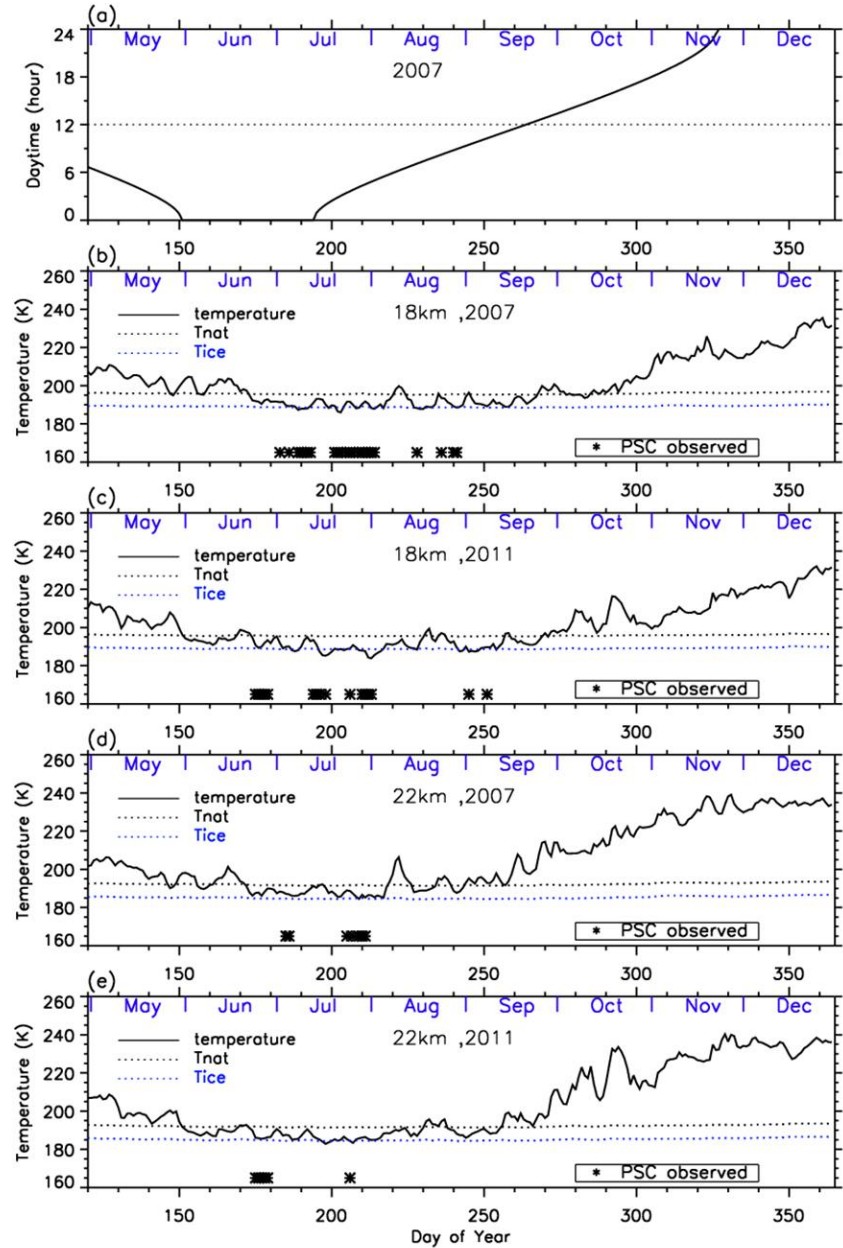

Figure 3. Time series of (a) daytime hour, temperatures at 18 km in (b) 2007 and (c) 2011, and at 22 km in (d) 2007 and (e) 2011 over Syowa Station using ERA-Interim data. Approximate saturation temperatures for nitric acid trihydrate ($T_{NAT}$) and ice ($T_{ICE}$) calculated by assuming 6 ppbv $HNO_3$ and 4.5 ppmv $H_2O$ are also plotted in the figures by dotted lines. Dates when PSCs were observed over Syowa Station are indicated by asterisks on the bottom of the figures.

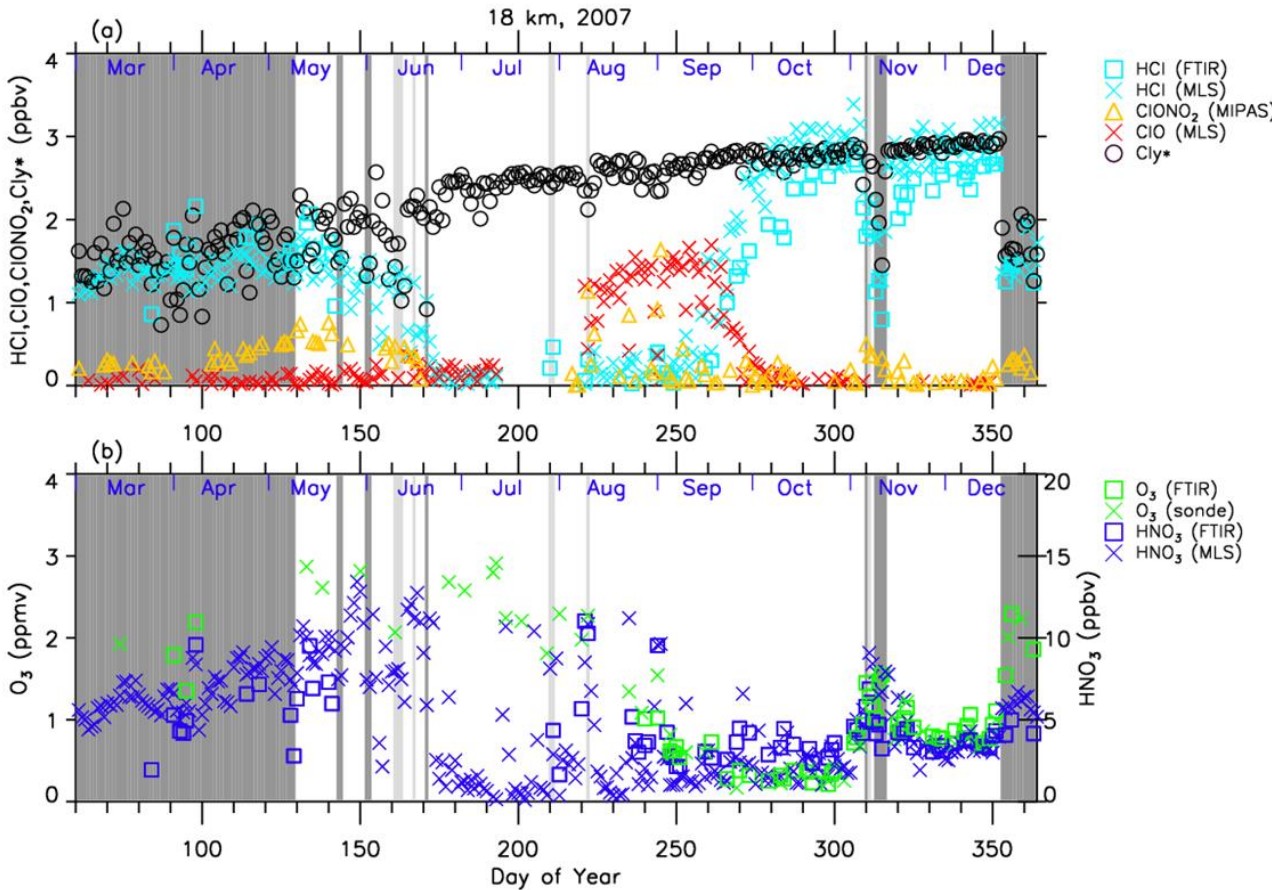

Figure 4. Time series of (a) HCl, $ClONO_2$, ClO, $Cl_y$*, (b) $O_3$, and $HNO_3$ mixing ratios at 18 km in 2007 over Syowa Station. $O_3$(FTIR), HCl(FTIR), and $HNO_3$(FTIR) were measured by FTIR at Syowa Station, while HCl(MLS), ClO(MLS), and $HNO_3$(MLS) were measured by Aura/MLS. $O_3$(sonde) was measured by ozonesonde. $ClONO_2$ was measured by Envisat/MIPAS. $Cl_y$* is calculated from Aura/MLS $N_2O$ value. See text in detail. The unit of $O_3$ is ppmv and the other gases are ppbv. The dark shaded area, the light shaded area, and the white area indicate the days when Syowa Station was located outside, in the boundary region, and inside the polar vortex, respectively.

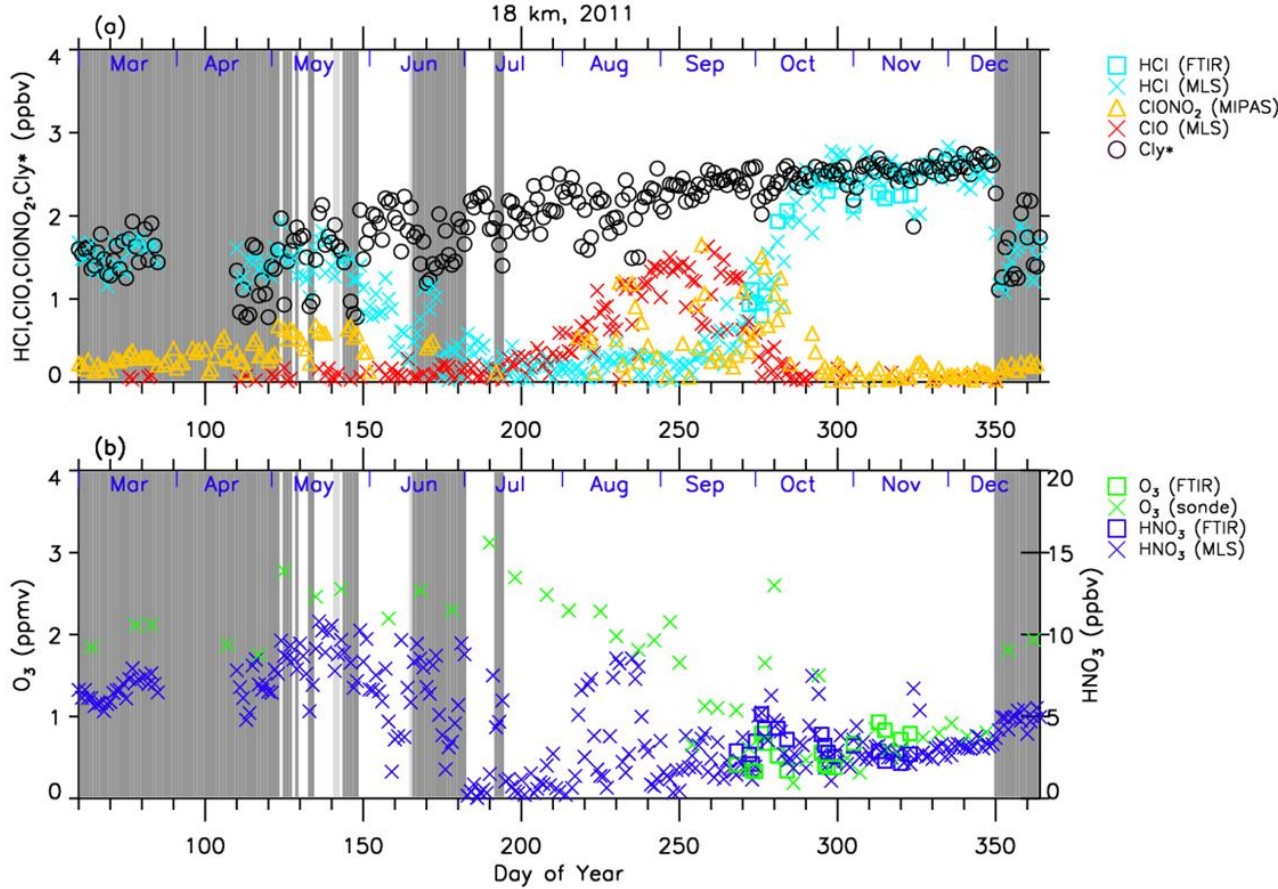

Figure 5.  Same as Figure 4 but in 2011.

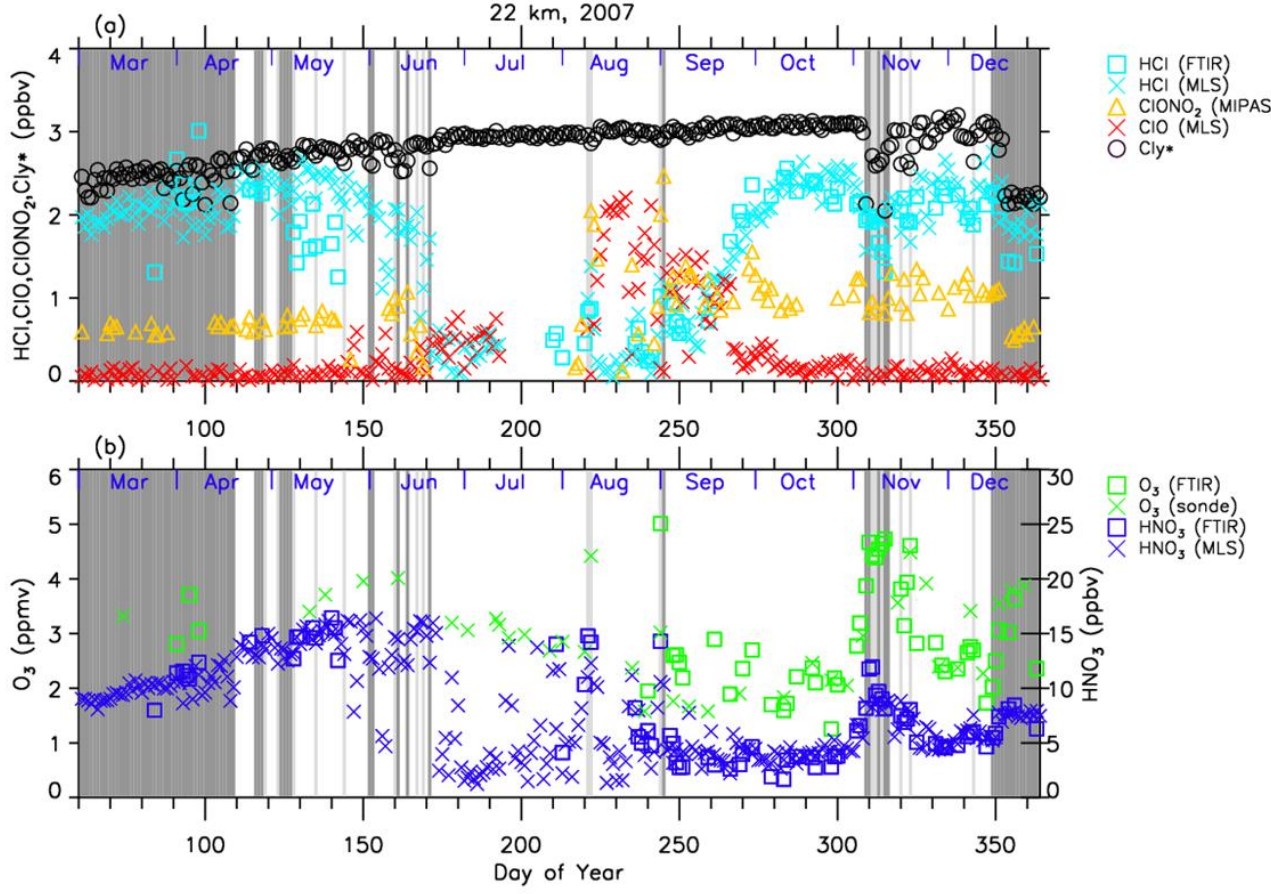

Figure 6. Same as Figure 4 but at 22 km.

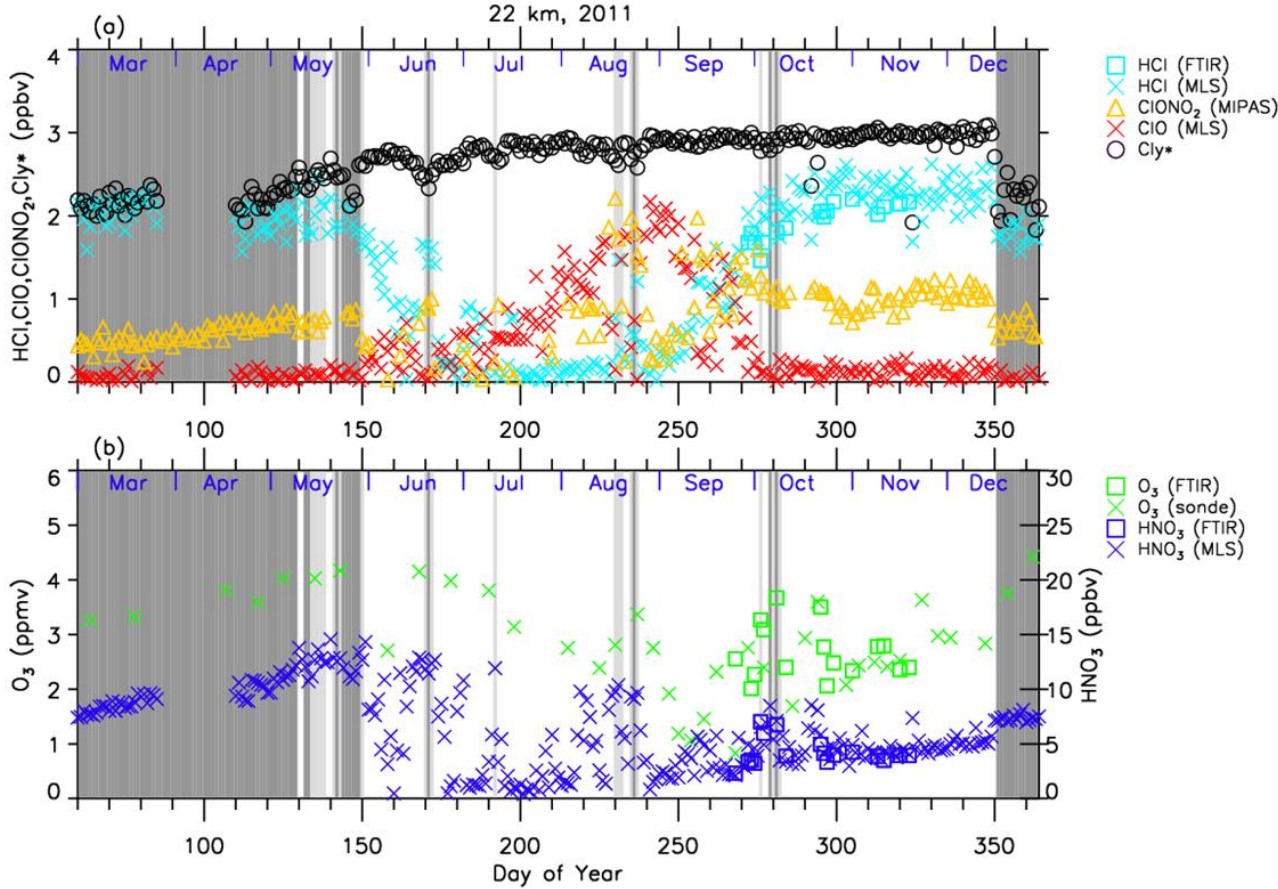

Figure 7. Same as Figure 5 but at 22 km.

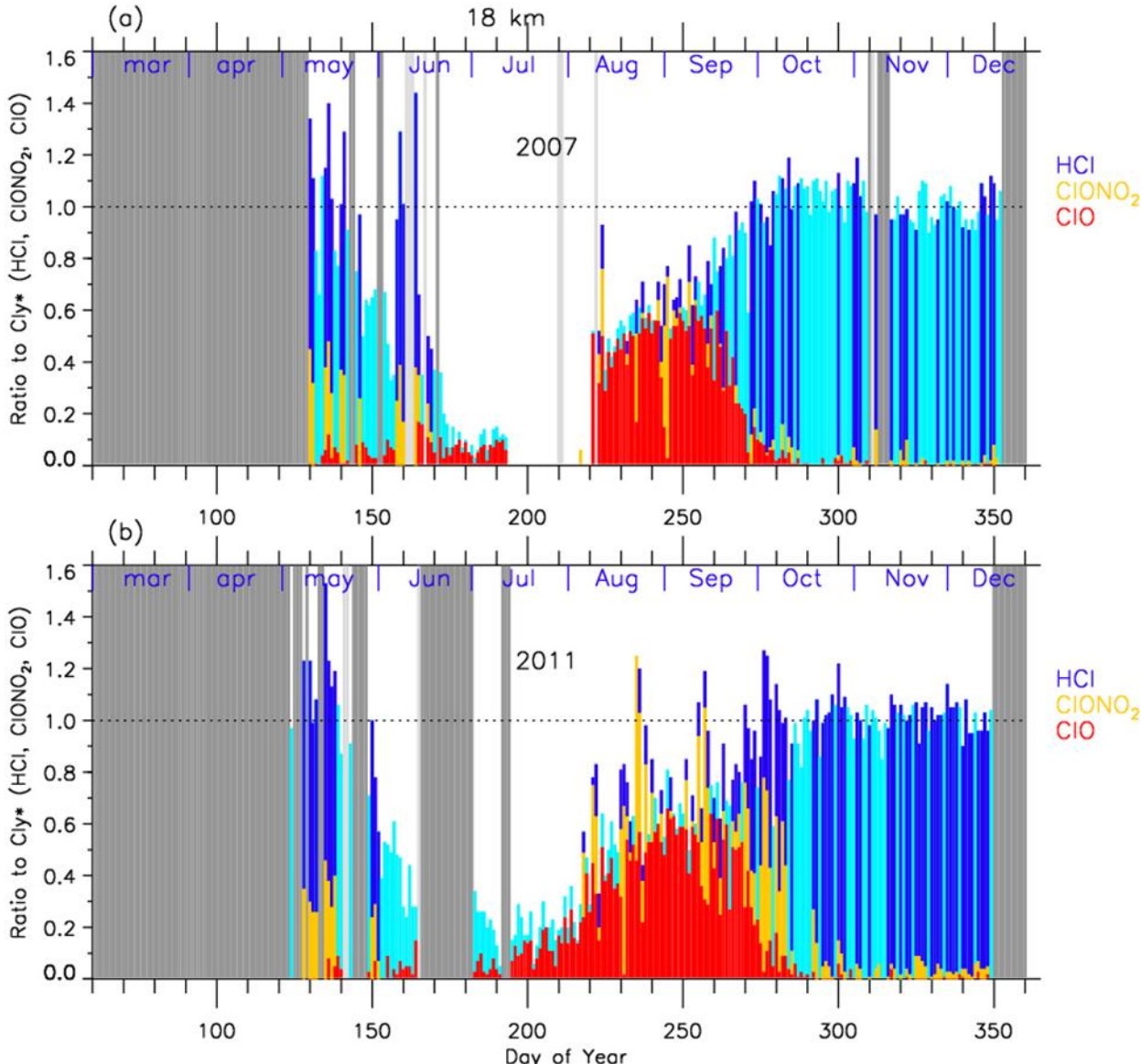

Figure 8. Time series of the ratios of HCl (dark blue or light blue), ClONO₂ (yellow), and ClO (red) to total chlorine (Cl_y*) over Syowa Station at 18 km in (a) 2007 and in (b) 2011. Light blue shows either ClONO₂ or ClO data was missing on that day, while dark blue shows all three data were available on that day. Shaded areas are the same as Figure 4.

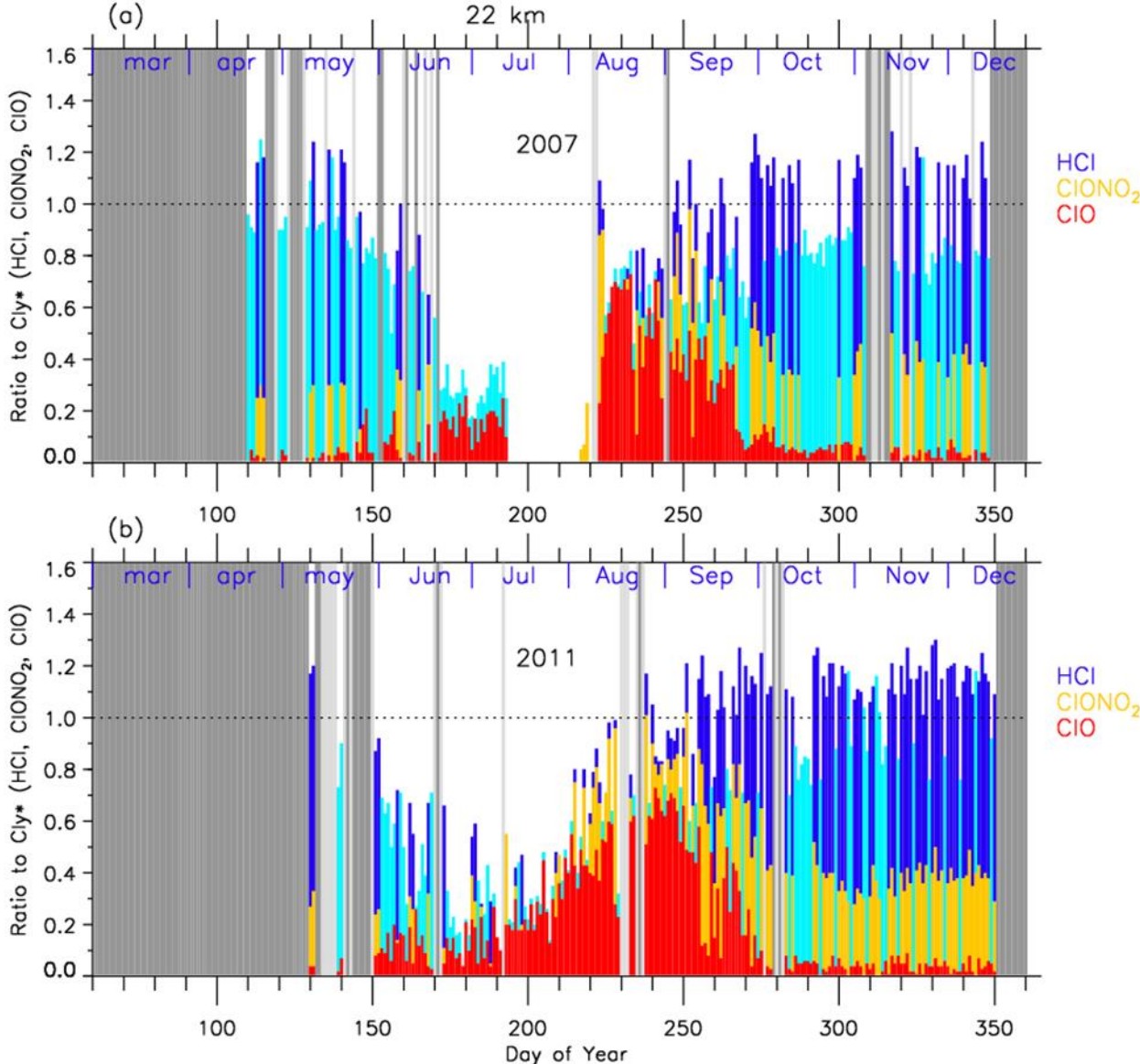

30 Figure 9. Same as Figure 8 but at 22 km.

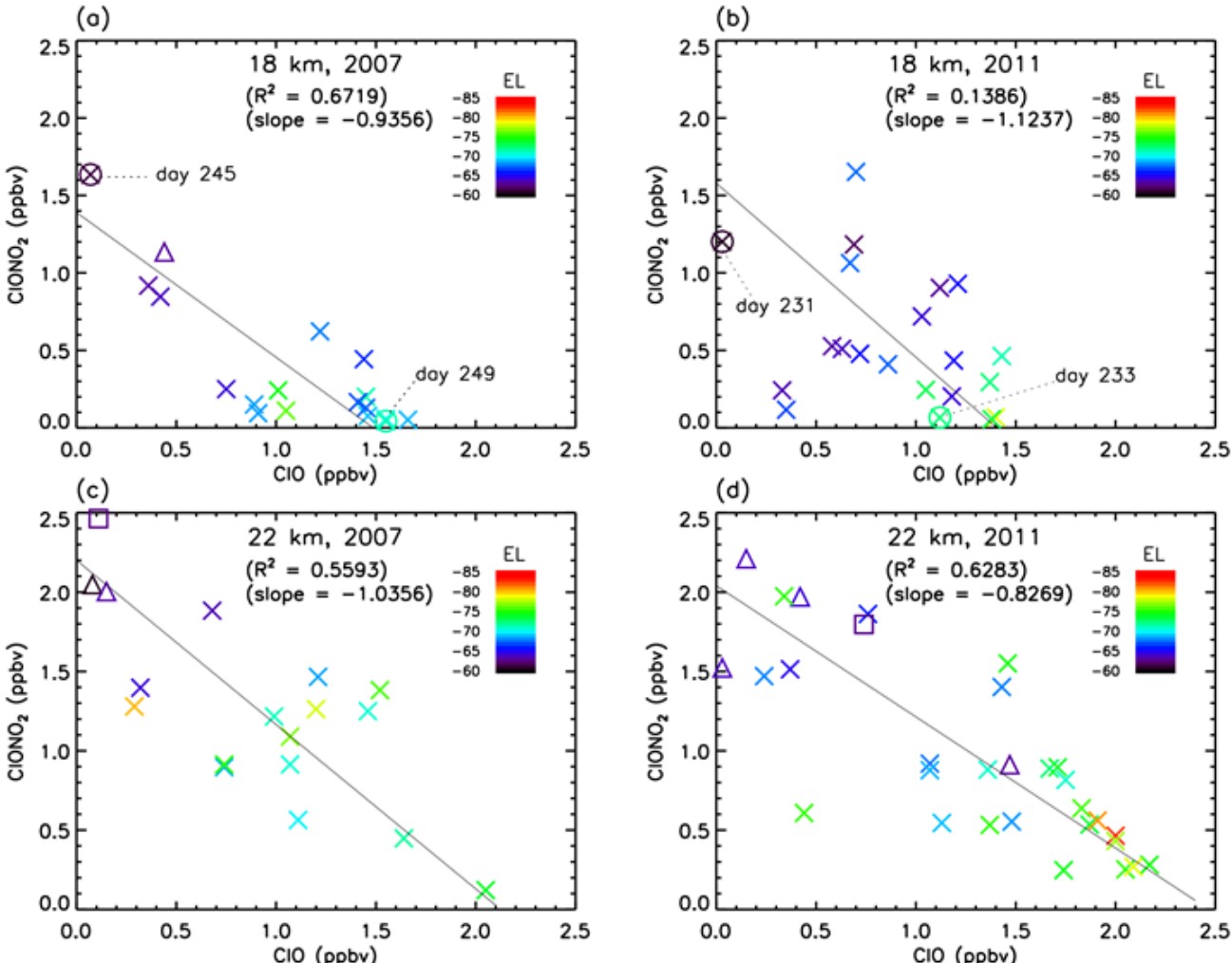

Figure 10.  Scatter plot between ClO (Aura/MLS) and ClONO$_2$ (Envisat/MIPAS) mixing ratios between August 8 and September 17 (day 220 – 260) at 18 km and 22 km in 2007 and 2011.  Crosses, triangles, and squares represent the data when Syowa Station was located inside the polar vortex, the boundary region, and outside the polar vortex, respectively.  Solid lines are regression lines obtained by RMA regression.  Color represents the equivalent latitude over Syowa Station on that day.

30   Circles with crosses represent the days which are shown in Figures 13 and 14.

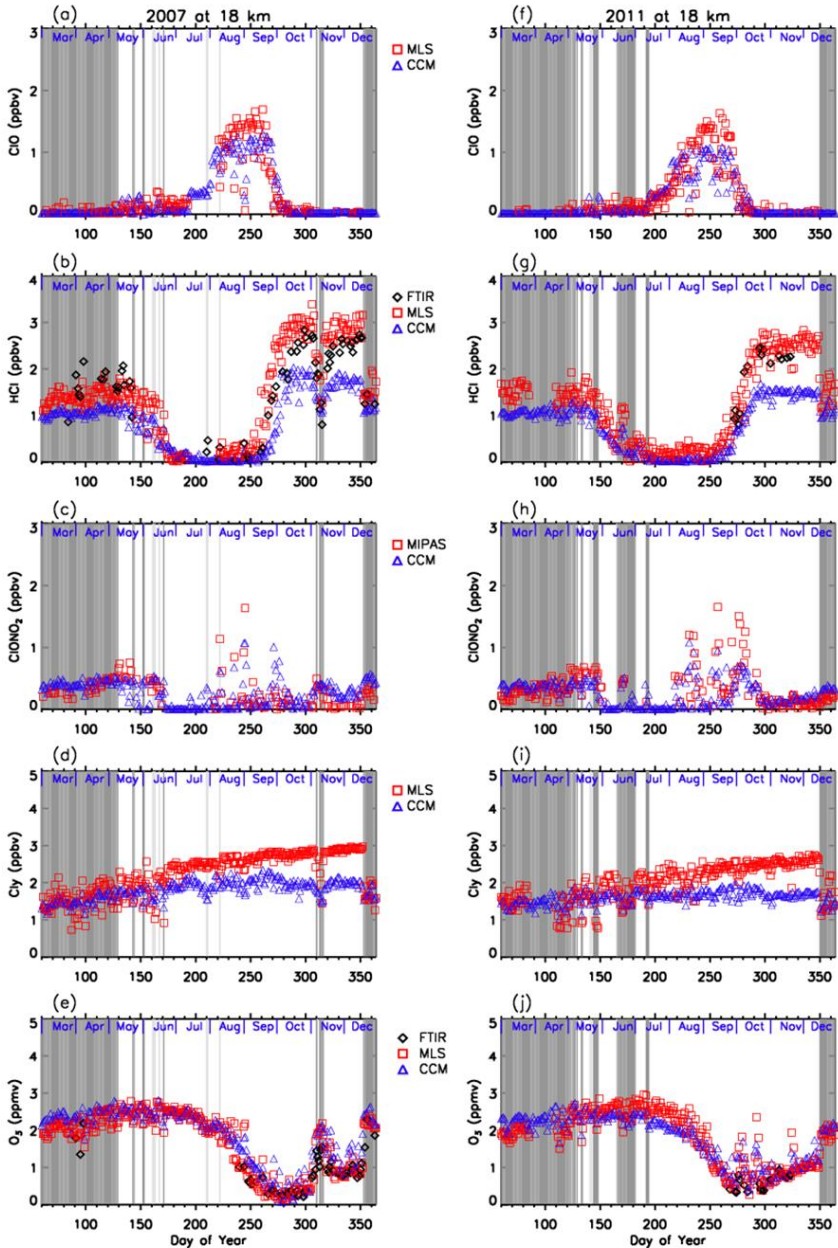

Figure 11. Daily time series of measured and modeled minor species over Syowa Station at 18 km. Black diamonds are data by FTIR, red squares are by Aura/MLS and Envisat/MIPAS, blue triangles are data by MIROC3.2 CCM. (a) is for ClO, (b) is for HCl, (c) is for ClONO$_2$, (d) is for Cly, and (e) is for O$_3$ in 2007. (f) is for ClO, (g) is for HCl, (h) is for ClONO$_2$, (i) is for Cly, and (j) is for O$_3$ in 2011.

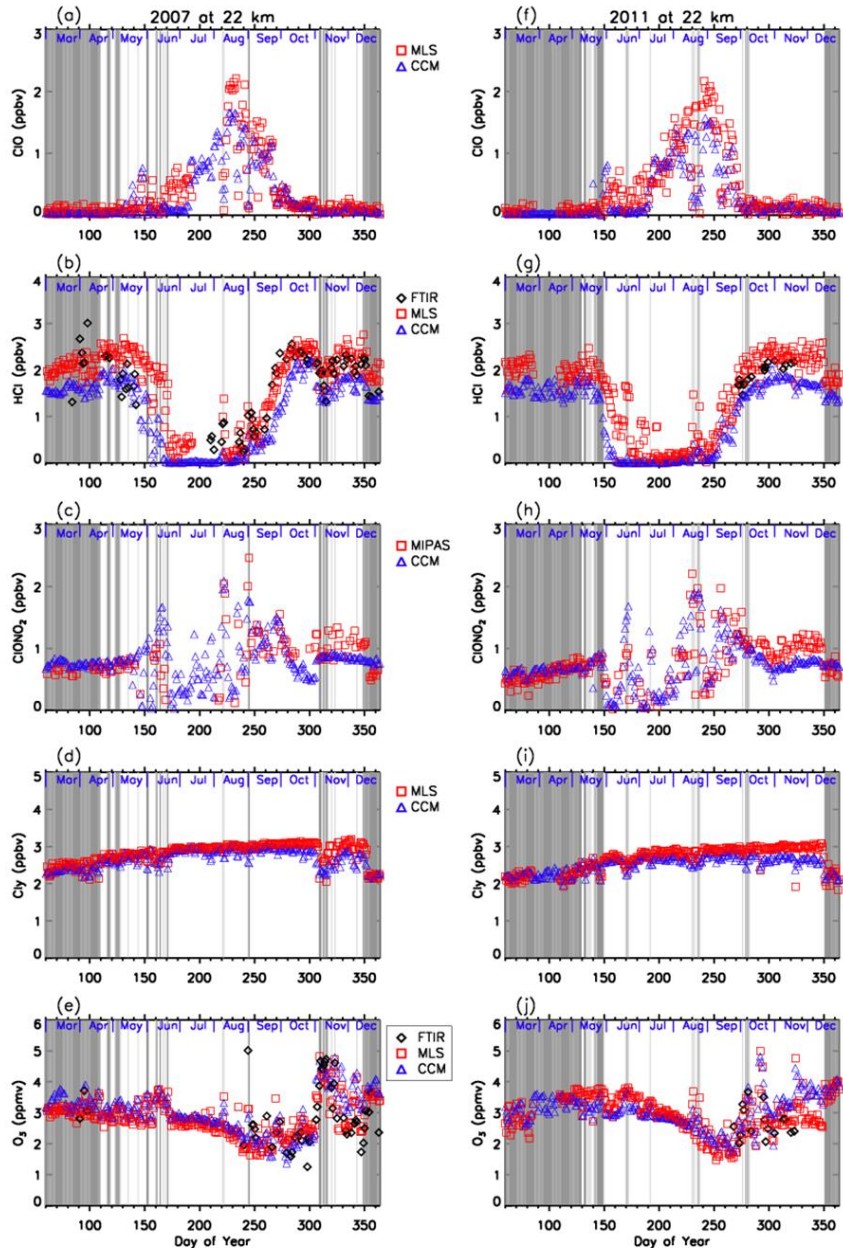

Figure 12. Same as Figure 11 but for 22 km.

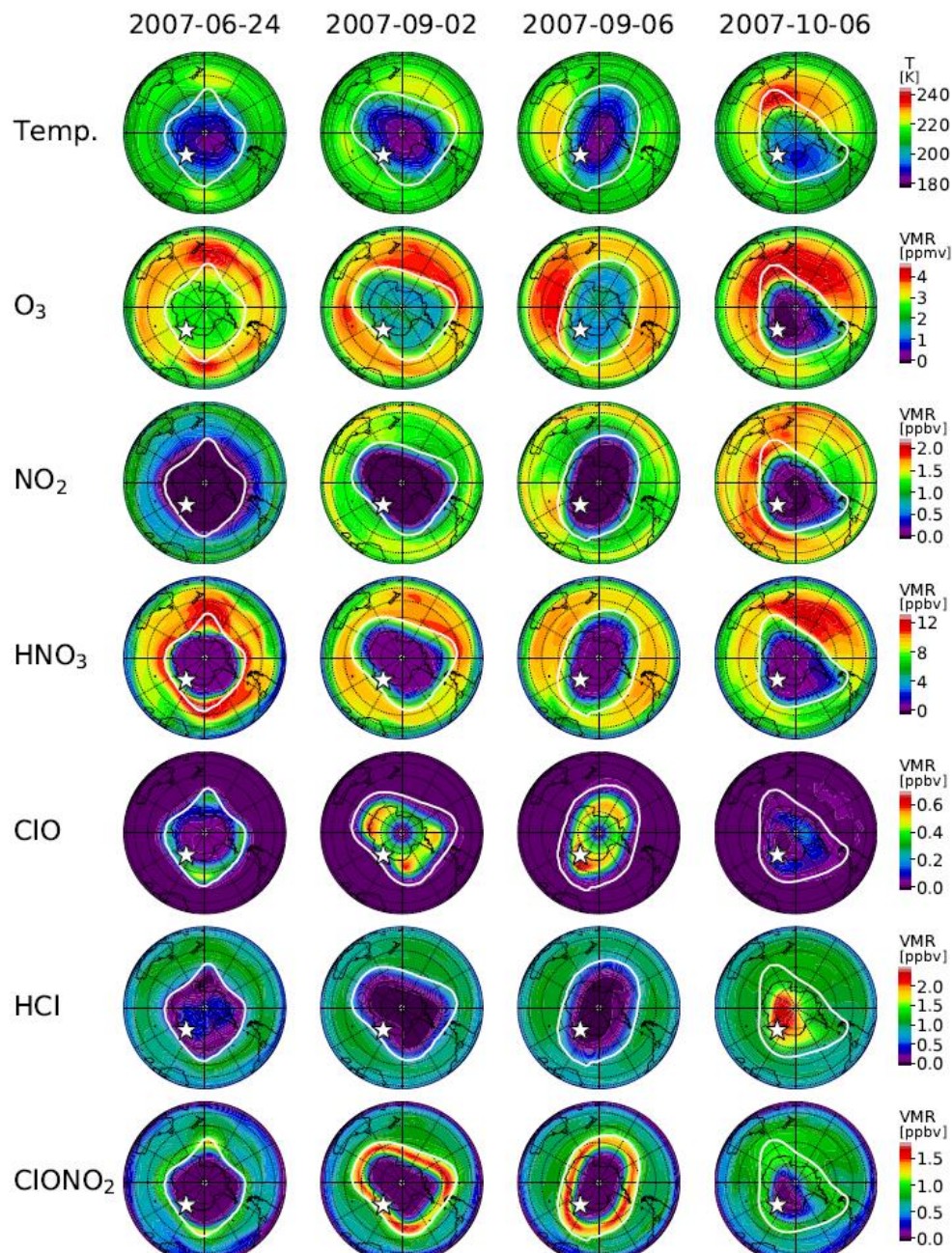

Figure 13. Polar southern hemispheric plots for temperature from the model nudged toward the ERA-Interim data, simulated mixing ratios of $O_3$, $NO_2$, $HNO_3$, ClO, HCl, and $ClONO_2$ by a MIROC3.2 chemistry-climate model (CCM) at 50 hPa for June 24 (day 175), September 2 (day 245), September 6 (day 249), and October 6 (day 279), 2007. Polar vortex (outer) edge defined by the method described in Appendix B at 450 K was plotted by white circle in each panel. The location of Syowa Station was shown by white star in each panel.

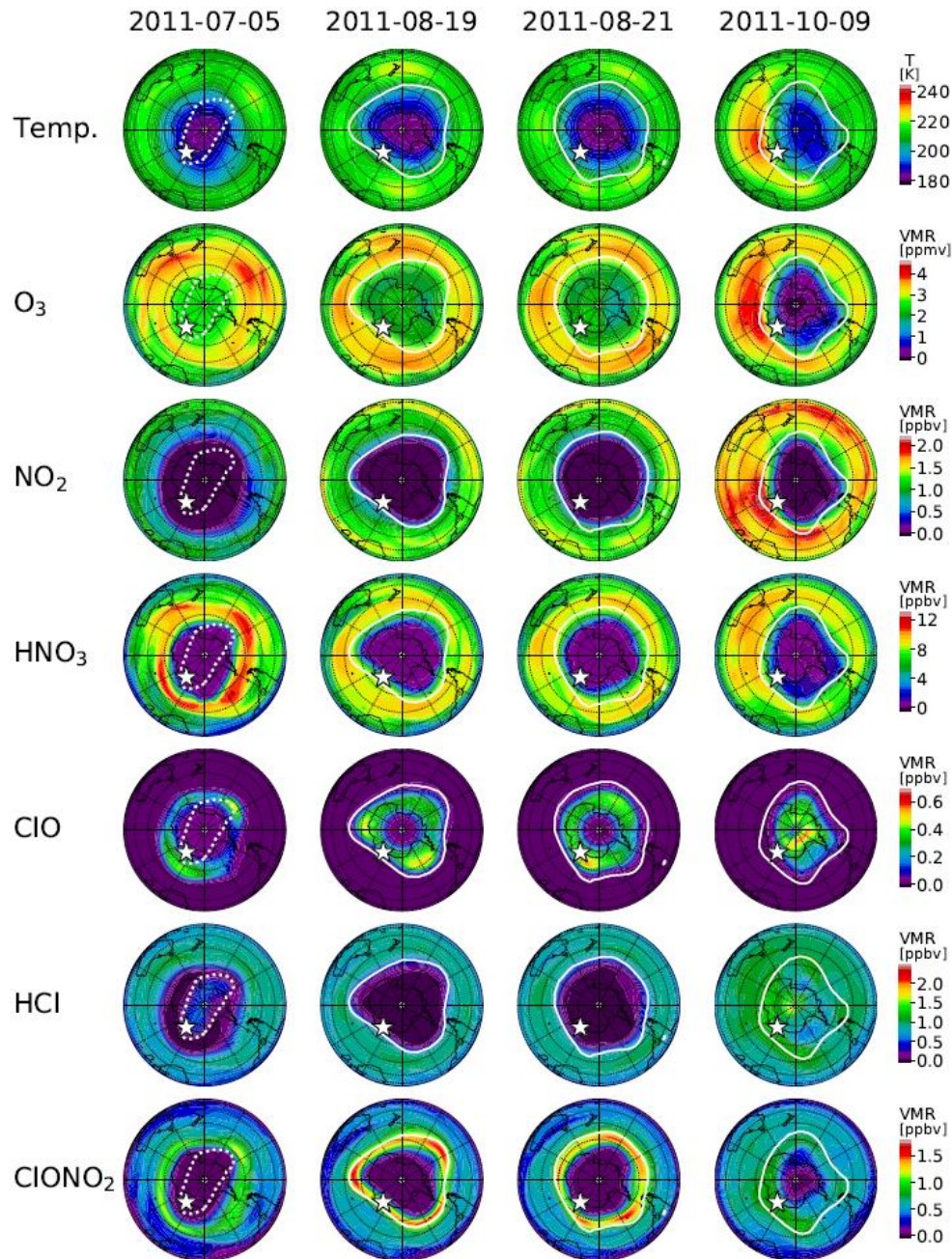

Figure 14. Same as Figure 13 but for July 5 (day 186), August 19 (day 231), August 21 (day 233), and October 9 (day 282), 2011. Polar vortex edge on July 5 plotted by dotted while circle indicates that the inner vortex edge was defined on this day.

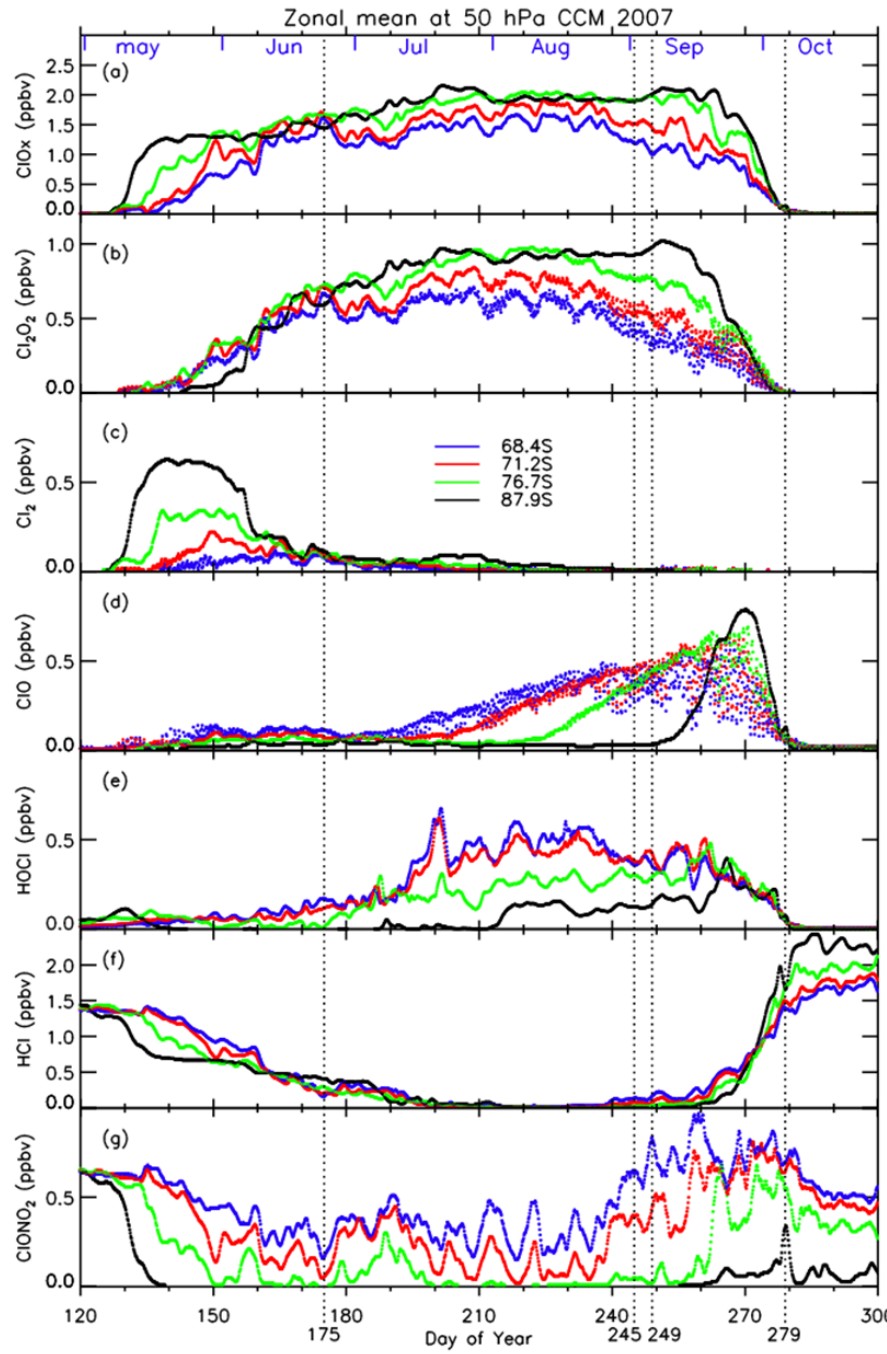

Figure 15. Three-hourly zonal-mean time series of MIROC3.2 CCM outputs for (a) $ClO+2*Cl_2O_2+2*Cl_2$, (b) $Cl_2O_2$, (c) $Cl_2$, (d) ClO, (e) HOCl, (f) HCl, and (g) $ClONO_2$ during day number 120–300 at 50 hPa in 2007.

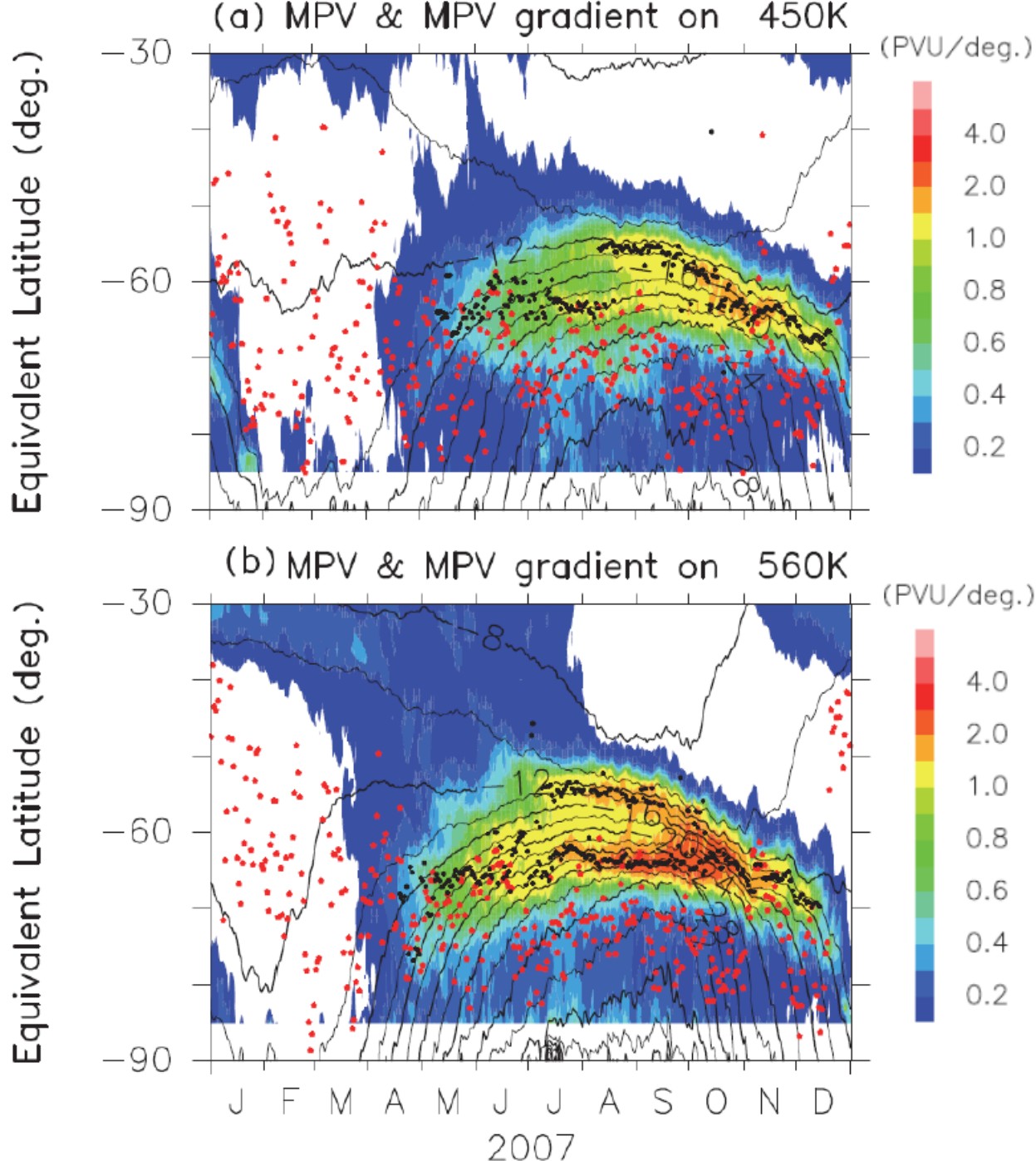

Figure A1.  Time-equivalent latitude sections of MPV (contours) and its gradient with respect to EL (colors) at (a) 450 K and (b) 560 K isentropic PT surfaces in 2007.  Black dots represent the inner and outer edge(s) of the polar vortex.  Red dots represent the EL of Syowa Station on each day.

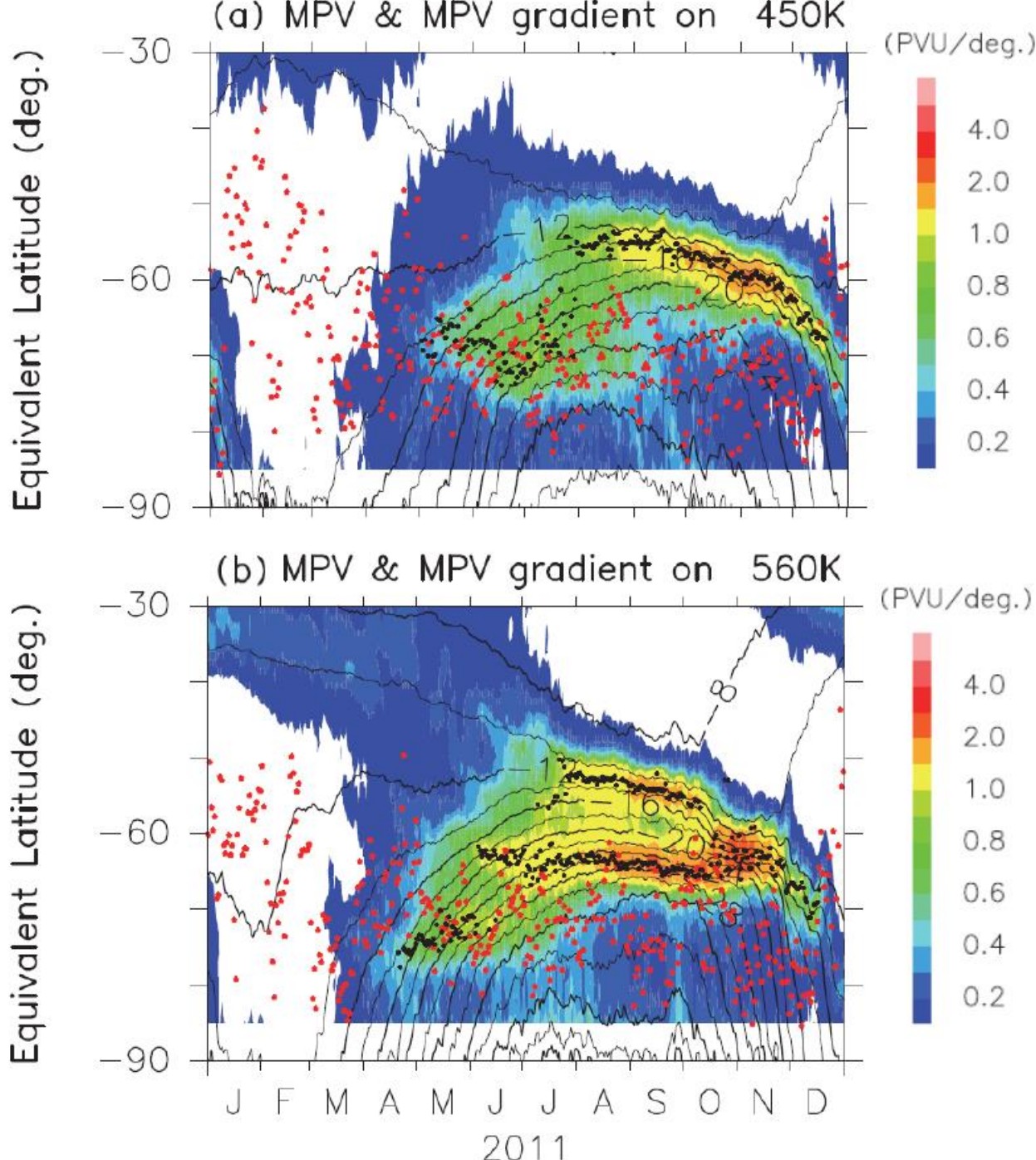

Figure A2. Same as Figure A1 but for the year in 2011.