# Peer review of "Chlorine partitioning near the polar vortex edge observed with ground-based FTIR and satellites at Syowa Station, Antarctica in 2007 and 2011"

_Atmospheric Chemistry and Physics, 2019_

## Author Comment (AC1) · 17 Jun 2019

The submitted draft has a mistake in Figure 7. We used wrong figure for Fig. 7 (22 km, 2007; instead of 22 km, 2011). I attached correct (new) Figure 7, in addition to the full draft which include correct Figure 7. We are sorry for the mistake.

Please also note the supplement to this comment:
https://www.atmos-chem-phys-discuss.net/acp-2019-443/acp-2019-443-AC1-supplement.pdf

[Figure]

[Figure]

Figure 7. Same as Figure 5 but at 22 km.

**Fig. 1.**

**Supplement:**

[revised manuscript text omitted]

---

## Short Comment (SC1) · 23 Jun 2019

This is an interesting paper. There is relevant history in the following references, which support some of the authors' points. J Chem Soc Farad Trans, 91,3063-3071,(1995). JGR-D, 102(D11),13,325-13,253 (1997). Faraday Discussions, 100, 389-410 (1995). Geophys. Res. Lett., 24, 2651-2654 (1997).

The last of these defines a process that has relevance to photochemistry at the vortex edge and which apparently has not been considered.

---

## Referee Comment (RC1) · Anonymous Referee #2 · 24 Jun 2019

Chlorine partitioning near the polar vortex boundary observed with ground-based FTIR and satellites at Syowa Station, Antarctica in 2007 and 2011.

The authors use a ground-based FTIR spectrometer in Antarctica as well as the satellite-based MIPAS instrument to observe ozone depletion chemistry during 2011 and 2007. The observed relationships between the chlorine reservoir and active species are investigated using a chemistry model, which reproduces the changes in abundances of ClONO2, HCl, ClO, HNO3, and O3.

[Figure]

General comments:

The manuscript is well-written and organized. The introduction does a good job of providing an overview. However, I'm curious to know how the study's results compare to other results, e.g., was the amount of depletion typical? Also, the FTIR was installed in March 2007, so why weren't results included from years other than 2007 and 2011? What is the current state of the instrument? Perhaps a comment about ongoing work could be added to the conclusions.

In terms of the FTIR retrieval method, why is SFIT2 used, rather than the up-to-date SFIT4? Also, it was helpful to see an example of the averaging kernel, but I'm curious to know what the DOFS were, especially near the altitude of interest.

For the validation, it is hard to draw firm conclusions from the ozone validation using only 14 coincidences. Indeed, these comparisons had notable scatter (e.g., P6L20-28). Why not also add MLS comparisons, especially since MLS is used for HNO3 and HCl measurement validation. Indeed, later in the manuscript, MLS O3 is shown (Fig. 11) with comparisons to the chemistry model.

Specific comments:

P2L12: Suggest removing "month" or reword "only in the month of September".

P2L22-25: The sentence grammar should be revised, e.g. the semi-colon.

P5L34: The list of tangent point altitudes is incomplete in the middle.

P6L18: Could you elaborate on "slit function"?

P7L10: Were the Livesey results covering the same latitudinal range as this study? Was this bias between MLS and HALOE only seen in HCl?

P7L10: Livesey et al. (2013) not in references. Couldn't locate the study. Please update.

[Figure]

P9L30: Remove "later" after Figure 10. Define EL when mentioning equivalent latitude in the Figure 10 caption since it's used in the figure colorbar label.

P10L32: Can the satellite coincidences be filtered to ensure similar equivalent latitude to Syowa Station, in addition to the distance/temporal criteria applied?

P11L1: It's preferable to spell the term and then put the acronym in brackets, e.g., "Reduced Major Axis (RMA)", which has been done elsewhere in this manuscript.

P13L27-31: Nice result about the ClONO2 transport.

P18L29: Misspelled "Livesey" as "Liversey".

Table 3: Suggest adding number of coincidences contributing to the altitudes of interest

Fig. 4: Between day 310 and 320, the time spent outside the vortex is a nice illustration of how significant the boundary is for ozone chemistry. Clear support for the criteria used to define these boundaries.

---

## Referee Comment (RC2) · Anonymous Referee #1 · 3 Jul 2019

This paper uses ground-based FTIR measurements of stratospheric trace gases at the Syowa Station, as well as satellite and model data to investigate relationships between chlorine species during spring-time ozone depletion. A variety of interesting datasets are presented, but it is difficult to follow the arguments that the authors are putting forward because of how the paper is organized and how figures are presented. Overall, I think that the paper could make a good contribution to ACP after major revisions, as suggested below.

*

[Figure]

GENERAL COMMENTS

*

More context for the FTIR data should be given. A few questions come to mind.

- Is this the first time that the Syowa FTIR data has been retrieved/written up? Or are there other papers that describe this instrument and the data?

- Why is there only 2007 and 2011 data? Was the instrument not deployed in other years?

- Are the data publicly available?

- How to these data fit in with other ground-based FTIR datasets collected at high latitudes? Is the data quality similar? Have other FTIR instruments been used to study trace gases during spring-time ozone depletion?

*

The validation sections should be more specific in terms of how validation methods are applied (see specific comments for details). It would also be useful to answer:

- What is the expected uncertainty in the FTIR measurements? Is this something you retrieve?

- How do your validation results compare with validation of other ground-based FTIR instruments? Is the instrument at Syowa performing similarly to other FTIR?

- Are there any other factors that could affect comparison results? Do any of these species vary diurnally? Is there a chance that the satellite and station are measuring very difference air masses (e.g., inside/outside the vortex) for some coincidences? If either of these are factors, how does this affect interpretation of the data in Sect. 4?

*

The discussion of results is repetitive and at times is difficult to follow. There are many

timeseries figures (Fig. 4-9; Fig. 11-12). I found it hard to track through these and determine what exactly the authors were trying to highlight. I also had a hard time connecting all of these pieces to the concluding arguments around transport, chlorine deactivation, etc. A few possible solutions:

- Determine what you are trying to show, perhaps starting from the statements made in the conclusion. Only include figures/discussion that are relevant to what you're trying to show. Move additional timeseries figures into an appendix or supplement. Some tables (Table 1, Table 4) could also be moved out of the main body of the paper.

- For Fig. 8-9 is there a more concise way of showing the relationships between the chlorine species than using the timeseries (e.g., through scatter plots or something else)?

- Can the comparison with the modelling data be merged in with the validation of the FTIR (Sect. 3)? Can the modelling timeseries figures be folded into the timeseries figures presented in previous sections?

*

I found it hard to follow the discussion around the mechanisms behind the decrease of HCl. It would be helpful if the discussion was expanded, to help answer the following:

- What hypotheses are there for the decrease in HCl in the literature? Is there just Solomon et al. (2015) and Grooss et al. (2018)? Are these hypotheses in conflict with each other or is it possible that they both contribute to the decrease in HCl together?

- For the existing hypotheses, what data/evidence were used to develop the hypotheses? How does the data that you collected add to the existing supporting data/evidence?

- Why does the sporadic increases in ClONO2 in the model data support the transport mechanism? Do you have any other data to support that ClONO2 is being transported (e.g., maps of ClONO2, tracers showing transport patterns, evidence in the satellite

data)? Are there alternative mechanisms that could explain your observations, such as chemistry?

- Can your data be used to refute any other hypotheses for decreasing HCl?

*

SPECIFIC COMMENTS:

- Page 1, Line 17: You state "This was the first continuous measurements of chlorine species throughout the ozone hole period from the ground in Antarctica" here and elsewhere in the paper. This statement is a bit vague. Have other studies looked at any chlorine species from the ground in Antarctica? Are they looking at fewer species? Or for shorter time periods? It would be helpful if you included a literature review of ground-based measurements (FTIR and maybe other measurements of chlorine, such as OClO from UV-vis?) in Antarctica in your introduction.

- Page 4, Line 25: Are there other ground-based FTIRs in Antarctica?

- Page 5, Line 1: What is the temporal resolution of the FTIR at Syowa? Does it only take measurements during sunny days? Does it require manual operation?

- Page 5, Line 18: Is there a way of using the averaging kernel to determine whether the 18 km and 22 km concentrations are independent from each other?

- Page 5, Line 25: State where various satellite datasets were obtained from? (E.g., URL?)

- Page 5, Line 24: The application of selection criteria should be clarified, perhaps with a full paragraph at the end of this section. Was this same criteria applied to all three satellite instruments (MLS, MIPAS, and CALIPSO)? Was the same criteria used for both the validation (Sect. 3) and the discussions of results over Syowa (Sect. 4)? E.g., for the timeseries figures was the 6 h criterion applied?

- Page 5, Line 30: Have you considered applying a selection criteria based on inside/outside the polar vortex to the satellite data along their line of sight instead of using values over the station for all measurements?

- Page 6, Line 18: What is the shape of the 5 km-wide slit function? Is this based on the FTIR resolution? Why was this used instead of the averaging kernel? Also, why are the MLS data smoothed? Is the vertical resolution for MLS much higher than the ground-based FTIR?

- Page 6, Line 25: State the expected uncertainty and any expected biases in ozonesonde measurements

- Page 6, Line 24: I'm a bit confused about the language used throughout this section. The figures show mean relative difference and mean absolute difference versus altitude. Replace "The absolute difference. . ." with "The mean absolute difference. . ." and "The mean relative difference" in this line.

- Page 6, Line 25: What do you mean by "mean relative difference" here? Is this the mean relative differences averaged again over an altitude range? It might be simpler just to state the mean relative difference values at 18 km and 22 km instead of taking another average over altitude.

- Page 6, Line 26: What to you mean by "within error bars"? Are you referring to the standard deviation in the mean differences? This isn't really an error – it's the variability in the comparisons. It would make more sense to determine whether the agreement is as expected based on estimated uncertainty in the FTIR measurements (if this exists) and known uncertainties/biases in the satellite data. Also, why is the standard deviation used for the comparisons instead of the standard error?

- Page 6, Line 31 – Page 7, Line 2: See comments for previous paragraph.

- Page 7, Line 10: "with a precision of 0.2-0.6%" – is this the precision in the MLS measurements? Or is this the precision of the systematic bias?

- Page 8, Line 6: You describe a set of steps for detecting the inner and outer edges of

the vortex. Is this a new method? Is this expected to work better than other established methods for some cases?

- Page 8, Lines 11-14: Please break this down into smaller steps – I have a hard time understand what was done. My best guess is that you calculated the isentropic potential vorticity gradient as a function of equivalent latitude. You found the local maxima. You defined the inner/outer edges of the vortex at the local maxima that both (?) exceeded the wind-speed threshold and were at least $5°$ in equivalent latitude apart? Are there always only two local maxima that meet these criteria?

- Page 8, Line 18: Did you filter the MLS data according to an error threshold? Or did you just remove suspect data when you looked at the timeseries? All filters applied to the satellite data (for, e.g., uncertainty, etc) should be described explicitly in Sect. 2.2.

- Page 11, Lines 10-24: Can the model comparisons be merged into Sect. 3? It would be nice if consistent comparison methods were applied. Similar to Sect. 3 – are the comparisons consistent with what is expected based on known model performance and known biases in the satellite instruments?

- Page 11, Line 28: Why was a different method used to determine the polar vortex than at Syowa station?

- Page 11, Line 31: What do you mean by "This boundary was located in between the inner and the outer edge of the polar vortex as were defined in Sect. 4.1"? Did you compare the two definitions of the vortex? If so, was this just done at Syowa station for all data or a subset of data? Or is it the case that the max gradient of PV at 475 K is always between the inner/outer edge values by definition?

- Table 4: Should describe how you came up with each of the various ranges presented in the table. Here are a few examples of questions that should be answered. How do you define the threshold for a ClONO2 enhancement or ozone starting-ending day of decrease? What are given in "HCl Value after increase" ranges? Is this the min/max

of individual FTIR measurements over a given time-period? For the ClO enhancement period, is the 80% of maximum value different in 2007 and 2011 or is a single value used? What does "Variation when HCl $\sim$0 ppbv" mean?

- Figure 3: Can you add a panel describing hours of sunlight or SZA here or to one of your other timeseries? It would be helpful to visualize this through discussions around available sunlight.

- Figure 3: Are the gaps in the PSC timeseries because CALIPSO observed no PSCs or because CALIPSO didn't collect any coincident data during this time? Either describe in text and/or add marker for CALIPSO measurements which did not observe PSCs.

- Figures 14/15: Why have you used model data instead of satellite data for the parameters that are available from satellite? Did you check to see if the satellite saw the same patterns as the model?

*

MINOR/TECHNICAL COMMENTS:

- Throughout the text, there are some awkwardly worded sentences and minor grammatical mistakes. I assume that these would be corrected through copy-editing, so have not listed these here, unless the meaning is unclear.

- Table 2: Define "PT"

- Table 3: Should define the fields included in the table. Is D(%) 18-22 km the average of the mean relative differences? Is (Min/Max %) 18-22 km the max/min mean relative difference across the various altitudes? (Might be better just to show mean relative differences +/- standard error at 18 km and at 22 km.)

- Table 3: See comment re: precision in Page 7, Line 10

- Table 3: Replace "Agreement" with "Range of mean absolute differences for 15-25

km"

- Figure 2, caption: Replace "Absolute (a) and percentage" with "Mean absolute (a) and mean percentage"

- Figure 2, caption: Replace "FTIR measurements and those from ozonesonde" with "FTIR measurements minus those from ozonesonde"

- Figure 2, caption: Move "Horizontal bars indicate the standard deviation of differences at each altitude." to the end of the caption, since it applies to all panels.

- Figure 2, caption: Describe what the horizontal dashed lines indicate.

- Figure 3, caption: Replace "from N2O value" with "from Aura/MLS N2O".

---

## Referee Comment (RC3) · Anonymous Referee #3 · 24 Jul 2019

**Review of "Chlorine partitioning near the polar vortex . . . "**

BY NAKAJIMA ET AL.

**General**

It is now more than three decades ago that the Antarctic ozone hole was discovered (WMO, 2019); by now the processes involved in its formation are thought to be under-

stood in some detail. Current state-of-the-art models reproduce the observed spring-time ozone loss in the polar stratosphere with good accuracy (e.g., Khosrawi et al., 2009; Chipperfield et al., 2017; WMO, 2019; Froidevaux et al., 2019). Nonetheless, there are open questions in Antarctic chlorine and ozone chemistry. Such questions can be addressed when new observations, such as those reported in the manuscript, become available. Insofar, this manuscript is an important contribution to ACP.

I suggest an extension of the discussion in the manuscript regarding several issues (see in particular detailed comments below). Briefly, the Cly correlation needs to be adjusted to 2007 and 2011 (or the adjustment made should be described, see below) and the model behaviour reported by Grooß et al. (2018) that I think is also found here (namely that the models show HCl remaining in the core of the vortex) should be discussed in terms of MIROC3.2.

Further, the manuscript could make a better contribution to addressing the issue of a "race" between chlorine activation and deactivation (Solomon et al., 2015; Müller et al., 2018; Zafar et al., 2018) and the question of which HCl formation processes are responsible for the observed HCl increase at the end of the ozone hole period.

Overall, the FTIR measurements presented here are certainly of great scientific interest and the measurements are combined in a meaningful way with satellite information. Moreover, a model simulation is included that helps the interpretation of the measurements. Notwithstanding these points, I suggest a substantial extension of the discussions on Antarctic chlorine chemistry in the paper. Provided the (in my view) necessary extensions are done, I would expect that this paper would make a great contribution to ACP.

**Comments in Detail**

Year-to-year variability of Cly

The main driver for antarctic ozone loss is the available Cly but there is a substantial year-to-year variability in this quantity; this issue is discussed by Strahan et al. (2014). How are the years discussed here (2007, 2011) ranked in the observed variability of Cly (Strahan et al., 2014)? Further, the applicability of the employed empirical relation for Cly to the years discussed here (2007, 2011) needs to be addressed in the paper.

HCl remains in the core of the polar vortex in austral winter

The authors find here for the MIROC3.2 CCM that some HCl remains in winter in the core of the polar vortex in darkness. Such a model behaviour is expected as there is not enough ClONO2 and no light in the core of the vortex. This model behaviour was reported by Grooß et al. (2018) for three models. Further they showed that this model feature is not found in observations of HCl. I suggest stating that the MIROC3.2 CCM shows the same issue – provided that the authors agree. If they do not agree, there should be a discussion of the issue in the paper.

Overestimated transport across the vortex edge in models

The paper states that transport across the vortex edge might be overestimated in models. I think here is the potential for the paper to make an important contribution. The issue that models might overestimate mixing into the vortex edge has been discussed (e.g.) by Hoppe et al. (2014); indeed for the same Eulerian transport scheme as employed in MIROC3.2. Observations of N2O (and the N2O gradient might help to eludicate the model issue. In any case a bit more discussion on this point is warranted

Continuous loss of HCl in the core of the polar vortex

The authors state the following conclusion in the paper: "Continuous loss of HCl was seen at 87.9°S between days 160 and 200 even after the disappearance of the counterpart of heterogeneous reaction (R1) (Figure 15(e)). The cause of this continuous loss was unknown until recently, where a hypothesis was proposed that includes the effect of decomposition of particulate HNO3 by some process like ionisation caused by galactic cosmic rays during the winter polar vortex (Grooß et al., 2018). Solomon et al. (2015) proposed a new mechanism on this issue: Continuous transport of ClONO2 from the subpolar regions near 55-65°S to higher latitudes near 65-75°S provides a flux of NOx from more sunlit latitudes into the polar vortex. Our result is consistent with the mechanism indicated by some sporadic increase in ClONO2 at around days 158, 179, and 189 at 76.7°S as shown in Figure 15(f)".

First, the processes described by Grooß et al. (2018) and Solomon et al. (2015) are very different: Grooß et al. (2018) describe a polar night process, whereas Solomon et al. (2015) describe a dynamical process (acting later in the course of the existence of the polar vortex), which needs light nonetheless as formation of ClONO2 is involved.

Further note that Grooß et al. (2018) discussed the transport mechanism and concluded that it *cannot* explain the so-called 'HCl-discrepancy'. Of course the present manuscript might come to a different conclusion but I think a more extensive evaluation of the arguments put forward by Grooß et al. (2018) is necessary here. In this context, Fig. 15 of the manuscript could be important; could it be that the 'discrepancy' reported by Grooß et al. (2018) is also noticeable in panel (e) of Fig. 15?

Second, an alternative explanation for the continuing decline of HCl could be the formation of HOCl under sunlit conditions which would allow HCl to decline to zero values (Grooß et al., 2011; Müller et al., 2018). If the authors cannot develop a preference for one of these mechanisms based on their data/observations, I suggest to state both alternatives in the discussion in the paper.

This issue reflects the importance of the reaction HCl + HOCl -> Cl2 + H2O (R4); early papers (Prather, 1992; Crutzen et al., 1992) have pointed to the importance of this reaction for bringing down the HCl concentrations in the polar vortex in Antarctica.

Finally, the transport of ClONO2 from the vortex edge to the vortex core does not occur in isolation; inspection of Fig. 15 would suggest to me that mixing from the vortex edge would transport an amount of x ppb of ClONO2 but at the same time an amount of x ppb HCl as well. Thus there is no net removal of HCl by "mixing". As stated above – the easiest solution would be to discuss both alternative explanations. Further, these alternative are not necessarily contradictory; both mechanisms could contribute partly to the observed HCl decline.

"Race" between chlorine activation and deactivation

An aspect of polar ozone and chlorine chemistry, where different concepts are discussed in the literature is the maintenance of enhanced levels of active chlorine during the time period (September and early October) when rapid ozone loss occurs. One concept is the one of a "race" between chlorine activation and deactivation, i.e., a competition of the heterogeneous reactions R1, R2, and R4 and gas-phase reformation of HCl and ClONO2 (R12, R13) (Solomon et al., 2015). The other concept is the one of so-called "HCl null-cycles", where the formation of HCl (R13) is followed by immediate reactivation of HCl (Müller et al., 2018; Zafar et al., 2018). (See also the discussion in WMO, 2019). The measurements presented in this manuscript might help to shed some light on these issues – this could be a contribution of this paper. Alternatively, if the measurements presented here cannot contribute to discriminating between the two discussed processes, this could also be a result of the paper (which should be mentioned).

Formation of HCl – Antarctic deactivation

The presented observations demonstrate that the deactivation in the Antarctic is through formation of HCl and that the deactivation is rapid. This is good and important. However, it is also stated that the formation of HCl is via the reaction CH4 + Cl -> HCl + CH3 (R13). I agree that this HCl formation is "common wisdom" (e.g. Crutzen et al., 1992; Douglass et al., 1995). However the authors have also mentioned the reaction CH2O + Cl -> HCl + CHO (R14). What is the evidence that the observed formation of HCl is indeed via R13? Further, it should be taken into account that there is also reactivation of chlorine even if reactions R14 and R13 occur at a considerable rate (Müller et al., 2018; Zafar et al., 2018). Again, the paper could make a contribution here, but at least there should be more discussion of all the processes playing a role here.

Negative correlation between ClO and ClONO2

The cause of the negative correlation between ClO and ClONO2 (e.g. Fig. 10) is discussed as being caused by the distance of Syowa station relative to the vortex. I do not think that this is entirely correct. The main reason for the negative correlation is that ClO is converted into ClONO2 and vice versa. (However the rather large scatter in Fig. 10 should also be acknowledged.) Given this fact, the higher values of ClONO2 seem to preferentially occur closer to the vortex edge (so that the location of Syowa indeed is relevant).

Near zero values of ozone

The observations of ozone reported here (ozone sonde measurements show very low values in October. Such near zero values of ozone have been reported before

(Solomon et al., 2005); are the ozone values reported here compatible with the reported low ozone values? Perhaps the ozone values could be replotted on a log-scale as is the earlier publication (Solomon et al., 2005) – perhaps it is too much to show such plots in the paper, but an electronic supplement might be an alternative.

FTIR measurements

The FTIR measurements are a major contribution of this paper. This is why I suggest to make the data available for other researchers as well. Further Toon and Farmer (1989) have reported measurements of the HOCl integrated vertical column abundance, which was inferred from high resolution infrared solar spectra measured by the JPL MkIV interferometer from the NASA DC-8 aircraft during flights over Antarctica in September 1987. Would the current set-up also allow measurements of HOCl? Even quantifying an upper limit might be helpful. Other species of potential interest would be methanol or formaldehyde.

Model description

The results of the MIROC3.2 model make an important contribution to the study. However, the model documentation is not sufficient. There is a short paragraph (on p. 15) and the reference to Akiyoshi et al. (2016). But even after consulting these pieces of information, many aspects of the model remain unclear. Which photolysis scheme is used for the calculations presented here; is the scheme using spherical geometry? I think a reference to the employed scheme would be appropriate. Which solver is employed for solving the set of differential equations that result from the considered chemical scheme? How exactly (which surfaces?) is heterogeneous chemistry (including particle formation) treated in the model?

Further, it is not clear which reactions (and which species) have been considered in

the presented model calculations. But this aspect could be important. I think it would be very helpful (and very easy) to add the information in question (e.g. add a list of reactions as an electronic appendix) to the paper.

Data Availability

There is no data availability statement in this paper. According to the rules of ACP such a statement should be added to the final version of the paper. This point would both regard all observations (including FTIR, ozone sondes and the satellite information shown in the plots of the paper) and the model results. It is up to ACP, but I believe that making the data (in particularly the unique FTIR data) available would enhance the impact of this paper.

**Details**

- p. 2, l. 10: "and the observed"

- p. 2, l. 11: add a citation for the observed ozone hole magnitude

- p. 3, l. 5: citations for these reactions? In particularly, R14 is often mentioned regarding chlorine deactivation.

- p 3, l 8: R14 is not often mentioned as a HCl forming reaction; suggest adding a reference. Perhaps also for R 12 and R 13.

- p. 3, l 21: in *all* years?

- p. 3, l. 28: "sometimes" is not right, is it? It happens in the Arctic always if chlorine activation occurs, I'd argue. Probably the first observations of this phenomenon were reported by von Clarmann et al. (1993); Oelhaf et al. (1994).

- p 4, l. 5: citation for the FTIR measurement? How much can we learn about the vertical resolution from Fig. 1?

- p 4, l. 22: Farman et al. (1985) did not show ozone sonde measurements. By the way, another ozone sonde measurement was conducted in 1985 by Gernandt (1987).

- p. 4, l 1: I think a further advantage is also the location inside and outside of the vortex core and the inner vortex transport barrier.

- p. 7, l. 19: 4.5 ppm is not the best value if dehydration occurs.

- p. 7, l. 30: have these ozone sondes been compared in sonde comparison studies?

- p. 8, l. 4: Such an empirical relation is not valid for arbitrary years: it should be explained how the adjustment to the conditions of 2007 and 2001 has been done (it is necessary to both correct N2O and Cly).

- p 8., l. 14: the issue of a transport barrier within the Antarctic polar vortex was also discussed by Lee et al. (2001). The transport barrier within the Antarctic polar vortex in the early vortex can also be seen in ILAS measurements (Tilmes et al., 2006).

- p 8., l. 28: Within the vortex (but at the boundary) much lower HCl would be expected than outside of the vortex. Is this not seen by the FTIR measurements?

- p. 9, l. 3: what is the uncertainty range of the FTIR measurements? (report a $\pm$ here).

- p. 9, l. 21: inside, but in the core or at the edge?

- p. 9., l. 26: "partitioning" is not really clear; I believe you mean HCl/Cly.

- p. 9, l. 33: probably not via R13, see above.

- p. 11, l. 17: "systematically smaller": could you quantify this statement?

- p. 11, l. 19: faster mixing in the model than in the real world might indeed be an issue (see above).

- p. 11, l. 21: for which year is the correlation – the correlation need to be adjusted to the years in question here (2007 and 2011).

- p. 11., l. 32: Note that the onset of heterogeneous chemistry occurs very likely before PSCs for and that PSCs do not form at NAT equilibrium. You could formulate: temperatures low enough for the onset of heterogeneous chemistry.

- p. 11., l. 32: NO2 does not condense (it needs to be chemically converted first).

- p 12, l. 2: "Some HCl remains" this is expected in the model as there is not enough ClONO2 (as stated here) and no light in the core of the vortex. This model behaviour was also shown by Grooß et al. (2018) and I suggest stating that the MIROC3.2 CCM shows the same issue (see also above).

- p. 12, l. 16: Note that HCl remains low even under these conditions, when a relatively fast rate of reaction R13 should occur in the stratosphere. HCl-null cycles (Müller et al., 2018) could be an explanation.

- p 12, l. 21: How sure can we be that the recovery is via R13??

- p. 13., l. 10: compare Grooß et al. (2018)

- p 13, l. 22: Again, how sure can we be that the recovery is via R13?? What about R14? What about other possible HCl forming reactions?

- p 13, l. 25 and below: The processes described by Grooß et al. (2018) and Solomon et al. (2015) should be distinguished: Grooß et al. (2018) describe a polar night process, whereas Solomon et al. (2015) describe a dynamical process which needs light nonetheless as formation of ClONO2 is involved.

- p. 14, l. 9: could you quantify "well below"

- p. 14., l. 13: the reason for the negative correlation is the conversion of ClO to ClONO2 (see above).

- p. 14, l. 21: As discussed above, it is not sure that the transport of ClONO2 is the only possible explanation for the behaviour of HCl.

- p. 14., l. 23: NOx rich would also mean rich in HCl – correct?

- p 15., l. 14: Sander et al 2010 or 2011? (see reference list).

- p. 16., l. 20: Stimpfle

- p. 17, l. 29: 1977?

- p. 20, l. 10: Karin Labitzke

- p 22, l. 2: activation (no hyphen)

- p. 24, 25: Could you add the information on the vertical resolution of the FTIR measurements to one of these tables?

- p. 39: pane -> panel

**References**

Akiyoshi, H., Nakamura, T., Miyasaka, T., Shiotani, M., and Suzuki, M.: A nudged chemistry-climate model simulation of chemical constituent distribution at northern high-latitude stratosphere observed by SMILES and MLS during the 2009/2010 stratospheric sudden warming, J. Geophys. Res., 121, 1361–1380, https://doi.org/10.1002/2015JD023334, https://agupubs.onlinelibrary.wiley.com/doi/abs/10.1002/2015JD023334, 2016.

Chipperfield, M. P., Bekki, S., Dhomse, S., Harris, N. R. P., Hassler, B., Hossaini, R., Steinbrecht, W., Thieblemont, R., and Weber, M.: Detecting recovery of the stratospheric ozone layer, Nature, 549, 211–218, https://doi.org/{10.1038/nature23681}, 2017.

Crutzen, P. J., Müller, R., Brühl, C., and Peter, T.: On the potential importance of the gas phase reaction $CH_3O_2 + ClO \rightarrow ClOO + CH_3O$ and the heterogeneous reaction $HOCl + HCl \rightarrow H_2O + Cl_2$ in "ozone hole" chemistry, Geophys. Res. Lett., 19, 1113–1116, https://doi.org/10.1029/92GL01172, 1992.

Douglass, A. R., Schoeberl, M. R., Stolarski, R. S., Waters, J. W., Russell III, J. M., Roche, A. E., and Massie, S. T.: Interhemispheric differences in springtime production of HCl and $ClONO_2$ in the polar vortices, J. Geophys. Res., 100, 13 967–13 978, 1995.

Farman, J. C., Gardiner, B. G., and Shanklin, J. D.: Large losses of total ozone in Antarctica reveal seasonal $ClO_x/NO_x$ interaction, Nature, 315, 207–210, 1985.

Froidevaux, L., Kinnison, D. E., Wang, R., Anderson, J., and Fuller, R. A.: Evaluation of CESM1 (WACCM) free-running and specified dynamics atmospheric composition simulations using global multispecies satellite data records, Atmos. Chem. Phys., 19, 4783–4821, https://doi.org/10.5194/acp-19-4783-2019, https://www.atmos-chem-phys.net/19/4783/2019/, 2019.

Gernandt, H.: The vertical ozone distribution above the GDR research base, Antarctica in 1985, Geophys. Res. Lett., 14, 84–86, 1987.

Grooß, J.-U., Brautzsch, K., Pommrich, R., Solomon, S., and Müller, R.: Stratospheric ozone chemistry in the Antarctic: What controls the lowest values that can be reached and their recovery?, Atmos. Chem. Phys., 11, 12 217–12 226, 2011.

Grooß, J.-U., Müller, R., Spang, R., Tritscher, I., Wegner, T., Chipperfield, M. P., Feng, W., Kinnison, D. E., and Madronich, S.: On the discrepancy of HCl processing in the core of the wintertime polar vortices, Atmos. Chem. Phys., pp. 8647–8666, https://doi.org/10.5194/acp-18-8647-2018, 2018.

Hoppe, C. M., Hoffmann, L., Konopka, P., Grooß, J.-U., Ploeger, F., Günther, G., Jöckel, P., and Müller, R.: The implementation of the CLaMS Lagrangian transport core into the chemistry climate model EMAC 2.40.1: application on age of air and transport of long-lived trace species, Geosci. Model Dev., 7, 2639–2651, https://doi.org/10.5194/gmd-7-2639-2014, http://www.geosci-model-dev.net/7/2639/2014/, 2014.

Khosrawi, F., Müller, R., Proffitt, M. H., Ruhnke, R., Kirner, O., Jöckel, P., Grooß, J.-U., Urban, J., Murtagh, D., and Nakajima, H.: Evaluation of CLaMS, KASIMA and ECHAM5/MESSy1 simulations in the lower stratosphere using observations of Odin/SMR and ILAS/ILAS-II, Atmos. Chem. Phys., 9, 5759–5783, 2009.

Lee, A., Roscoe, H., Jones, A., Haynes, P., Shuckburgh, E., Morrey, M., and Pumphrey, H.: The impact of the mixing properties within the Antarctic stratospheric vortex on ozone loss in spring,, J. Geophys. Res., 106, 3203–3211, https://doi.org/10.1029/2000JD900398, 2001.

Müller, R., Grooß, J.-U., Zafar, A. M., Robrecht, S., and Lehmann, R.: The maintenance of elevated active chlorine levels in the Antarctic lower stratosphere through HCl null cycles, Atmos. Chem. Phys., 18, 2985–2997, https://doi.org/10.5194/acp-18-2985-2018, https://www.atmos-chem-phys.net/18/2985/2018/, 2018.

Oelhaf, H., v. Clarmann, T., Fischer, H., Friedl-Vallon, F., Fritzsche, C., Linden, A., Piesch, C., Seefeldner, M., and Völker, W.: Stratospheric $ClONO_2$ and $HNO_3$ profiles inside the Arctic vortex from MIPAS-B limb emission spectra obtained during EASOE, Geophys. Res. Lett., 21, 1263–1266, https://doi.org/10.1029/93GL01303, 1994.

Prather, M. J.: More rapid ozone depletion through the reaction of HOCl with HCl on polar stratospheric clouds, Nature, 355, 534–537, 1992.

Solomon, S., Portmann, R. W., Sasaki, T., Hofmann, D. J., and Thompson, D. W. J.: Four decades of ozonesonde measurements over Antarctica, J. Geophys. Res., 110, D21311, https://doi.org/10.1029/2005JD005917, 2005.

Solomon, S., Kinnison, D., Bandoro, J., and Garcia, R.: Simulation of polar ozone depletion: An update, J. Geophys. Res., 120, 7958–7974, https://doi.org/10.1002/2015JD023365, 2015.

Strahan, S. E., Douglass, A. R., Newman, P. A., and Steenrod, S. D.: Inorganic chlorine variability in the Antarctic vortex and implications for ozone recovery, J. Geophys. Res., https://doi.org/10.1002/2014JD022295, http://dx.doi.org/10.1002/2014JD022295, 2014.

Tilmes, S., Müller, R., Grooß, J.-U., Nakajima, H., and Sasano, Y.: Development of tracer relations and chemical ozone loss during the setup phase of the polar vortex, J. Geophys. Res., 111, D24S90, https://doi.org/10.1029/2005JD006726, 2006.

Toon, G. C. and Farmer, C. B.: Detection of HOCl in the Antarctic stratosphere, Geophys. Res. Lett., 16, 1375–1377, https://doi.org/10.1029/GL016i012p01375, https://agupubs.onlinelibrary.wiley.com/doi/abs/10.1029/GL016i012p01375, 1989.

von Clarmann, T., Fischer, H., Friedl-Vallon, F., Linden, A., Oelhaf, H., Piesch, C., Seefeldner, M., and Völker, W.: Retrieval of Stratospheric $O_3$, $HNO_3$ and $ClONO_2$ profiles from 1992 MIPAS–B limb emission spectra: method, results and error analysis, J. Geophys. Res., 98, 20 495–20 506, 1993.

WMO: Scientific assessment of ozone depletion: 2018, Global Ozone Research and Monitoring Project–Report No. 58, Geneva, Switzerland, 2019.

Zafar, A. M., Müller, R., Grooß, J.-U., Robrecht, S., Vogel, B., and Lehmann, R.: The relevance of reactions of the methyl peroxy radical ($CH_3O_2$) and methylhypochlorite ($CH_3OCl$) for Antarctic chlorine activation and ozone loss, Tellus B: Chemical and Physical Meteorology, 70, 1–18, https://doi.org/10.1080/16000889.2018.1507391, https://doi.org/10.1080/16000889.2018.1507391, 2018.

---

## Author Comment (AC2) · 12 Sep 2019

**Reply to Short Comment #1**

We thank Dr Adrian Tuck for his comment on the history of related work on chlorine chemistry.

*This is an interesting paper. There is relevant history in the following references, which support some of the authors' points. J Chem Soc Farad Trans, 91,3063-3071,(1995). JGR-D, 102(D11),13,325-13,253 (1997). Faraday Discussions, 100, 389-410 (1995). Geophys. Res. Lett., 24, 2651-2654 (1997). The last of these defines a process that has relevance to photochemistry at the vortex edge and which apparently has not been considered.*

Thank you for the information on the related previous work to this current study. We now added Tuck et al. (1995) and Jaeglé et al. (1997) to show examples of high-altitude aircraft measurements in the Introduction. Also, the loss of HCl at the vortex edge by the resupply of HOCl and ClONO$_2$ mentioned in Jaeglé et al. (1997) was cited in the discussion in Section 4.6.

In the revised draft with corrections by track change, the above corrections were shown by yellow-marked sentences, while red, purple, and blue corrections are the revisions suggested by reviewers #1, #2, and #3, respectively.

---

## Author Comment (AC3) · 12 Sep 2019

**Reply to reviewer #2**

We thank anonymous reviewer #2 for his/her constructive review that would improve the contents of our paper. The review comments by anonymous reviewer #2 are numbered and repeated below as *in italic letters*, followed by our answers.    In the new draft with corrections (supplement file), red, purple, and blue corrections are the revisions suggested by reviewers #1, #2, and #3, respectively.    Yellow-marked sentences were also added in response to Short Comment #1 by Dr. Adrian Tuck.

*General comments:*

*(1) The manuscript is well-written and organized. The introduction does a good job of providing an overview. However, I'm curious to know how the study's results compare to other results, e.g., was the amount of depletion typical? Also, the FTIR was installed in March 2007, so why weren't results included from years other than 2007 and 2011? What is the current state of the instrument? Perhaps a comment about ongoing work could be added to the conclusions.*

According to WMO's Scientific Assessment of Ozone Depletion (2018), 2007 is almost in average, and 2011 is a bit larger ozone loss year compared with other recent average years (Chapter 4, Figure 4-6, P. 4.11).    Due to the operator's man power problem, operation of FTIR was only done in 2007 and 2011.    After short operation of FTIR in 2016, it was now brought back to Japan.    This was explicitly explained in the text now.

*(2) In terms of the FTIR retrieval method, why is SFIT2 used, rather than the up-to-date SFIT4? Also, it was helpful to see an example of the averaging kernel, but I'm curious to know what the DOFS were, especially near the altitude of interest.*

When we started the FTIR analysis for Syowa Station several years ago, SFIT4 was not available yet.    After a while, SFIT4 became available, but we have already processed all the Syowa FTIR data by SFIT2.    We have processed few Syowa spectra by SFIT4 and compared the result with the one by SFIT2.    Fundamentally, we have found no major differences between them.    Therefore, we decided to use the results processed by SFIT2 in this paper.    The information of typical vertical resolution and mean DOFS were now shown in Table 2.

*(3) For the validation, it is hard to draw firm conclusions from the ozone validation using only 14 coincidences. Indeed, these comparisons had notable scatter (e.g., P6L20-28). Why not also add MLS comparisons, especially since MLS is used for HNO3 and HCl measurement validation? Indeed, later in the manuscript, MLS O3 is shown (Fig. 11) with comparisons to the chemistry model.*

We first used MLS data for the ozone validation as well as HCl and HNO₃.    Later, we used ozonesondes for

validating ozone, because the data quality of ozonesondes is thought to be better than MLS measurements. However, there are less coincidences and altitude coverages are smaller. Now, we will show both ozonesondes and MLS data for validating FTIR ozone data.

***Specific comments:***

*(4) P2L12: Suggest removing "month" or reword "only in the month of September".*

"month" was removed.

*(5) P2L22-25: The sentence grammar should be revised, e.g. the semi-colon.*

";" was replaced with ":"

*(6) P5L34: The list of tangent point altitudes is incomplete in the middle.*

The list of tangent point altitudes was corrected.

*(7) P6L18: Could you elaborate on "slit function"?*

It is a 5 km-wide running mean. The text was modified.

*(8) P7L10: Were the Livesey results covering the same latitudinal range as this study? Was this bias between MLS and HALOE only seen in HCl?*

Livesey et al. (2013) doesn't show any latitudinal information for the comparison with HALOE. For comparison of each species, Livesey et al. (2013) shows good agreement for $H_2O$ and temperature data between MLS and HALOE.

*(9) P7L10: Livesey et al. (2013) not in references. Couldn't locate the study. Please update.*

We added Livesey et al. (2013) in the references.

*(10) P9L30: Remove "later" after Figure 10. Define EL when mentioning equivalent latitude in the Figure 10 caption since it's used in the figure colorbar label.*

The word "later" was removed. Equivalent latitude was already defined in Section 4.1. We modified the description of EL in Figure 10.

*(11) P10L32: Can the satellite coincidences be filtered to ensure similar equivalent latitude to Syowa Station, in addition to the distance/temporal criteria applied?*

Although we set relatively strict collocation criteria (within 300 km radius and +/- 6 hours) for validation, there is a chance that satellite and station are measuring rather different airmasses when Syowa Station was located near the polar vortex edge. In order to check whether such a situation occurs or not, we looked at equivalent latitudes (EL) of measurement locations of MLS and Syowa Station for the collocated coincidence pairs. The result is shown in Figure A (below). As a result, differences of EL are always within 10 degrees in all cases. We now picked up the collocation pairs whose difference in EL are within +/- 5 degrees. The results are shown in Figures B and C (below). As you can see in these figures, the comparison results show almost similar features. Therefore, we concluded that the difference in EL within 10 degrees does not affect the validation results.

[Figure]

Figure A. Equivalent latitudes (EL) of Syowa Station (black crosses) at 18 km, EL of MLS (red triangle) at 56 hPa, distance between Syowa and MLS measurement locations (black circles), and differences in EL (green crosses) in 2007. Note that the values of differences in EL are multiplied by 10.

[Figure]

Figure B. Absolute differences with all collocation pairs (left columns) and those for differences in EL < 5 degrees (right columns).

Figure C. Percentage differences with all collocation pairs (left columns) and those for differences in EL < 5 degrees (right columns).

*(12) P11L1: It's preferable to spell the term and then put the acronym in brackets, e.g., Reduced Major Axis (RMA)", which has been done elsewhere in this manuscript.*

It was corrected as suggested.

*(13) P13L27-31: Nice result about the ClONO₂ transport.*

Thank you for your comment.   However, reviewer #3 required more discussions on ClONO₂ transport and continuous loss of HCl at the vortex core.   Please refer our replies to reviewer-#3 on this issue.

*(14) P18L29: Misspelled "Livesey" as "Liversey".*

It was corrected to "Livesey".

*(15) Table 3: Suggest adding number of coincidences contributing to the altitudes of interest*

Number of coincidences and typical error values were added in Table 3.

*(16) Fig. 4: Between day 310 and 320, the time spent outside the vortex is a nice illustration of how significant the boundary is for ozone chemistry. Clear support for the criteria used to define these boundaries.*

Thank you for your comment.  Now, some description on this issue is added in Section 4.1

[revised manuscript text omitted]

---

## Author Comment (AC4) · 12 Sep 2019

**Chlorine partitioning near the polar vortex edgeboundary observed with ground-based FTIR and satellites at Syowa Station, Antarctica in 2007 and 2011**

Hideaki Nakajima1,2, Isao Murata2, Yoshihiro Nagahama1, Hideharu Akiyoshi1, Kosuke Saeki2,3, Takeshi Kinase4, Masanori Takeda2, Yoshihiro Tomikawa5,6, Eric Dupuy1, and Nicholas B. Jones7

1National Institute for Environmental Studies, Tsukuba, Ibaraki, 305-8506, Japan
 2Graduate School of Environmental Studies, Tohoku University, Sendai, Miyagi, 980-8572, Japan
 3now at Weathernews Inc., Chiba, 261-0023, Japan
 4Meteorological Research Institute, Tsukuba, Ibaraki, 305-0052, Japan
 5National Institute of Polar Research, Tachikawa, Tokyo, 190-8518, Japan
 6The Graduate University for Advanced Studies, Tachikawa, Tokyo, 190-8518, Japan
 7University of Wollongong, Wollongong, New South Wales, 2522, Australia

Correspondence to: Hideaki Nakajima (nakajima@nies.go.jp)

**Abstract.**

5

- 15 We retrieved lower stratospheric vertical profiles of O3, HNO3, and HCl from solar spectra taken with a ground-based Fourier-Transform infrared spectrometer (FTIR) installed at Syowa Station, Antarctica (69.0°S, 39.6°E) from March to December 2007 and September to November 2011. This was the first continuous measurements of chlorine species throughout the ozone hole period from the ground in Antarctica. We analyzed temporal variation of these species combined with ClO, HCl, and HNO3 data taken with the Aura/MLS (Microwave Limb Sounder) satellite sensor, and ClONO2 data taken with the
- 20 Envisat/MIPAS (The Michelson Interferometer for Passive Atmospheric Sounding) satellite sensor at 18 and 22 km over Syowa Station. HCl and ClONO2 decrease occurred at both 18 and 22 km, and soon ClONO2 was almost depleted in early winter. When the sun returned to Antarctica in spring, enhancement of ClO and gradual O3 destruction were observed. During the ClO enhanced period, negative correlation between ClO and ClONO2 was observed in the time-series of the data at Syowa Station. This negative correlation was associated with the relative distance between Syowa Station and the inner-edge of the
- 25 polar vortex. We used MIROC3.2 Chemistry-Climate Model (CCM) results to see the comprehensive behavior of chlorine and related species inside the polar vortex and the boundaryedge region in more detail. From CCM model results, rapid conversion of chlorine reservoir species (HCl and ClONO2) into Cl2, gradual conversion of Cl2 into Cl2O2, increase of HOCl in winter period, increase of ClO when sunlight became available, and conversion of ClO into HCl, was successfully reproduced. HCl decrease in the winter polar vortex core continued to occur due to either the transport of ClONO2 from the
- 30 subpolar region to higher latitudes, providing a flux of CIONO2 from more sunlit latitudes into the polar vortexor the heterogeneous reaction with HOCI. Temporal variation of chlorine species over Syowa Station was affected by both heterogeneous chemistry related to Polar Stratospheric Cloud (PSC) occurrence deep-inside the polar vortex, and transport of an NOx-rich airmass from lower latitudinal-the polar vortex boundary region which can produce additional CIONO2 by reaction

of ClO with NO2. The deactivation pathways from active chlorine into reservoir species (HCl and/or ClONO2) were confirmed to be highly dependent on the availability of ambient O3. At an altitude where most ozone was depleted in Antarctica (18 km), most ClO was converted to HCl. However, at an altitude where there were some O3 available (22 km), additional increase of ClONO2 from pre-winter value can occur, similar to the case as in the Arctic.

5

**1. Introduction**

Discussion of the detection of "recovery" of the Antarctic ozone hole as the result of chlorofluorocarbon (CFC) regulations has been attracting attention. The occurrence of the Antarctic ozone hole is considered to continue at least until the middle of this century. The world's leading Chemistry-Climate Models (CCMs) indicate that the multi-model mean time series of the

- 10 springtime Antarctic total column ozone will return to 1980 levels shortly after mid-century (about 2060) (WMO, 2019). In fact, the recovery time predicted by CCMs has large uncertainty, and the observed ozone hole magnitude also shows year-to-year variability (e.g., see Figure 4-6 in WMO (2019)). Although Solomon et al. (2016) and de Laat et al. (2017) reported signs of healing in the Antarctic ozone layer only in September-month, there is no statistically conclusive report on the Antarctic ozone hole recovery (Yang et al., 2008; Kuttippurath et al, 2010; WMO, 2019).
- To understand ozone depletion processes in polar regions, understanding of the behavior and partitioning of active chlorines  $(ClO_x=Cl+Cl_2+ClO+ClOO+Cl_2O_2+HOCl+ClNO_2)$  and chlorine reservoirs (HCl and ClONO\_2) are crucial. Recently, the importance of ClONO\_2 was reviewed by von Clarmann and Johansson (2018). Chlorine reservoir is converted to active chlorine that destroys ozone on polar stratospheric clouds (PSCs) and/or cold binary sulphate through heterogeneous reactions:

$$CIONO_2(g) + HCl(s, l) \rightarrow Cl_2(g) + HNO_3$$
 (R1)

20

$$CIONO_2(g) + H_2O(s, l) \rightarrow HOCl(g) + HNO_3$$
(R2)

where g, s, and l represent the gas, solid, and liquid phases, respectively (Solomon et al., 1986; Solomon, 1999; Drdla and Müller, 2012, Wegner et al., 2012; Nakajima et al., 2016).

Heterogeneous reactions :;;

$$N_2O_5(g) + HCl(s, l) \rightarrow ClNO_2(g) + HNO_3$$
 (R3)

30

$$HOCl (g) + HCl (s, l) \rightarrow Cl_2 (g) + H_2O$$
(R4)

are responsible for additional chlorine activation. When solar illumination is available,  $Cl_2$ , HOCl, and  $ClNO_2$  are photolyzed to produce chlorine atoms by reactions:

| $Cl_2 + h\underline{\vee}\underline{\vee} \rightarrow Cl + Cl$ | (R5) |
|----------------------------------------------------------------|------|
| $HOCl + h\underline{\vee} \rightarrow Cl + OH$                 | (R6) |
| $ClNO_2 + h\underline{v}\underline{v} \rightarrow Cl + NO_2.$  | (R7) |

The yielded chlorine atoms then start to destroy ozone catalytically through reactions (Canty et al., 2016):

$$Cl + O_3 \rightarrow ClO + O_2$$
 (R8)

[revised manuscript text omitted]

Figure 7 shows that temporal CIO enhancement and decrease of O3, CIONO2, and HNO3 occurred in early winter (May 30-June 19; day 150-170) at 22 km in 2011. This small ozone depletion event before winter may be due to an airmass movement from the polar night area to a sunlit area at lower latitudes. Table 4 summarized the characteristics of variation of minor atmospheric species for 2007 and 2011 at altitudes of 18 and 22 km.

30

25

**4.2 Time series of ratios of chlorine species**

In order to discuss the temporal variations of the chlorine partitioning, the ratios of observed HCl, ClONO2, and ClO, and Cly with respect to  $Cl_y^*$  were calculated. Hereafter, we will discuss the ratios of chlorine species only for the cases when Syowa Station was located inside the polar vortex. Here, observed  $Cl_y$  is determined as:

**$-Cl_{\star}(FTIR) = HCl(FTIR) + ClONO_2(MIPAS) + ClO(MLS)$ (3)(4)**

**$Cl_{*}(MLS) = HCl(MLS) + ClONO_{2}(MIPAS) + ClO(MLS).$**

Figures 8 and 9 show the time series of the ratios of each chlorine species with respect to  $Cl_v^*$  in 2007 (a) and in 2011 (b) at 18 km and 22 km, respectively. In these plots, HCl data by Aura/MLS were used. Note that light blue in these figures

- 5 shows either ClONO2 or ClO data was missing on that day, while dark blue shows all three data were available on that day. For both in 2007 and 2011 at 18 km (Figure 8), the ratio of HCl was 0.6-0.8 and the ratio of ClONO2 was 0.2-0.3 before winter (May 10-20; day 130-140). The ratio partitioning of HCl to  $Cl_v$  was three times larger than that of ClONO2 at that time. The ratio of ClO increased to ~0.5-0.6 during the ClO enhanced period (the period when ClO values were more than 80 % of its maximum value: August 18-September 17; day230-260. See Table 4). The ratio of HCl was 0-0.2 and the ratio of ClONO2
- 10 was 0-0.6 during this same period. ClONO2 shows negative correlation with ClO, while HCl kept low even when ClO was low during this period. This negative correlation is shown in Figure 10-later. When ClO was enhanced, the  $O_3$  amount gradually decreased, and finally reached <0.5 ppmy (>80% destruction) in October (October 7: day 280) (See Figures 4 and 5). The ratios became 0.9-1.0 for HCl and 0-0.1 for ClONO2 after the recovery in spring (after October 17; day 290), indicating that almost all chlorine reservoir species became HCl via reactions (R13) and/or (R14), due to the lack of  $O_3$  and  $NO_2$  during
- 15 this period. The sum ratios of HCl + ClONO2 + ClO  $Cl_{*}$  (FTIR) and  $Cl_{*}$  (MLS)-were both around 0.5-0.80.7 at the time of CIO enhanced period (August 18 September 17; day 230 260). The remaining chlorine is thought to be either  $Cl_2O_{27}$  or HOCl, which will be shown in model simulation result in Section 4.6. The sum ratio of  $\frac{HCI + CIONO_2CL}{CL}$  became close to 1 after the recovery period (after October 7; day 280).
- For both in 2007 and 2011 at 22 km (Figure 9), the ratio of HCl was 0.84-0.9 and the ratio of ClONO2 was 0.2-0.3 before 20 winter (April 20-May 20; day 110-140). The ratiopartitioning of HCl was three wo to four three times larger than that of ClONO2. The ratio of ClO increased to 0.56-0.7 during the ClO enhanced period (August 8-28; day 220-240 in 2007, August 18-September 7; day 230-250 in 2011). The ratio of HCl was 0-0.23 and the ratio of ClONO2 was 0-0.6 during this period. ClONO2 shows negative correlation with ClO, while HCl kept low even when ClO was low during this period as in the case at 18 km. The  $O_3$  amount gradually decreased during the CIO enhanced period but kept the concentration more than 1.5 ppmv
- 25 (less than half destruction) at this altitude (See Figures 6 and 7). When the CIO enhancement ended, temporal-increase of both ClONO2 and HCl occurred simultaneously up to a ratio of 0.5 occurred-in early spring (September 17-October 7; day 260-280). Then, the reservoir ratios became 0.6-0.78 for HCl and 0.32-0.4 for ClONO2 in spring (after October 7; after day 280). This phenomenon shows that more chlorine deactivation via reaction (R12) occurred towards  $CIONO_2$  at 22 km rather than at 18 km. This is attributed to the existence of  $O_3$  and  $NO_2$  during this period at 22 km, which was different from the case at 18
- 30 km. The sum ratios of HCl + ClONO2 + ClO<del>Cl2 (FTIR) and Cl2 (MLS)</del> were both around 0.7-1.00.8 at the time of ClO enhanced period (August 8-28; day 220-240 in 2007, August 18 September 7; day 230-250 in 2011). The remaining chlorine is thought to be either  $Cl_2O_{27}$  or HOCl. The sum ratio of HCl + ClONO2 + ClOCh2 became around 1.1 after the recovery period (after September 27; day 270). The reason why observed sum ratio  $Cl_v$  values exceed calculated  $Cl_v$  values might be because the  $N_2O-Cl_v$  correlation from the one in the equation (2) is not applicable at this altitude.

In 2011 at 18 km (Figure 8), another temporal increase of ClONO2 up to a ratio of 0.46 occurred in early spring (around October 2-127; day 275-2850) in accordance with HCl increase, then the ClONO2 amount gradually decreased to nearly zero after late October (after October 27; day 300). This temporal increase in ClONO2 could be attributed to temporal change of the location of Syowa Station in the polar vortex. Although Syowa Station was judged to be inside the polar vortex during

- 5 July 14-December 16 (day 195-350) by our analysis, the difference between the equivalent latitude over Syowa Station and that at inner edge became less than 10 degrees at around October 7 (day 280), while it was typically between 15 and 20 degrees in other days. O3 and HNO3 showed higher values around October 7 (day 280) (see Figure 5), indicating that Syowa Station was located close to the boundary region at this period (See Figure A2). Therefore, the temporal increase of ClONO2 in 2011 at 18 km was attributed to spatial variation, not to chemical evolution.
- 10

**4.3 Correlation between ClO and ClONO2**

Figure 10 shows the correlation between CIO and CIONO2 during the CIO enhanced period (August 8-September 17; day 220-260) at 18 km in 2007 (a) and 2011 (b), and at 22 km in 2007 (c) and 2011 (d). In this plot, the location of Syowa Station with respect to the polar vortex (inside, the boundary region, and outside the polar vortex) is indicated by different symbols.

- 20 The negative correlation between CIO and CIONO2 at Syowa Station is explained by the difference in the concentration of CIO, NO2, CIONO2, and HNO3 inside, outside, and at the boundary region of the polar vortex around the station. Outside of the polar vortex, CIO concentration is lower and NO2 concentration is higher than those inside the polar vortex. Inside of the polar vortex, HNO3 is taken into PSCs and removed by the sedimentation of PSCs from the lower stratosphere (denitrification process). Then NOx concentration is low because HNO3 is a reservoir of NOx through the reactions;

 $\underline{\text{NO}_2 + \text{OH} + \text{M} \rightarrow \text{HNO}_3 + \text{M}}$ (R17)

and

**$HNO_3 + hv \rightarrow NO_2 + OH.$ (R18)**

Then NO2 concentration is low and  $ClONO_2$  concentration is also low due to the consumption of  $ClONO_2$  by heterogeneous reaction (R2) inside the polar vortex. In spring, ClO amount gets high due to the activation of chlorine species by reactions

30 (R1~R8) inside the polar vortex. At the boundary region, ClO and NO2 concentrations indicate the value between inside and outside of the polar vortex, that is, ClO concentration is much higher than that outside of the polar vortex and NO2 concentration is much higher than that inside of the polar vortex. Thus, ClONO2 concentration there is elevated in August-September due to the reaction (R12). This cause the The cause of this negative correlation between ClO and ClONO2 might be due to the variation of the relative distance between Syowa Station and the boundary regionedge of the polar vortex. When Syowa

Station was located deep inside the polar vortex, there was more ClO and less ClONO2. On the contrary when Syowa Station was located near the vortex edge, there was less ClO and more ClONO2. The equivalent latitude (EL) over Syowa Station was calculated as described in Appendix B for each correlation point. The EL in each correlation point is now shown by the color code in Figure 10. It generally shows theis tendency, that warm coloured higher equivalent latitude points are located more towards the bottom right-hand side. This is further confirmed by 3-dimensional model simulation as shown later.

4.4 Comparison with model results

5

Figures 11 and 12 show comparisons of daily time series of simulated mixing ratios of ClO, HCl, ClONO2, Cly, and O3 by the MIROC3.2 Chemistry-Climate Model (CCM) (Akiyoshi et al., 2016) with FTIR, Aura/MLS, and Envisat/MIPAS
measurements at 18 km and 22 km, respectively. For a description of the MIROC3.2 CCM, please see Appendix A for detail. In these figures, Cly for Aura/MLS in the panels (d) and (i) actually represents the Cly\* value calculated by equation (2) using the N2O value measured by Aura/MLS. Cly from the MIROC3.2 CCM is the sum of total reactive chlorines, i.e., Cly = Cl + 2\*Cl2 + ClO + 2\*Cl2O2 + OClO + HCl + HOCl + ClONO2 + ClNO2 + BrCl. Note that we plotted modeled values at 12h UTC (~15h local time of Syowa Station) calculated by the MIROC3.2 CCM in order to compare the daytime measurements of FTIR
and satellites. In Figures 11(b), (d), (g), and (i), modeled HCl and Cly showed systematically smaller by 20-40% values compared with FTIR or MLS measurements. The cause of this discrepancy may be partly due to either smaller downward advection and/or faster horizontal mixing of airmass across the subtropical barrier in MIROC3.2 CCM (Akiyoshi et al., 2016). Another possibility of the discrepancy is the difference of Cly\*-N2O correlation used to calculate the Cly\* value by equation

20 are in winter in 2007 and 2011, when projected Cly was -0.7% and -12.8% smaller than in 1997, respectively (Strahan et al., 2014). Nevertheless, evolutions of measured HClO and ClONO2 for the period are well simulated by the MIROC3.2 CCM. Modeled O3 were in very good agreement with FTIR and/or MLS measurements throughout the year in both altitudes for both years. Hereafter, the result of MIROC3.2 CCM at 50 hPa (~18 km) is discussed.

(2), since this correlation comes from the aircraft measurement in summer in 1997 (Bonne et al., 2000), and our observations

**25 4.5 Polar distribution of minor species**

Figure 13 shows distributions of temperature from the model nudged toward the ERA-Interim data, simulated mixing ratios of O3, NO2, HNO3, ClO, HCl, and ClONO2 by the MIROC3.2 CCM at 50 hPa for June 24 (day 175), September 2 (day 245), September 6 (day 249), and October 6 (day 279) in 2007. Polar vortex edgesboundary defined by the method described in Appendix Bmaximum gradient of potential vorticity at 475 K calculated from ERA Interim reanalysis data wereas plotted by

30 whitedotted circles. This boundary was located in between the inner and the outer edges of the polar vortex as were defined in Section 4.1. The location of Syowa Station is shown by a white star in each panel. On June 24 (day 175), stratospheric temperatures over Antarctica were already low enough for the onset of heterogeneous chemistryto allow PSCs to form. Consequently, NO2 was converted into HNO3 via reaction (R17), and HNO3 
[revised manuscript text omitted]
           | 25                                           |                                       | 0-/-0                                                       | 0-/-0                                                                             | <del>1/0</del>                                               | $\frac{1}{1}$                                   |
| April           | <del>1, 3, 4, 5, 8, 24, 26, 28</del>         |                                       | 0-/-0                                                       | 0/0                                                                               | 8-/-0                                                        | 8/0                                             |
| May             | 8, 9, 10, 13, 14, 15, 20, 21, 22             |                                       | 7/0                                                         | 0/0                                                                               | 2/0                                                          | <del>9/0</del>                                  |
| June            |                                              |                                       | 0/0                                                         | 0/0                                                                               | 0./-0                                                        | 0/0                                             |
| July            | <del>29, 30</del>                            |                                       | 0/0                                                         | $\frac{2}{2}$                                                                     | 0/0                                                          | $\frac{2}{0}$                                   |
| August          | <del>1, 8, 9, 10, 24, 25, 26, 28, 29</del>   |                                       | 8/0                                                         | $\frac{1}{1}$                                                                     | 0./-0                                                        | <del>9/0</del>                                  |
| September       | <del>1, 4, 5, 6, 7, 8, 16, 18, 23, 26,</del> | <del>25, 29, 30</del>                 | $\frac{12/3}{3}$                                            | 0/0                                                                               | 0./-0                                                        | <del>12/3</del>                                 |
|                 | 27,30                                        |                                       |                                                             |                                                                                   |                                                              |                                                 |
| October  | <del>6, 10, 11, 14, 19, 20, 25, 26, 27</del> | <del>1, 3, 4, 8, 11, 22, 23,</del>    | 9/9                                                  | 0/0                                                                               | 0./-0                                                        | <del>9/9</del>                                  |
|                 |                                              | <del>24, 26</del>                     |                                                             |                                                                                   |                                                              |                                                 |
| November        | <del>2, 3, 5, 6, 7, 8, 9, 10, 11, 16,</del>  | <del>1, 2, 3, 9, 11, 16, 19</del>     | <del>12/7</del>                                             | $\frac{1}{1}$                                                                     | 4/0                                                          | <del>17/7</del>                                 |
|                 | <del>17, 18, 19, 21, 27, 29, 30</del>        |                                       |                                                             |                                                                                   |                                                              |                                                 |
| December | 4, 7, 8, 9, 13, 15, 16, 17, 20,              |                                       | <del>8 / 0</del>                                            | 0-/-0                                                                             | <del>3/0</del>                                               | $\frac{11}{0}$                                  |
|                 | <del>22, 29</del>                            |                                       |                                                             |                                                                                   |                                                              |                                                 |
| Total           |                                              |                                       | <del>56 / 19</del>                                          | $\frac{4}{0}$                                                                     | <del>18/0</del>                                              | 78/19                                           |

Table 1. FTIR observation dates at Syowa Station in 2007 and 2011

| Species                                                       | $O_3$                                                                             | HNO 3                                                                            | HCI                                                                  |
|---------------------------------------------------------------|-----------------------------------------------------------------------------------|---------------------------------------------------------------------------------------------|----------------------------------------------------------------------|
| Spectroscopy                                                  | HITRAN 2008                                                                       | HITRAN 2008                                                                                 | HITRAN 2008                                                          |
| P ressure and                                          | Daily sonde (0-30 km)                                                             | Daily sonde (0-30 km)                                                                       | Daily sonde (0-30 km)                                                |
| temperature +
Pprofile                              | CIRA 86 (30-100 km)                                                               | CIRA 86 (30-100 km)                                                                         | CIRA 86 (30-100 km)                                                  |
| A priori profiles                                             | Monthly averaged by
ozonesonde (0-30 km)
& ILAS-II (30-100 km)              | Monthly averaged by
ILAS-II                                                              | Monthly averaged by
HALOE                                         |
| Microwindows                                                  | 1002.578 - 1003.500                                                               | 867.000 - 869.591                                                                           | 2727.730 - 2727.830                                                  |
| (cm -1 )                                           | 1003.900 - 1004.400 $872.800 - 874.000$                                           |                                                                                             | 2775.700 - 2775.800                                                  |
|                                                               | 1004.578 - 1005.000                                                               |                                                                                             | 2925.800 - 2926.000                                                  |
| Retrieved
interfering
species                           | O 3 (668), O 3 (686),
CO 2 , H 2 O | H 2 O, OCS, NH 3 , CO 2 ,
C 2 H 6 | CO 2 , H 2 O, O 3 , NO 2 |
| Typical retrieval
error (%) for 15-
25 km | 5                                                                          | 15                                                                                   | 17                                                            |
| Typical vertical
resolution (km)                    | 7                                                                                 | 5                                                                                    | 6                                                             |
| Mean degrees of
freedoms (DOFS)              | 4.9                                                                        | 2.8                                                                                  | 2.3                                                           |

Table 23. Summary of validation results of FTIR profiles compared with ozonesonde and Aura/MLS measurements, and possible Aura/MLS biases from literatures

|                        | Number of
coincidences | Root mean
squares of
official errors*
(%) at 18-22 km | D (%) at 18-
22 km | Min/Max (%) at
18-22 km  | Range of mean
absolute
differences.Agreemen
t at for 15-25 km
(ppmv/ppbv) | Literature values                                                                      |
|------------------------|-----------------------------------------|----------------------------------------------------------------|------------------------------|------------------------------------|---------------------------------------------------------------------------------------|----------------------------------------------------------------------------------------|
| O 3 (sonde) | 14                               | 7.1                                                     | +6.1                         | -10.4/+19.2                        | -0.02~+0.40                                                                    |                                                                                        |
| O 3 (MLS)       | 33                               | 9.4                                                     | -5.5 +6.2             | -10.4 6.3 /+19.2 4.5 | -0. 1302 ~+0. 16 40                                                     | Aura/MLS is +8% higher than
ACE-FTS at 70°S (Froidevoux et al., 2008) |
| HNO 3       | 47                               | 19.2                                                    | +13.2                        | +0.2/+21.9                         | -0.56~+0.57                                                                           | Aura/MLS no bias with errors (0.6 ppbv) (Livesey et al., 2011)                         |
| HCl                    | 50                               | 39.5                                                    | -9.7                         | -14.6/-3.0                         | -0.2 0 ~+0.09                                                                  | Aura/MLS > HALOE by 10-
15%, precision 0.2-0.6 ppbv
(Livesey et al., 2013)       |

\*Root mean squares of official absolute and relative errors given by each data set.

| Month            | Dates                                    | Dates                          | Number of days             | Number of days                     | Number of days              | Number of                  |
|------------------|------------------------------------------|--------------------------------|----------------------------|------------------------------------|-----------------------------|----------------------------|
|                  | (2007)                            | (2011)                  | inside the polar
vortex | in the boundary
region of the   | outside the polar
vortex | measurement
days |
|                  |                                          |                                | (2007/2011)         | polar vortex
(2007/2011) | (2007/2011)          | (2007/2011)         |
| March            | 25                                |                                | 0 / 0               | 0/0                                | 1 / 0                | 1 / 0               |
| April            | 1, 3, 4, 5, 8, 24, 26, 28         |                                | 0 / 0               | 0 / 0                       | 8 / 0                | 8 / 0               |
| May              | 8, 9, 10, 13, 14, 15, 20, 21, 22         |                                | 7 / 0               | 0 / 0                       | 2 / 0                | 9 / 0               |
| June             |                                          |                                | 0 / 0               | 0 / 0                       | 0 / 0                | 0 / 0               |
| July             | 29, 30                            |                                | 0 / 0               | 2/0                                | 0 / 0                | 2 / 0               |
| August           | 1, 8, 9, 10, 24, 25, 26, 28, 29   |                                | 8 / 0               | 1 / 0                       | 0 / 0                | 9 / 0               |
| September | 1, 4, 5, 6, 7, 8, 16, 18, 23, 26, | 25, 29, 30              | 12/3                | 0/0                                | 0 / 0                | 12/3                |
|                  | 27,30                                    |                                |                            |                                    |                             |                            |
| October   | 6, 10, 11, 14, 19, 20, 25, 26, 27 | 1, 3, 4, 8, 11, 22, 23, | 9/9                 | 0/0                                | 0 / 0                       | 9 / 9               |
|                  |                                          | 24, 26                  |                            |                                    |                             |                            |
| November         | 2, 3, 5, 6, 7, 8, 9, 10, 11, 16,  | 1, 2, 3, 9, 11, 16, 19  | 12 / 7              | 1/0                         | 4 / 0                | 17 / 7              |
|                  | 17, 18, 19, 21, 27, 29, 30        |                                |                            |                                    |                             |                            |
| December         | 4, 7, 8, 9, 13, 15, 16, 17, 20,          |                                | 8 / 0               | 0/0                                | 3 / 0                | 11 / 0              |
|                  | 22, 29                            |                                |                            |                                    |                             |                            |
| Total            |                                          |                                | 56 / 19             | 4/0                                | 18 / 0               | 78 / 19             |

**Table A1. FTIR observation dates at Syowa Station in 2007 and 2011**

**Table 4. Summary of minor atmospheric species variations**

| Altitude                                              | <del>18</del>      | km                 | 22                 | <del>22 km</del>   |  |
|-------------------------------------------------------|--------------------|--------------------|--------------------|--------------------|--|
| Year                                                  | 2007               | 2011               | 2007               | 2011               |  |
| ClO enhanced period (day)                             | <del>230-260</del> | <del>230-260</del> | 220-240            | 230-250            |  |
| Variation when ClO enhanced (ppbv)                    | 0-1.3              | 0-1.5              | 0-2.2              | <del>0-2.2</del>   |  |
| HCl value before winter (ppbv)                        | 1.5-1.8            | <del>1.2–1.6</del> | 2.1-2.4            | 1.8-2.2            |  |
| HCl starting-ending day of decrease (day)             | <del>140-180</del> | <del>140-180</del> | <del>130-180</del> | <del>140-170</del> |  |
| Variation when HCl~0 (ppbv)                           | 0-0.3              | 0-0.3              | 0.1-1.0            | <del>0.1–0.9</del> |  |
| HCl starting-ending day of increase (day)             | <del>250-300</del> | <del>250-300</del> | 240-280            | 240-300            |  |
| HCl Value after increase (ppbv)                       | <del>2.6-3.0</del> | <del>2.5–2.8</del> | 2.1-2.4            | <del>2.0-2.5</del> |  |
| HCl Value outside polar vortex (ppbv)                 | 1.5-2.0            | <del>1.0–1.8</del> | 1.5-2.0            | <del>1.5–2.0</del> |  |
| ClONO 2 Value before winter (ppbv)         | ~ <del>0.5</del>   | ~0.4               | <del>0.6 0.9</del> | <del>0.6-0.7</del> |  |
| Variation when CIONO2~0 (ppbv)                        | <del>0-1.5</del>   | 0-1.5              | 0-2.0              | 0-2.0              |  |
| Day of ClONO2 enhancement                             | -                  | <del>270-300</del> | <del>270-280</del> | <del>270–280</del> |  |
| Value of ClONO2 enhancement (ppbv)                    | -                  | <del>1.5</del>     | <del>1.5</del>     | <del>1.5</del>     |  |
| ClONO2 value after enhancement (ppbv)                 | 0-0.3              | 0-0.2              | <del>0.8–1.3</del> | 0.8-1.1            |  |
| CIONO 2 value outside polar vortex (ppbv)  | 0.3-0.4            | 0.2-0.3            | 0.5-0.7            | <del>0.6–0.8</del> |  |
| O3-value before winter (ppmv)                         | 2.5                | 2.5                | 4 <del>.0</del>    | 4 <del>.0</del>    |  |
| O 2 -starting-ending day of decrease (day) | <del>190-280</del> | 200-270            | <del>170-260</del> | <del>170-270</del> |  |
| O 2 -minimum value (ppmv)                  | <del>0.3</del>     | <del>0.5</del>     | <del>2.0</del>     | <del>1.0</del>     |  |
| O 2 value after recovery (ppmv)            | <del>0.8</del>     | <del>0.8</del>     | 2.4-4.0            | <del>2.0-3.5</del> |  |
| HNO3 value before winter (ppbv)                       | <del>6-10</del>    | <del>8-10</del>    | <del>15-16</del>   | <del>13-15</del>   |  |
| HNO2 starting-ending day of decrease (day)            | <del>160-190</del> | <del>150-180</del> | 140-180            | <del>150-180</del> |  |
| HNO3-minimum value (ppbv)                             | θ                  | θ                  | 2                  | 1                  |  |
| HNO3 value after recovery (ppbv)                      | 3-4                | 3-4                | 4-6                | 4 5         |  |

\* 'ClO enhanced period' is defined as the period when ClO values were more than 80 % of its maximum value.

Figures

Figure 1. Averaging kernel functions of the SFIT2 retrievals for (a)  $O_3$  (b)  $HNO_3$  (b), and (c) HCl (c).

- 5 Figure 2. (a) Mean aAbsolute (a) and (b) mean relativepercentage (b) differences of O3 profiles retrieved from FTIR measurements minusand those from ozonesonde measurements. Horizontal bars indicate the standard deviation of differences at each altitude. (c) Mean aAbsolute (c) and (d) mean relative (d) differences of O3 profiles retrieved from FTIR measurements minusand those from Aura/MLS measurements. (e) Mean aAbsolute (c) and (f) mean relative percentage (fd) differences of HNO3 profiles retrieved from FTIR measurements minusand those from FTIR measurements minusand those from FTIR measurements minusand those from Aura/MLS measurements. (g) Mean aAbsolute (ge)
- 10 and (h) mean relativepercentage (hf) differences of HCl profiles retrieved from FTIR measurements minusand those from Aura/MLS measurements. Horizontal bars indicate the root mean squares standard deviation of differences at each altitude. Horizontal dashed bars indicate the altitude range of our focus (15-25 km).
- Figure 3. Time series of (a) daytime hour, temperatures at 18 km in (ba) 2007 and (cb) 2011, and at 22 km in (de) 2007 and
  (ed) 2011 over Syowa Station using ERA-Interim data. Approximate saturation temperatures for nitric acid trihydrate PSC (TNAT) and ice PSC (TICE) calculated by assuming 6 ppbv HNO3 and 4.5 ppmv H2O are also plotted in the figures by dotted lines. Dates when PSCs were observed over Syowa Station are indicated by asterisks on the bottom of the figures.
- Figure 4. Time series of (a) HCl, ClONO2, ClO, Cly\*, (b) O3, and HNO3 mixing ratios at 18 km in 2007 over Syowa Station.
  O3(FTIR), HCl(FTIR), and HNO3(FTIR) were measured by FTIR at Syowa Station, while HCl(MLS), ClO(MLS), and HNO3(MLS) were measured by Aura/MLS. O3(sonde) was measured by ozonesonde. ClONO2 was measured by Envisat/MIPAS. Cly\* is calculated from Aura/MLS N2O value. See text in detail. The unit of O3 is ppmv and the other gases are ppbv. The dark shaded area, the light shaded area, and the white area indicate the days when Syowa Station was located outside, in the boundary region, and inside the polar vortex, respectively.

25

Figure 5. Same as Figure 4 but in 2011.

Figure 6. Same as Figure 4 but at 22 km.

30 Figure 7. Same as Figure 5 but at 22 km.

Figure 8. Time series of the ratios of HCl (dark blue or light blue),  $CIONO_{27}$  (yellow), and CIO (red), and  $Cl_{37}$  (HCl+ClONO\_2+CIO) to total chlorine (Cly\*) over Syowa Station at 18 km in (a) 2007 and in (b) 2011. Light blue shows

either ClONO2 or ClO data was missing on that day, while dark blue shows all three data were available on that day. Shaded areas are the same as Figure 4.

Figure 9. Same as Figure 8 but at 22 km.

5

10

15

Figure 10. Scatter plot between CIO (Aura/MLS) and CIONO2 (Envisat/MIPAS) mixing ratios between August 8 and September 17 (day 220 - 260) at 18 km and 22 km in 2007 and 2011. Crosses, triangles, and squares represent the data when Syowa Station was located inside the polar vortex, the boundary region, and outside the polar vortex, respectively. Solid lines are regression lines obtained by RMA-(Reduced Major Axis) regression. Color represents the equivalent latitude over Syowa Station on that day. Circles with crosses represent the days which are shown in Figures 13 and 14.

Figure 11. Daily time series of measured and modeled minor species over Syowa Station at 18 km. Black diamonds are data by FTIR, red squares are by Aura/MLS and Envisat/MIPAS, blue triangles are data by MIROC3.2 CCM. (a) is for ClO, (b) is for HCl, (c) is for ClONO2, (d) is for Cly, and (e) is for O3 in 2007. (f) is for ClO, (g) is for HCl, (h) is for ClONO2, (i) is for Cly, and (j) is for O3 in 2011.

Figure 12. Same as Figure 11 but for 22 km.

25 Figure 14. Same as Figure 13 but for July 5 (day 186), August 19 (day 231), August 21 (day 233), and October 9 (day 282), 2011. Polar vortex edges on July 5 plotted by dotted while circles indicate that the inner vortex edge was defined on this day.

Figure 15. Three-hourly zonal-mean time series of MIROC3.2 CCM outputs for (a) ClO+2\*Cl2O2+2\*Cl2, (b) Cl2O2, (c) Cl2, (d) ClO, (e) HOCl, (f) HCl, and (g4) ClONO2 during day number 120–300 at 50 hPa in 2007.

30

Figure A1. Time-equivalent latitude sections of MPV (contours) and its gradient with respect to EL (colors) at (a) 450 K and (b) 560 K isentropic PT surfaces in 2007. Black dots represent the inner and outer edge(s) of the polar vortex. Red dots represent the EL of Syowa Station on each day.

Figure A2. Same as Figure A1 but for the year in 2011.

---

## Author Comment (AC6) · 12 Sep 2019

**Reply to reviewer #1**

We thank anonymous reviewer #1 for his/her constructive review that would improve the contents of our paper. The review comments by anonymous reviewer #1 are numbered and repeated below as *in italic letters*, followed by our answers. In the new draft with corrections (supplement file), red, purple, and blue corrections are the revisions suggested by reviewers #1, #2, and #3, respectively. Yellow-marked sentences were also added in response to Short Comment #1 by Dr. Adrian Tuck.

***GENERAL COMMENTS***

*(1) More context for the FTIR data should be given. A few questions come to mind.*
*- Is this the first time that the Syowa FTIR data has been retrieved/written up? Or are there other papers that describe this instrument and the data?*

Yes, this is the first time that the Syowa FTIR data were retrieved and submitted to a journal paper.

*(2) - Why is there only 2007 and 2011 data? Was the instrument not deployed in other years?*

The FTIR was installed at Syowa Station in March 2007, and we made one year of operation in 2007. After that time, there were no one who could operate the FTIR at Syowa Station. In 2011, another FTIR operator wintered at Syowa Station, and made one year of FTIR operation. Then, after a few more operation in 2016, the FTIR was brought back to Japan in 2017. This sequence was now apparently described in Section 2.1.

*(3) - Are the data publicly available?*

Yes, we put the Syowa FTIR data in our data repository of our institute and put DOI information for them. The MIROC3.2 CCM results are stored at the CCMI site of BADC. This information was described in the "Data availability" now.

*(4) - How to these data fit in with other ground-based FTIR datasets collected at high latitudes? Is the data quality similar? Have other FTIR instruments been used to study trace gases during spring-time ozone depletion?*

As far as we know, there have been only two other FTIR instruments in Antarctica. One is at South Pole Station (90°S), and the other is at New Zealand's Scott Station in Antarctica. We added description related to South Pole FTIR in Section 2.1. The data quality should be similar. However, due to the location of Scott Station (78°S), they cannot measure solar infrared spectra until September when ozone hole is already in progress. I don't know any publications which focused spring-time ozone depletion using the other stations'

FTIR data so far. Therefore, the Syowa Station's FTIR has an advantage to study ozone-hole-related issues.

*(5) The validation sections should be more specific in terms of how validation methods are applied (see specific comments for details). It would also be useful to answer:*
*- What is the expected uncertainty in the FTIR measurements? Is this something you retrieve?*

We added typical retrieval error of each species in old Table 2 (new Table 1).

*(6) - How do your validation results compare with validation of other ground-based FTIR instruments? Is the instrument at Syowa performing similarly to other FTIR?*

By comparing our results with other FTIR validation results by Schneider et al. (2008), they look very similar. We described this issue in Section 3.

*(7) - Are there any other factors that could affect comparison results? Do any of these species vary diurnally? Is there a chance that the satellite and station are measuring very difference air masses (e.g., inside/outside the vortex) for some coincidences? If either of these are factors, how does this affect interpretation of the data in Sect. 4?*

Among the chemical species we analyzed, only ClO has diurnal variation. Therefore, we used only daytime ClO data for the analysis. Although we set relatively strict collocation criteria (within 300 km radius and +/- 6 hours) for validation, there is a chance that satellite and station are measuring rather different airmasses when Syowa Station was located near the polar vortex edge. In order to check whether such a situation occurs or not, we looked at equivalent latitudes (EL) of measurement locations of MLS and Syowa Station for the collocated coincidence pairs. The result is shown in Figure A (below). As a result, differences of EL are always within 10 degrees in all cases. We now picked up the collocation pairs whose difference in EL are within +/- 5 degrees. The results are shown in Figures B and C (below). As you can see in these figures, the comparison results show almost similar features. Therefore, we concluded that the difference in EL within 10 degrees does not affect the validation results.

[Figure]

Figure A. Equivalent latitudes (EL) of Syowa Station (black crosses) at 18 km, EL of MLS (red triangle) at 56 hPa, distance between Syowa and MLS measurement locations (black circles), and differences in EL (green crosses) in 2007. Note that the values of differences in EL are multiplied by 10.

[Figure]

Figure B. Absolute differences with all collocation pairs (left columns) and those for differences in EL < 5 degrees (right columns).

Figure C. Percentage differences with all collocation pairs (left columns) and those for differences in EL < 5 degrees (right columns).

*(8) The discussion of results is repetitive and at times is difficult to follow. There are many timeseries figures (Fig. 4-9; Fig. 11-12). I found it hard to track through these and determine what exactly the authors were trying to highlight. I also had a hard time connecting all of these pieces to the concluding arguments around transport, chlorine deactivation, etc. A few possible solutions:*
*- Determine what you are trying to show, perhaps starting from the statements made in the conclusion. Only include figures/discussion that are relevant to what you're trying to show. Move additional timeseries figures into an appendix or supplement. Some tables (Table 1, Table 4) could also be moved out of the main body of the paper.*

We modified several descriptions in Sections 4. Table 1 is now moved to Appendix, and Table 4 was deleted from the draft.

*(9) - For Fig. 8-9 is there a more concise way of showing the relationships between the chlorine species than using the timeseries (e.g., through scatter plots or something else)?*

We modified Figures 8 and 9. Now, partitioning of chlorine species are expressed by bar graphs instead of scatter plots in order to gain visibility. Also, descriptions of these figures were modified accordingly in Section 4.2.

*(10) - Can the comparison with the modelling data be merged in with the validation of the FTIR (Sect. 3)? Can the modelling timeseries figures be folded into the timeseries figures presented in previous sections?*

We showed modelling data in order to show other chlorine species which cannot be measured by FTIR and/or satellites, not for validating our measurements. Model data validation is out of the focus of this paper. Therefore, the modelling data section is kept as it is.

*(11) I found it hard to follow the discussion around the mechanisms behind the decrease of HCl. It would be helpful if the discussion was expanded, to help answer the following:*
*- What hypotheses are there for the decrease in HCl in the literature? Is there just Solomon et al. (2015) and Grooss et al. (2018)? Are these hypotheses in conflict with each other or is it possible that they both contribute to the decrease in HCl together?*

Also, there was several comments by reviewer-#3 on the decrease of HCl. The discussion on the decrease of HCl is much expanded. In the current modified manuscript, we showed two hypotheses in parallel; one is the $ClONO_2$ transport from the sunlit lower latitude towards the core of the vortex based on Solomon et al. (2015), and the other is the heterogeneous reaction between HCl and HOCl based on Müller et al. (2018). Please see the revised Section 4.6.

*(12) - For the existing hypotheses, what data/evidence were used to develop the hypotheses? How does the data that you collected add to the existing supporting data/evidence?*

The main data we used are based on our MIROC3.2 CCM results.   However, our measurements of ClO, HCl, and $ClONO_2$ gains insights for these hypotheses.

*(13) - Why does the sporadic increases in $ClONO_2$ in the model data support the transport mechanism? Do you have any other data to support that $ClONO_2$ is being transported (e.g., maps of $ClONO_2$, tracers showing transport patterns, evidence in the satellite data)? Are there alternative mechanisms that could explain your observations, such as chemistry?*

Although there was no direct evidence of $ClONO_2$ transport, if $ClONO_2$-rich airmass was transported from the vortex boundary towards the vortex core region, it could decrease HCl by heterogeneous reaction (R1). Another alternative mechanism to decrease HCl was the heterogeneous reaction with HOCl (R4), which is now shown by another increase pattern on July 7 (day 188) and July 20 (day 201) in Figure 15(e).

*(14) - Can your data be used to refute any other hypotheses for decreasing HCl?*

Our data are consistent with both $ClONO_2$ transport hypothesis and reaction with HOCl hypothesis.

***SPECIFIC COMMENTS:***

*(15) - Page 1, Line 17: You state "This was the first continuous measurements of chlorine species throughout the ozone hole period from the ground in Antarctica" here and elsewhere in the paper. This statement is a bit vague. Have other studies looked at any chlorine species from the ground in Antarctica? Are they looking at fewer species? Or for shorter time periods? It would be helpful if you included a literature review of ground-based measurements (FTIR and maybe other measurements of chlorine, such as OClO from UV-vis?) in Antarctica in your introduction.*

We now described history of several ground-based observations except for FTIR (UV-visible and microwave) related to ozone chemistry in the last paragraph of Section 1, and the history of Antarctic ground-based FTIR measurements in the first paragraph of Section 2.1.   Other FTIR stations in Antarctica can measure ozone-related species only after September, when ozone hole was at the recovery phase.   This was now described in Section 2.1.

*(16) - Page 4, Line 25: Are there other ground-based FTIRs in Antarctica?*

Yes, there were two other ground-based FTIRs in Antarctica in history; one at South Pole Station, and the other

at McMurdo/Scott Station, which is now described in Section 2.1. We think the South Pole FTIR is not in operation now.

*(17) - Page 5, Line 1: What is the temporal resolution of the FTIR at Syowa? Does it only take measurements during sunny days? Does it require manual operation?*

One measurement takes about 10 minutes and we can make observations only on sunny days. It requires manual operation. This was explained in Section 2.1 now.

*(18) - Page 5, Line 18: Is there a way of using the averaging kernel to determine whether the 18 km and 22 km concentrations are independent from each other?*

The concentrations of 18 km and 22 km are expected to be almost independent when you look at the averaging kernel in Figure 1 and the degree of freedoms (DOFS) in new Table 1.

*(19) - Page 5, Line 25: State where various satellite datasets were obtained from? (E.g., URL?)*

We now showed the URLs of satellite dataset pages.

*(20) - Page 5, Line 24: The application of selection criteria should be clarified, perhaps with a full paragraph at the end of this section. Was this same criteria applied to all three satellite instruments (MLS, MIPAS, and CALIPSO)? Was the same criteria used for both the validation (Sect. 3) and the discussions of results over Syowa (Sect. 4)? E.g., for the timeseries figures was the 6 h criterion applied?*

The selection criteria for MLS and MIPAS are already stated in the draft (within 300 km radius from Syowa Station and within +/-6 hours of the FTIR measurement). The same criteria were used both for the validation and the results. For the CALIPSO data, we selected the closest orbital data of the day, whose maximum distance to Syowa Station is 320 km. This is now stated in the draft.

*(21) - Page 5, Line 30: Have you considered applying a selection criteria based on inside/outside the polar vortex to the satellite data along their line of sight instead of using values over the station for all measurements?*

No. As is stated in (old draft's) page 6, line-16, we collocated the locations of tangent height of 20 km for the direction of the sun from Syowa Station at the time of the FTIR measurement, and the tangent point at 20 km for the satellites' data.

*(22) - Page 6, Line 18: What is the shape of the 5 km-wide slit function? Is this based on the FTIR resolution? Why was this used instead of the averaging kernel? Also, why are the MLS data smoothed? Is the vertical*

*resolution for MLS much higher than the ground-based FTIR?*

It is a 5 km-wide running mean.    The description was reworded.    5 km is the typical vertical resolution of FTIR measurements.    Since the vertical resolution of satellite measurements are a bit better than FTIR resolution (3-4 km for MLS, and 3 km for MIPAS), smoothing of ozonesondes and satellites' data were performed in order to merge the vertical resolution of each measurement.

*(23) - Page 6, Line 25: State the expected uncertainty and any expected biases in ozonesonde measurements*

Typical precision and accuracy of the ECC-type ozone sondes are considered to be $\pm(3\text{-}5)\%$ and $\pm(4\text{-}5)\%$, respectively (Komhyr, 1986).    This is now described in the second paragraph of Section 3.

*(24) - Page 6, Line 24: I'm a bit confused about the language used throughout this section. The figures show mean relative difference and mean absolute difference versus altitude. Replace "The absolute difference. . ." with "The mean absolute difference. . ." and 「The mean relative difference" in this line.*

We now added the word "mean" at several places in Section 3.    Also, "mean absolute difference" was used instead of "agreement" at several places for clarity.

*(25) - Page 6, Line 25: What do you mean by "mean relative difference" here? Is this the mean relative differences averaged again over an altitude range? It might be simpler just to state the mean relative difference values at 18 km and 22 km instead of taking another average over altitude.*

First, altitude range is different (15-25 km) and (18-22 km).    Second, the "averaged" value in 18-22 km range was calculated.    We rewrote "the average of mean relative differences ..." here.

*(26) - Page 6, Line 26: What to you mean by "within error bars"? Are you referring to the standard deviation in the mean differences? This isn't really an error – it's the variability in the comparisons. It would make more sense to determine whether the agreement is as expected based on estimated uncertainty in the FTIR measurements (if this exists) and known uncertainties/biases in the satellite data. Also, why is the standard deviation used for the comparisons instead of the standard error?*

The typical errors in FTIR measurements are shown in (new) Table 1 (old Table 2).    We added some more official error values of satellite measurements in Section 3.    We now used standard root mean square errors of FTIR and validation data instead of standard deviation, and rewrote to: "FTIR data agree with validation data within root mean squares of typical errors in FTIR and validation data at the altitude of interest."    Also, the figure caption in Figure 2 had a mistake.    We now rewrote to: "Horizontal bars indicate the root mean squares of differences at each altitude."    Root mean squares of official errors in FTIR and satellite measurements are now added in (new) Table 2 (old Table 3).

*(27) - Page 6, Line 31 – Page 7, Line 2: See comments for previous paragraph.*

Same as above.

*(28) - Page 7, Line 10: "with a precision of 0.2-0.6%" – is this the precision in the MLS measurements? Or is this the precision of the systematic bias?*

It was a precision in the MLS measurements, but not 0.2-0.6%, but 0.2-0.6 ppbv, which corresponds to 10-30 % (See Table 3.9.1 in Livesey et al., 2013). We corrected the description here.

*(29) - Page 8, Line 6: You describe a set of steps for detecting the inner and outer edges of the vortex. Is this a new method? Is this expected to work better than other established methods for some cases?*

When we looked at the actual variation of the isentropic potential vorticity gradient with respect to equivalent latitude, we often saw apparent double-peak structure of polar vortex boundaries at 500-600 K in the case of Antarctic winter (see Figure 5 in Tomikawa et al. (2015)). Therefore, we decided to determine the inner and outer edges of the vortex when there were double peak structures. Actually, this method seems to work better with interpreting the observed data related to polar chlorine chemistry, rather than the conventional single vortex boundary categorization (e.g., Nash et al. 1996). This description was now added in the text.

*(30) - Page 8, Lines 11-14: Please break this down into smaller steps – I have a hard time understand what was done. My best guess is that you calculated the isentropic potential vorticity gradient as a function of equivalent latitude. You found the local maxima. You defined the inner/outer edges of the vortex at the local maxima that both (?) exceeded the wind-speed threshold and were at least 5 in equivalent latitude apart? Are there always only two local maxima that meet these criteria?*

Your understanding is correct. When there was only one local maxima, the regions are categorized into only two categories (inside and outside the polar vortex, i.e., no boundary region). In order to describe the definition of polar vortex boundaries in more detail, the polar vortex boundary definition part was now moved to Appendix B, and new Figures A1 and A2 were added.

*(31) - Page 8, Line 18: Did you filter the MLS data according to an error threshold? Or did you just remove suspect data when you looked at the timeseries? All filters applied to the satellite data (for, e.g., uncertainty, etc) should be described explicitly in Sect. 2.2.*

The lack of ClO and HCl data from day 195 to 219, 2007 is due to extremely large error values in the MLS data products (~1.2 ppbv error compared with ~0.2 ppbv error in other periods). The lack of ClONO$_2$ data from day 170 to 216, 2007 are negative values (-999.9) in MIPAS data products. This was now described in

the text.

*(32) - Page 11, Lines 10-24: Can the model comparisons be merged into Sect. 3? It would be nice if consistent comparison methods were applied. Similar to Sect. 3 – are the comparisons consistent with what is expected based on known model performance and known biases in the satellite instruments?*

We showed modelling data in order to show other chlorine species which cannot be measured by FTIR and/or satellites, not for validating our measurements.   Model data validation is out of the focus of this paper. Therefore, the modelling data section is kept as it is.

*(33) - Page 11, Line 28: Why was a different method used to determine the polar vortex than at Syowa station?*

We now unified the method of polar vortex definition at Syowa Station.   The definition method of polar vortex boundaries is now moved to Appendix B.   The description of polar vortex boundary in Section 4.5 was modified.

*(34) - Page 11, Line 31: What do you mean by "This boundary was located in between the inner and the outer edge of the polar vortex as were defined in Sect. 4.1"? Did you compare the two definitions of the vortex? If so, was this just done at Syowa station for all data or a subset of data? Or is it the case that the max gradient of PV at 475 K is always between the inner/outer edge values by definition?*

Now the definitions of polar vortex boundary were unified.

*(35) - Table 4: Should describe how you came up with each of the various ranges presented in the table. Here are a few examples of questions that should be answered. How do you define the threshold for a ClONO2 enhancement or ozone starting-ending day of decrease? What are given in "HCl Value after increase" ranges? Is this the min/max of individual FTIR measurements over a given time-period? For the ClO enhancement period, is the 80% of maximum value different in 2007 and 2011 or is a single value used? What does "Variation when HCl 0 ppbv" mean?*

We thought this table can be used by readers to see typical values of each species during the development and recovery of ozone holes in 2007 and 2011.   However, it is rather arbitrary to pick up each value.   Moreover, the contents of this table are not discussed in the paper.   Therefore, we decided to omit this table from our paper.

*(36) - Figure 3: Can you add a panel describing hours of sunlight or SZA here or to one of your other timeseries? It would be helpful to visualize this through discussions around available sunlight.*

We added a panel for showing daytime hours in Figure 3(a).

*(37) - Figure 3: Are the gaps in the PSC timeseries because CALIPSO observed no PSCs or because CALIPSO didn't collect any coincident data during this time? Either describe in text and/or add marker for CALIPSO measurements which did not observe PSCs.*

The gaps in the PSC timeseries are the days when CALIPSO observed no PSCs over Syowa Station. We looked at the closest CALIPSO orbit to Syowa Station in each day. This daily sampling is now described in the last paragraph of Section 2.2.

*(38) - Figures 14/15: Why have you used model data instead of satellite data for the parameters that are available from satellite? Did you check to see if the satellite saw the same patterns as the model?*

The comparison between satellite observations and model results were shown in Figures 11 and 12, although comparison or validation of model results with actual measurements are not the main issue of this paper. In Figures 14 and 15, we focused on MIROC3.2 model results, in order to keep the consistency of each chlorine species.

**MINOR/TECHNICAL COMMENTS:**

*(39) - Throughout the text, there are some awkwardly worded sentences and minor grammatical mistakes. I assume that these would be corrected through copy-editing, so have not listed these here, unless the meaning is unclear.*

We think grammatical mistakes will be corrected through copy-editing.

*(40) - Table 2: Define "PT"*

We reworded to "Pressure and temperature profiles".

*(41) - Table 3: Should define the fields included in the table. Is D(%) 18-22 km the average of the mean relative differences? Is (Min/Max %) 18-22 km the max/min mean relative difference across the various altitudes? (Might be better just to show mean relative differences +/- standard error at 18 km and at 22 km.)*

Yes, your understanding is correct. We modified (new) Table 2 (old Table 3) to explain the validation results more comprehensively. The numbers in this table are now explained in the text in Section 3 as well.

*(42) - Table 3: See comment re: precision in Page 7, Line 10*

It was modified in the (new) Table 2 (old Table 3).

*(43) - Table 3: Replace "Agreement" with "Range of mean absolute differences for 15-25 km"*

It was replaced as suggested.

*(44) - Figure 2, caption: Replace "Absolute (a) and percentage" with "Mean absolute (a) and mean percentage"*

They were replaced as suggested.

*(45) - Figure 2, caption: Replace "FTIR measurements and those from ozonesonde" with "FTIR measurements minus those from ozonesonde"*

They were replaced as suggested.

*(46) - Figure 2, caption: Move "Horizontal bars indicate the standard deviation of differences at each altitude." to the end of the caption, since it applies to all panels.*

It was moved as suggested.

*(47) - Figure 2, caption: Describe what the horizontal dashed lines indicate.*

They indicate the altitude range of our focus (15-25 km). We also described it at the last paragraph of Section 2.1.

*(48) - Figure 4, caption: Replace "from $N_2O$ value" with "from Aura/MLS $N_2O$".*

It was replaced as suggested.

[revised manuscript text omitted]

---

## Author Comment (AC7) · 12 Sep 2019

**Reply to reviewer #3**

We thank anonymous reviewer #3 very much for his/her detailed and constructive review that would improve the contents of our paper. The review comments by anonymous reviewer #3 are numbered and repeated below as *in italic letters*, followed by our answers. In the new draft with corrections (supplement file), red, purple, and blue corrections are the revisions suggested by reviewers #1, #2, and #3, respectively. Yellow-marked sentences were also added in response to Short Comment #1 by Dr. Adrian Tuck.

**General**

*(1) It is now more than three decades ago that the Antarctic ozone hole was discovered (WMO, 2019); by now the processes involved in its formation are thought to be understood in some detail. Current state-of-the-art models reproduce the observed springtime ozone loss in the polar stratosphere with good accuracy (e.g., Khosrawi et al., 2009; Chipperfield et al., 2017; WMO, 2019; Froidevaux et al., 2019). Nonetheless, there are open questions in Antarctic chlorine and ozone chemistry. Such questions can be addressed when new observations, such as those reported in the manuscript, become available. Insofar, this manuscript is an important contribution to ACP.*

Thank you for your comment.

*(2) I suggest an extension of the discussion in the manuscript regarding several issues (see in particular detailed comments below). Briefly, the Cly correlation needs to be adjusted to 2007 and 2011 (or the adjustment made should be described, see below) and the model behaviour reported by Grooß et al. (2018) that I think is also found here (namely that the models show HCl remaining in the core of the vortex) should be discussed in terms of MIROC3.2.*

We made some modifications in the text on Cly correlation in 2007 and 2011, considering the differences in available total Cly on those years. The remaining HCl issue in the core of the vortex is now further discussed in the paper.

*(3) Further, the manuscript could make a better contribution to addressing the issue of a "race" between chlorine activation and deactivation (Solomon et al., 2015; Müller et al., 2018; Zafar et al., 2018) and the question of which HCl formation processes are responsible for the observed HCl increase at the end of the ozone hole period.*

We further discussed the issue of a "race" between chlorine activation and deactivation by referring the paper you mentioned.

*(4) Overall, the FTIR measurements presented here are certainly of great scientific interest and the measurements are combined in a meaningful way with satellite information. Moreover, a model simulation is included that helps the interpretation of the measurements. Notwithstanding these points, I suggest a substantial extension of the discussions on Antarctic chlorine chemistry in the paper. Provided the (in my view) necessary extensions are done, I would expect that this paper would make a great contribution to ACP.*

We extended the discussion on Antarctic chlorine chemistry in the paper as is described below.

**Comments in Detail**

*(5) Year-to-year variability of Cly*
  *The main driver for antarctic ozone loss is the available Cly but there is a substantial year-to-year variability in this quantity; this issue is discussed by Strahan et al. (2014). How are the years discussed here (2007, 2011) ranked in the observed variability of Cly (Strahan et al., 2014)? Further, the applicability of the employed empirical relation for Cly to the years discussed here (2007, 2011) needs to be addressed in the paper.*

According to Strahan et al. (2014), 2007 was the year when there were about +4.3% more $Cl_y$ (2.88/2.76 ppbv), and 2011 was the year when there were about -5.2% less $Cl_y$ (2.53/2.67 ppbv) than the projected $Cl_y$ value. This was now described at the end of first paragraph in Section 2.1.

*(6) HCl remains in the core of the polar vortex in austral winter*
  *The authors find here for the MIROC3.2 CCM that some HCl remains in winter in the core of the polar vortex in darkness. Such a model behaviour is expected as there is not enough $ClONO_2$ and no light in the core of the vortex. This model behaviour was reported by Grooß et al. (2018) for three models. Further they showed that this model feature is not found in observations of HCl. I suggest stating that the MIROC3.2 CCM shows the same issue – provided that the authors agree. If they do not agree, there should be a discussion of the issue in the paper.*

We do not agree the reviewer's opinion that MIROC3.2 CCM shows the same issue.   In the MIROC3.2 simulation, the amount of HCl became nearly zero even in the core of the polar vortex in winter (please see HCl value at 87.9°S in Figure 15).   We added discussion on this issue in Section 4.6 by adding new possible cause of the HCl loss by HOCl.   The MIROC3.2 HOCl result is now added in Figure 15 (e).

*(7) Overestimated transport across the vortex edge in models*
  *The paper states that transport across the vortex edge might be overestimated in models. I think here is the potential for the paper to make an important contribution. The issue that models might overestimate mixing into the vortex edge has been discussed (e.g.) by Hoppe et al. (2014); indeed for the same Eulerian transport scheme as employed in MIROC3.2. Observations of $N_2O$ (and the $N_2O$ gradient might help to elucidate the model issue. In any case a bit more discussion on this point is warranted.*

We looked at the differences between observed and modeled $N_2O$ gradients at the vortex edge, but could not find apparent differences between them. The coarse resolution of MIROC3.2 CCM (T42) may be the cause of overestimation of $ClONO_2$ transport across the polar vortex.

*(8) Continuous loss of HCl in the core of the polar vortex*

 *The authors state the following conclusion in the paper: "Continuous loss of HCl was seen at 87.9 S between days 160 and 200 even after the disappearance of the counterpart of heterogeneous reaction (R1) (Figure 15(e)). The cause of this continuous loss was unknown until recently, where a hypothesis was proposed that includes the effect of decomposition of particulate $HNO_3$ by some process like ionisation caused by galactic cosmic rays during the winter polar vortex (Grooß et al., 2018). Solomon et al. (2015) proposed a new mechanism on this issue: Continuous transport of $ClONO_2$ from the subpolar regions near 55-65 S to higher latitudes near 65-75 S provides a flux of NOx from more sunlit latitudes into the polar vortex. Our result is consistent with the mechanism indicated by some sporadic increase in $ClONO_2$ at around days 158, 179, and 189 at 76.7 S as shown in Figure 15(f)".*

 *First, the processes described by Grooß et al. (2018) and Solomon et al. (2015) are very different: Grooß et al. (2018) describe a polar night process, whereas Solomon et al. (2015) describe a dynamical process (acting later in the course of the existence of the polar vortex), which needs light nonetheless as formation of ClONO2 is involved.*

 *Further note that Grooß et al. (2018) discussed the transport mechanism and concluded that it cannot explain the so-called 'HCl-discrepancy'. Of course the present manuscript might come to a different conclusion but I think a more extensive evaluation of the arguments put forward by Grooß et al. (2018) is necessary here. In this context, Fig. 15 of the manuscript could be important; could it be that the 'discrepancy' reported by Grooß et al. (2018) is also noticeable in panel (e) of Fig. 15?*

 *Second, an alternative explanation for the continuing decline of HCl could be the formation of HOCl under sunlit conditions which would allow HCl to decline to zero values (Grooß et al., 2011; Müller et al., 2018). If the authors cannot develop a preference for one of these mechanisms based on their data/observations, I suggest to state both alternatives in the discussion in the paper.*

 *This issue reflects the importance of the reaction $HCl + HOCl \rightarrow Cl_2 + H_2O$ (R4); early papers (Prather, 1992; Crutzen et al., 1992) have pointed to the importance of this reaction for bringing down the HCl concentrations in the polar vortex in Antarctica.*

 *Finally, the transport of ClONO2 from the vortex edge to the vortex core does not occur in isolation; inspection of Fig. 15 would suggest to me that mixing from the vortex edge would transport an amount of x ppb of $ClONO_2$ but at the same time an amount of x ppb HCl as well. Thus there is no net removal of HCl by "mixing". As stated above –the easiest solution would be to discuss both alternative explanations. Further, these alternative are not necessarily contradictory; both mechanisms could contribute partly to the observed HCl decline.*

We agree to the reviewer that the processes described by Grooß et al. (2018) and Solomon et al. (2015) are

very different, and transport mechanism cannot fully explain the so-called 'HCl-discrepancy'. We also checked our MIROC3.2 CCM model result on HOCl and found that HOCl certainly plays an important role in winter HCl continuous loss. Therefore, we added new column showing HOCl amount by MIROC3.2 CCM in Figure 15 (e) and added discussion on the HOCl issue in Section 4.6. However, we still believe that the transport of $ClONO_2$ from the vortex edge toward inside the vortex would partly explain the continuous loss of HCl in the polar vortex in early winter period (June-July), because there are much more $ClONO_2$ available at the edge region of the polar vortex, while there are almost no HCl at this place (please see HCl and $ClONO_2$ values at the edge of polar vortex on June 24, 2007 in Figure 13).

*(9) "Race" between chlorine activation and deactivation*

*An aspect of polar ozone and chlorine chemistry, where different concepts are discussed in the literature is the maintenance of enhanced levels of active chlorine during the time period (September and early October) when rapid ozone loss occurs. One concept is the one of a "race" between chlorine activation and deactivation, i.e., a competition of the heterogeneous reactions R1, R2, and R4 and gas-phase reformation of HCl and $ClONO_2$ (R12, R13) (Solomon et al., 2015). The other concept is the one of so-called "HCl null-cycles", where the formation of HCl (R13) is followed by immediate reactivation of HCl (Müller et al., 2018; Zafar et al., 2018). (See also the discussion in WMO, 2019). The measurements presented in this manuscript might help to shed some light on these issues – this could be a contribution of this paper. Alternatively, if the measurements presented here cannot contribute to discriminating between the two discussed processes, this could also be a result of the paper (which should be mentioned).*

We believe that our MIROC3.2 CCM model results also supports the "HCl null-cycles" by our HCl and HOCl results shown in Figures 15(d) and 15(e). We added description on "race" between chlorine activation and deactivation at the end of Section 4.6.

*(10) Formation of HCl – Antarctic deactivation*

*The presented observations demonstrate that the deactivation in the Antarctic is through formation of HCl and that the deactivation is rapid. This is good and important. However, it is also stated that the formation of HCl is via the reaction $CH_4 + Cl \rightarrow HCl + CH_3$ (R13). I agree that this HCl formation is "common wisdom" (e.g. Crutzen et al., 1992; Douglass et al., 1995). However the authors have also mentioned the reaction $CH_2O + Cl \rightarrow HCl + CHO$ (R14). What is the evidence that the observed formation of HCl is indeed via R13? Further, it should be taken into account that there is also reactivation of chlorine even if reactions R14 and R13 occur at a considerable rate (Müller et al., 2018; Zafar et al., 2018). Again, the paper could make a contribution here, but at least there should be more discussion of all the processes playing a role here.*

At the beginning, we did not notice the importance of reaction (R14) in our draft. Now both reactions (R13) and/or (R14) are mentioned for the possible mechanisms of chlorine deactivation into HCl.

*(11) Negative correlation between ClO and $ClONO_2$*

*The cause of the negative correlation between ClO and ClONO₂ (e.g. Fig. 10) is discussed as being caused by the distance of Syowa station relative to the vortex. I do not think that this is entirely correct. The main reason for the negative correlation is that ClO is converted into ClONO₂ and vice versa. (However the rather large scatter in Fig. 10 should also be acknowledged.) Given this fact, the higher values of ClONO₂ seem to preferentially occur closer to the vortex edge (so that the location of Syowa indeed is relevant).*

We agree to the reviewer that the negative correlation is because of the conversion between ClO and $ClONO_2$, and one way ($ClO + NO_2 + M \rightarrow ClONO_2 + M$) reaction is limited due to the loss of $NO_2$ by denitrification by PSCs inside the polar vortex.    We explained this issue in Section 4.3 by adding new reactions (R17) and (R18).

*(12) Near zero values of ozone*

*The observations of ozone reported here (ozone sonde measurements) show very low values in October. Such near zero values of ozone have been reported before (Solomon et al., 2005); are the ozone values reported here compatible with the reported low ozone values? Perhaps the ozone values could be replotted on a log-scale as is the earlier publication (Solomon et al., 2005) – perhaps it is too much to show such plots in the paper, but an electronic supplement might be an alternative.*

Yes, the near zero ozone values in October at 18 km in 2007 and 2011 are typical in these years at Syowa Station.    We think that the current linear scale for the ozone values (Figures 4 and 5) are better to show temporal variation of ozone throughout the winter-spring than log-scale.    Even Solomon et al. (2005) used linear scale in Figure 2.

*(13) FTIR measurements*

*The FTIR measurements are a major contribution of this paper. This is why I suggest to make the data available for other researchers as well. Further Toon and Farmer (1989) have reported measurements of the HOCl integrated vertical column abundance, which was inferred from high resolution infrared solar spectra measured by the JPL MkIV interferometer from the NASA DC-8 aircraft during flights over Antarctica in September 1987. Would the current set-up also allow measurements of HOCl? Even quantifying an upper limit might be helpful. Other species of potential interest would be methanol or formaldehyde.*

We are working to put our FTIR data (in hdf format) in the data repository of our institute and put DOI number for it.    Hopefully, it will be realized within this September.

Thank you for the information of HOCl paper by Toon and Farmer (1989).    We looked at the spectra taken at Syowa Station for the period when the most HOCl amount is expected.    However, it is very hard to distinguish the spectral feature of HOCl in those spectra, because of the weak absorption by HOCl and small amount of HOCl in the atmosphere.    We may work to retrieve HOCl from Syowa FTIR data by co-adding several spectra and reducing noise in future work.    It requires a bit too much effort for the current work.

*(14) Model description*

*The results of the MIROC3.2 model make an important contribution to the study. However, the model documentation is not sufficient. There is a short paragraph (on p. 15) and the reference to Akiyoshi et al. (2016). But even after consulting these pieces of information, many aspects of the model remain unclear. Which photolysis scheme is used for the calculations presented here; is the scheme using spherical geometry? I think a reference to the employed scheme would be appropriate. Which solver is employed for solving the set of differential equations that result from the considered chemical scheme? How exactly (which surfaces?) is heterogeneous chemistry (including particle formation) treated in the model?*

*Further, it is not clear which reactions (and which species) have been considered in the presented model calculations. But this aspect could be important. I think it would be very helpful (and very easy) to add the information in question (e.g. add a list of reactions as an electronic appendix) to the paper.*

A more detailed description of MIROC3.2 is now described in Appendix in addition to some more citations.

*(15) Data Availability*

*There is no data availability statement in this paper. According to the rules of ACP such a statement should be added to the final version of the paper. This point would both regard all observations (including FTIR, ozone sondes and the satellite information shown in the plots of the paper) and the model results. It is up to ACP, but I believe that making the data (in particularly the unique FTIR data) available would enhance the impact of this paper.*

We are working to put our FTIR data (in hdf format) in the data repository of our institute and put DOI number for it. Hopefully, it will be realized within this September. Also, MIROC3.2 CCM model results can be obtained from the CCMI site. The doi information was now described in the "Data availability:" statement.

***Details***

*(16) p. 2, l. 10: "and the observed"*

It was corrected as suggested.

*(17) p. 2, l. 11: add a citation for the observed ozone hole magnitude*

We added Figure 4-6 in WMO (2019) for citation.

*(18) p. 3, l. 5: citations for these reactions? In particularly, R14 is often mentioned regarding chlorine deactivation.*

We added Grooβ et al. (2011) and Müller et al. (2018) for the citations.

*(19) p 3, l 8: R14 is not often mentioned as a HCl forming reaction; suggest adding a reference. Perhaps also for R 12 and R 13.*

We added Grooβ et al. (2011) and Müller et al. (2018) for the citations.

*(20) p. 3, l 21: in all years?*

It was corrected as suggested.

*(21) p. 3, l. 28: "sometimes" is not right, is it? It happens in the Arctic always if chlorine activation occurs, I'd argue. Probably the first observations of this phenomenon were reported by von Clarmann et al. (1993); Oelhaf et al. (1994).*

We deleted "sometimes" here.    von Clarmann et al. (1993), Muller et al. (1994), and Oelhaf et al. (1994) are already listed here.

*(22) p 4, l. 5: citation for the FTIR measurement? How much can we learn about the vertical resolution from Fig. 1?*

Rinsland et al. (1988) was added as a citation of FTIR measurement.    The typical vertical resolution was added in Table 2.

*(23) p 4, l. 22: Farman et al. (1985) did not show ozone sonde measurements. By the way, another ozone sonde measurement was conducted in 1985 by Gernandt (1987).*

We modified the description at Halley Bay.    Since this part describes the first ozone hole measurements in Antarctica, we did not mention Gernandt (1987).

*(24) p. 5, l 1: I think a further advantage is also the location inside and outside of the vortex core and the inner vortex transport barrier.*

We added description on the advantage of Syowa Station here.

*(25) p. 7, l. 19: 4.5 ppm is not the best value if dehydration occurs.*

This is the typical value before the occurrence of dehydration.    We used this value to show the initial condition of PSC occurrence at the beginning of the winter.

*(26) p. 7, l. 30: have these ozone sondes been compared in sonde comparison studies?*

Yes, they participated in the Juelich Ozone Sonde Intercomparison Experiment (JOSIE)-2000. We added citation of JOSIE-2000 (Smit and Straeter, 2004) in the citation.

*(27) p. 8, l. 4: Such an empirical relation is not valid for arbitrary years: it should be explained how the adjustment to the conditions of 2007 and 2001 has been done (it is necessary to both correct N2O and Cly).*

We agree with the reviewer that this empirical relation should be modified for 2007 and 2011 by the change of $Cl_y$ values. We added description on this issue in the text.

*(28) p 8., l. 14: the issue of a transport barrier within the Antarctic polar vortex was also discussed by Lee et al. (2001). The transport barrier within the Antarctic polar vortex in the early vortex can also be seen in ILAS measurements (Tilmes et al., 2006).*

The transport barrier issue is now added in the text, in addition to the citations (Lee et al., 2001; Tilmes et al., 2006).

*(29) p 8., l. 28: Within the vortex (but at the boundary) much lower HCl would be expected than outside of the vortex. Is this not seen by the FTIR measurements?*

The HCl value at the boundary region are in between the outside and inside the polar vortex. The gradual change of HCl was observed. We modified the text in this part.

*(30) p. 9, l. 3: what is the uncertainty range of the FTIR measurements? (report a ± here).*

Typical errors of the FTIR measurements are now shown in new Table 1 (old Table 2).

*(31) p. 9, l. 21: inside, but in the core or at the edge?*

We defined three categories; inside, boundary region, and outside the polar vortex. The data we showed for Figures 8 and 9 are for the first category (inside the polar vortex) data.

*(32) p. 9., l. 26: "partitioning" is not really clear; I believe you mean HCl/Cly.*

We rewrote to "ratio of HCl to $Cl_y$*".

*(33) p. 9, l. 33: probably not via R13, see above.*

We rewrote to "via reactions (R13) and/or (R14)"

*(34) p. 11, l. 17: "systematically smaller": could you quantify this statement?*

We rewrote to "systematically smaller by 20-40% compared with ..."

*(35) p. 11, l. 19: faster mixing in the model than in the real world might indeed be an issue (see above).*

In this part, smaller downward advection and/or faster horizontal mixing are proposed for the explanation of discrepancies in HCl and $Cl_y$ between the model and the observations.

*(36) p. 11, l. 21: for which year is the correlation – the correlation needs to be adjusted to the years in question here (2007 and 2011).*

The effect of different $Cl_y$ values between the correlation year (1997) and our observations (2007 and 2011) is now described in the text.

*(37) p. 11., l. 32: Note that the onset of heterogeneous chemistry occurs very likely before PSCs form and that PSCs do not form at NAT equilibrium. You could formulate: temperatures low enough for the onset of heterogeneous chemistry.*

We rewrote as suggested.

*(38) p. 11., l. 32: $NO_2$ does not condense (it needs to be chemically converted first).*

We rewrote to "$NO_2$ was converted into $HNO_3$ via reaction (R17), and $HNO_3$ ..."

*(39) p 12, l. 2: "Some HCl remains" this is expected in the model as there is not enough $ClONO_2$ (as stated here) and no light in the core of the vortex. This model behaviour was also shown by Grooß et al. (2018) and I suggest stating that the MIROC3.2 CCM shows the same issue (see also above).*

We added the statement "as was also shown by CLaMS model simulation by Grooß et al. (2018)" here.

*(40) p. 12, l. 16: Note that HCl remains low even under these conditions, when a relatively fast rate of reaction R13 should occur in the stratosphere. HCl-null cycles (Müller et al., 2018) could be an explanation.*

We added explanation of HCl-null cycles (Müller et al., 2018) here.

*(41) p 12, l. 21: How sure can we be that the recovery is via R13??*

We rewrote to "reactions (R13) and/or (R14)".

*(42) p. 13., l. 10: compare Grooß et al. (2018)*

We cannot compare our Figure 15 result with Grooß et al. (2018), because Figure 15 deals with the year in 2007, while Grooß et al. (2018) deals with the year in 2011.

*(43) p 13, l. 22: Again, how sure can we be that the recovery is via R13?? What about R14? What about other possible HCl forming reactions?*

We rewrote to "reactions (R13) and/or (R14)".

*(44) p 13, l. 25 and below: The processes described by Grooß et al. (2018) and Solomon et al. (2015) should be distinguished: Grooß et al. (2018) describe a polar night process, whereas Solomon et al. (2015) describe a dynamical process which needs light nonetheless as formation of ClONO$_2$ is involved.*

We now distinguished processes by Grooß et al. (2018) and Solomon et al. (2015) and rewrote this part.    Also, a process with HOCl was now described.    Also, see our answer to your comment (8).

*(45) p. 14, l. 9: could you quantify "well below"*

We rewrote to "fell ~4K below ...".    We also added this explanation in the first paragraph of Section 4.1

*(46) p. 14., l. 13: the reason for the negative correlation is the conversion of ClO to ClONO$_2$ (see above).*

Even the conversion of ClO to ClONO$_2$ occurs, it is related to the availability of NO$_2$, which is proportional to the relative distance to the vortex edge.    We added the word "relative" here and in the abstract.

*(47) p. 14, l. 21: As discussed above, it is not sure that the transport of ClONO$_2$ is the only possible explanation for the behaviour of HCl.*

We added the possibility of the heterogenous reaction with HOCl here and in the abstract.

*(48) p. 14., l. 23: NOx rich would also mean rich in HCl – correct?*

No, not always.    NOx rich airmass can produce ClONO$_2$ by reaction (R12), but not always rich in HCl.

*(49) p 15., l. 14: Sander et al 2010 or 2011? (see reference list).*

It was Sander et al. (2011), not (2010).    Thank you for your comment.

*(50) p. 16., l. 20: Stimpfle*

It was corrected as suggested.

*(51) p. 17, l. 29: 1977?*

It was 1997, not 1977.

*(52) p. 20, l. 10: Karin Labitzke*

It was corrected as suggested.

*(53) p 22, l. 2: activation (no hyphen)*

Hyphen was deleted.

*(54) p. 24, 25: Could you add the information on the vertical resolution of the FTIR measurements to one of these tables?*

Vertical resolutions of the FTIR measurements were added in Table 2.

*(55) p. 39: pane -> panel*

It was corrected as suggested.

[revised manuscript text omitted]

---

## Referee Report (RR1)

**Second review of "Chlorine partitioning near the polar vortex ..."**

BY NAKAJIMA ET AL.

**General**

In response to the three reviews, the authors have substantially changed and improved their manuscript. I think that the discussion has improved in many parts of the paper and also the description of the FTIR data is better in the revised version. I think that in particularly adding information on HOCl to the manuscript constitutes a significant improvement of the paper.

However, there are still some issues where I think the paper needs improvement. In my first review I had the point that "the initial Cly correlation needs to be adjusted to 2007 and 2011 – or the adjustment made should be described". I think this aspect is not completely resolved (see below). Further I stated that "the model behaviour ('HCl discrepancy') reported by Grooß et al. (2018) should be discussed in terms of MIROC3.2". Here there still seems to be discrepancy in opinion, which will require some further discussion (see below). I would be optimistic, however, that this problem can be resolved. Finally, I think the model issue of the transport barrier at the vortex edge needs some attention.

Overall, clearly, the new FTIR measurements presented here are of great scientific interest and are combined in a meaningful way with satellite measurements and model (MIROC3.2) results. In spite of the fact that I recommend a further revision of the paper; I would be optimistic that such a revised version would be further improved and would be close to being accepted by ACP.

**Comments in Detail**

**Year-to-year variability and temporal trend of Cly**

As stated in the first review, the main driver for Antarctic ozone loss is the available Cly. The authors now discuss the year-to-year variability in this quantity (Strahan et al., 2014), which is good. However, I think the point of the applicability of the employed empirical relation for Cly (Eq. 2) to the years discussed here (2007, 2011) needs to be addressed in the paper better than in the present draft (see also details below).

There is a statement in the paper that observed Cly in 2011 (2.53 ppbv) was about -5.2% less than the projected Cly from Newman et al. (2007) (2.76 ppbv for 2007 and 2.67 ppbv for 2011). It is not clear to me that the projected values by Newman et al. (2007) were indeed used in the analysis here. Moreover, there is also an alternative projection of Cly values (Engel et al., 2018; WMO, 2019).

**HCl remains in the core of the polar vortex in austral winter**

In my first review it was stated: "The authors find here for the MIROC3.2 CCM that some HCl remains in winter in the core of the polar vortex in darkness. Such a model behaviour is expected as there is not enough $ClONO_2$ and no light in the core of the vortex. This model behaviour was reported by Grooß et al. (2018) for three models. Further they showed that this model feature is not found in observations of HCl. I suggest stating that the MIROC3.2 CCM shows the same issue – provided that the authors agree."

In their reply, the authors state that they "do not agree with the reviewer's opinion that MIROC3.2 CCM shows the same issue". They have added information and discussion on the role of HOCl, which I consider an important improvement. However, if I look at the HCl values at 87.9° in figure 15, I see that HCl remains constant (and above 0.5 ppb) for days 140-160 (this airmass is in darkness as can be seen from $Cl_2$, panel c). Isn't this the issue discussed by Grooß et al. (2018)? HCl is only close to zero at day $\approx 200$ (I would argue due to HOx production in sunlight); after this day HOCl starts increasing (see also below).

If the authors really do not agree that they see the "HCl discrepancy", this is only possible if they have a process in the model which removes the remaining HCl in darkness. Which process could there be in the model? Does the model contain some process producing NOx in polar night? The latter point would be very interesting. Another possibility would be that MIROC3.2 does not see this issue because of a too coarse spatial resolution, but this would be a model artefact.

**Overestimated transport across the vortex edge in models**

The paper originally stated that transport across the vortex edge might be overestimated in the MIROC3.2 model results. This is perhaps no surprise given the relatively coarse (T42) horizontal spatial resolution. I the reply, the authors now state that they "looked at the differences between observed and modeled $N_2O$ gradients at the vortex edge, but could not find apparent differences between them". Thus they do not seem to find the same result as Hoppe et al. (2014) who found that models might overestimate mixing across the vortex edge.

Such a point should not be covered only in passing in the reply. I suggest to include the relevant plots in the reply (which will remain accessible at ACP) or, even better, include this information as an appendix to the paper.

**Continuous loss of HCl in the core of the polar vortex**

In response to the review comments, the authors have substantially revised the discussion on continuous loss of HCl in the core of the polar vortex. Nonetheless, I would like to come back on a few points of which I think that they could be better represented

in the manuscript.

Note that the HCl loss processes described by Grooß et al. (2018) and Solomon et al. (2015) are very different: Grooß et al. (2018) describe a *polar night* process, whereas Solomon et al. (2015) describe a dynamical process (acting later in the course of the existence of the polar vortex). The dynamical process suggested by Solomon et al. (2015) requires light as formation of $ClONO_2$ is involved. This is also true for the explanation for the continuing decline of HCl under sunlit conditions through the formation of HOCl (Grooß et al., 2011; Müller et al., 2018).

Finally, $ClONO_2$ is not enhanced at the vortex edge in June or May (see Figs. 13 and 14). Also transport of $ClONO_2$ from the vortex edge to the vortex core does not occur in isolation but in mid-winter likely mixes in air with enhanced HCl and ozone (see below and first review). A very different issue again is the mixing of 'out-of-vortex' air; first, this is less likely and, second, it would certainly bring in non-activated air with substantial concentrations of HCl.

**Model description**

In response to my first review, the authors have substantially extended the model description. However, the detailed description of the heterogeneous reactions is in Japanese and the paper still does not contain a list of reactions (e.g. add a list of reactions as an electronic appendix). Perhaps a bit more could be done regarding model description. As discussed below, some model behaviour raises the questions, exactly which reactions are taken into account in the model formulation.

**Data Availability**

A data availability statement has now been added to this paper. I think in particular making the FTIR data available is very helpful. You might want to add statements regarding MLS, MIPAS and MIROC3.2.

**Some details regarding the revised version**

- p 1, l 21: $ClONO_2$ was almost depleted; but what about HCl?

- p 1, l 25 "comprehensive behaviour" is unclear, what is meant here?

- p. 3, l 6: perhaps you want to add a reference here for R 12 also? Santee et al. (2008) would be a possibility.

- p. 3, l 22: replace 'formation' by 'existence'

- p. 8, l 31: This relation (Eq. 2) is only true for conditions of 1997. Assuming that $N_2O$ is constant, the Cly calculated from this equation is only valid for 1997.

Correct? So one needs to take into account the temporal trend of Cly between 1997 and 2007 or 2011. In reality there are also temporal changes of $N_2O$ that should be taken into account. Only thereafter the adjustments suggested by Strahan et al. (2014) should be taken into account.

- p 9, l 5: state here to which figure the 'light shaded region' etc belongs.

- p 9, l 16: "was occurred" $\longrightarrow$ "occurred"

- p 10, l 20: 'ratio of HCl' etc – this is unclear. I think you mean mixing ratio here? Or HCl/Cly ratio? This should be clarified. (similar in l. 29).

- p. 11, l 32: taken *up* by PSCs (or similar)

- p 12, l 2 (R18): there is also the reaction $HNO_3 + OH \longrightarrow NO_3 + H_2O$

- p 12, l. 27: this is indeed not unlikely, however the change of stratospheric Cly between 1997 and 2007 as well as between 1997 and 2011 can be quantified and should not be ignored.

- 13, l 8: note that enhanced $ClONO_2$ is *not* seen in June!

- p 13, l 11: not only CLaMS but also SD-WACCM and TOMCAT/SLIMCAT

- p 14, l 21: "the decrease of HCl stopped ..." – I think the authors have an important point here. But is this not the model behaviour reported by Grooß et al. (2018)? And a model behaviour which is *not* in agreement with HCl observations. I suggest more discussion here.

- p. 15, l 2: note that there is also HOCl production in the gas-phase ($ClO + HO_2$; the details might depend on the $HO_2$ production in the photolysis of $CH_2O$, see e.g. Müller et al. (2018). I would be interesting to analyse here in detail the assumptions in the MIROC3.2 model.

- p. 15., l 7: I think cycles C3 and C4 are not defined in the manuscript. I suggest to include a brief explanation here not just a reference.

- p. 15, l 19: For this interpretation, it is important to take into account that the poleward transport of $ClONO_2$ by mixing (this is the point here if I understand the text correctly) does not occur in isolation. The impact of such mixing depends on the time period considered. In deep winter mixing would not transport only $ClONO_2$, but also HCl and ozone. On the other hand, in June (Fig. 13) there is no enhancement of $ClONO_2$ at the vortex edge, so what could be the effect of mixing of vortex edge air in June?

- p 16, l 10: PSCs do not *form* at NAT saturation temperature a stated in this sentence. In general; I suggest using the same wording throughout the paper for this issue. Be clear if you mean onset of heterogeneous chemistry, PSC formation/existence etc. For example it is NAT saturation temperature, not NAT PSC saturation temperature.

- p 16, l 15: 'comprehensive behaviour'?

- Fig 13: It is difficult to see two white circles here – I can see only one.

- Fig 14: It is difficult to see two white circles here for August and October – I can see only one. The last sentence of the caption is difficult to understand. The dotted line for 5 July 2011 is confusing – is it correct? It looks rather different than the various chemical species. Is the point here that on 5 July 2011 only an inner vortex edge is defined but no actual vortex edge?

**References**

Engel, A., Bönisch, H., Ostermöller, J., Chipperfield, M. P., Dhomse, S., and Jöckel, P.: A refined method for calculating equivalent effective stratospheric chlorine, Atmos. Chem. Phys., 18, 601–619, https://doi.org/10.5194/acp-18-601-2018, 2018.

Grooß, J.-U., Brautzsch, K., Pommrich, R., Solomon, S., and Müller, R.: Stratospheric ozone chemistry in the Antarctic: What controls the lowest values that can be reached and their recovery?, Atmos. Chem. Phys., 11, 12 217–12 226, 2011.

Grooß, J.-U., Müller, R., Spang, R., Tritscher, I., Wegner, T., Chipperfield, M. P., Feng, W., Kinnison, D. E., and Madronich, S.: On the discrepancy of HCl processing in the core of the wintertime polar vortices, Atmos. Chem. Phys., pp. 8647–8666, https://doi.org/10.5194/acp-18-8647-2018, 2018.

Hoppe, C. M., Hoffmann, L., Konopka, P., Grooß, J.-U., Ploeger, F., Günther, G., Jöckel, P., and Müller, R.: The implementation of the CLaMS Lagrangian transport core into the chemistry climate model EMAC 2.40.1: application on age of air and transport of long-lived trace species, Geosci. Model Dev., 7, 2639–2651, https://doi.org/10.5194/gmd-7-2639-2014, URL http://www.geosci-model-dev.net/7/2639/2014/, 2014.

Müller, R., Grooß, J.-U., Zafar, A. M., Robrecht, S., and Lehmann, R.: The maintenance of elevated active chlorine levels in the Antarctic lower stratosphere through HCl null cycles, Atmos. Chem. Phys., 18, 2985–2997, https://doi.org/10.5194/acp-18-2985-2018, URL https://www.atmos-chem-phys.net/18/2985/2018/, 2018.

Newman, P. A., Daniel, J. S., Waugh, D. W., and Nash, E. R.: A new formulation of equivalent effective stratospheric chlorine (EESC), Atmos. Chem. Phys., 7, 4537–4552, 2007.

Santee, M. L., MacKenzie, I. A., Manney, G. L., Chipperfield, M. P., Bernath, P. F., Walker, K. A., Boone, C. D., Froidevaux, L., Livesey, N. J., and Waters, J. W.: A study of stratospheric chlorine partitioning based on new satellite measurements and modeling, J. Geophys. Res., 113, D12307, https://doi.org/10.1029/2007JD009057, 2008.

Solomon, S., Kinnison, D., Bandoro, J., and Garcia, R.: Simulation of polar ozone depletion: An update, J. Geophys. Res., 120, 7958–7974, https://doi.org/10.1002/2015JD023365, 2015.

Strahan, S. E., Douglass, A. R., Newman, P. A., and Steenrod, S. D.: Inorganic chlorine variability in the Antarctic vortex and implications for ozone recovery, J. Geophys. Res., https://doi.org/10.1002/2014JD022295, URL `http://dx.doi.org/10.1002/2014JD022295`, 2014.

WMO: Scientific assessment of ozone depletion: 2018, Global Ozone Research and Monitoring Project–Report No. 58, Geneva, Switzerland, 2019.

---

## Author Response (AR2)

**Reply to reviewer #3**

We thank anonymous reviewer #3 very much for his/her detailed and constructive review that would improve the contents of our paper. The review comments by anonymous reviewer #3 are numbered and repeated below as *in italic letters*, followed by our answers. In the new draft with track changes (supplement file), red corrections are the revisions suggested by reviewer #3.

**General**

*(1) In response to the three reviews, the authors have substantially changed and improved their manuscript. I think that the discussion has improved in many parts of the paper and also the description of the FTIR data is better in the revised version. I think that in particularly adding information on HOCl to the manuscript constitutes a significant improvement of the paper.*

Thank you for your comment.

*(2) However, there are still some issues where I think the paper needs improvement. In my first review I had the point that "the initial Cly correlation needs to be adjusted to 2007 and 2011 – or the adjustment made should be described". I think this aspect is not completely resolved (see below). Further I stated that "the model behaviour ('HCl discrepancy') reported by Grooß et al. (2018) should be discussed in terms of MIROC3.2". Here there still seems to be discrepancy in opinion, which will require some further discussion (see below). I would be optimistic, however, that this problem can be resolved. Finally, I think the model issue of the transport barrier at the vortex edge needs some attention.*

We will try to answer your questions in the following answers.

*(3) Overall, clearly, the new FTIR measurements presented here are of great scientific interest and are combined in a meaningful way with satellite measurements and model (MIROC3.2) results. In spite of the fact that I recommend a further revision of the paper; I would be optimistic that such a revised version would be further improved and would be close to being accepted by ACP.*

Thank you for your comments.

**Comments in Detail**

**(5) Year-to-year variability and temporal trend of Cly**
*As stated in the first review, the main driver for Antarctic ozone loss is the available Cly. The authors now*

*discuss the year-to-year variability in this quantity (Strahan et al., 2014), which is good. However, I think the point of the applicability of the employed empirical relation for Cly (Eq. 2) to the years discussed here (2007, 2011) needs to be addressed in the paper better than in the present draft (see also details below).*

*There is a statement in the paper that observed Cly in 2011 (2.53 ppbv) was about -5.2% less than the projected Cly from Newman et al. (2007) (2.76 ppbv for 2007 and 2.67 ppbv for 2011). It is not clear to me that the projected values by Newman et al. (2007) were indeed used in the analysis here. Moreover, there is also an alternative projection of Cly values (Engel et al., 2018; WMO, 2019).*

We agree to the reviewer that the year-to-year variability and temporal trend of $Cl_y$ and $N_2O$ should not be ignored to derive the empirical correlation between $N_2O$ and $Cl_y$* using Equation (2).   Therefore, we now consider the value for $Cl_y$ and $N_2O$ in each year (1997, 2007, and 2011) to derive $Cl_y$ via Eq. (2).   The Newman's calculated $Cl_y$ values for 1997, 2007, 2011 are; 2.90, 2.76, and 2.67 ppbv, respectively by Strahan et al. (2014), and measured $N_2O$ values for those years are; 313, 321, and 324 ppbv, respectively by WMO (2011); WMO (2014).   We have now considered these differences to derive $Cl_y$* from MLS $N_2O$ values using Eq. (2).   The results of $Cl_y$*, chlorine related ratios are all changed, and Figures 4, 5, 6, 7, 8, 9, 11, and 12 are modified.

**(6) HCl remains in the core of the polar vortex in austral winter**
*In my first review it was stated: "The authors find here for the MIROC3.2 CCM that some HCl remains in winter in the core of the polar vortex in darkness. Such a model behaviour is expected as there is not enough ClONO2 and no light in the core of the vortex. This model behaviour was reported by Grooß et al. (2018) for three models. Further they showed that this model feature is not found in observations of HCl. I suggest stating that the MIROC3.2 CCM shows the same issue – provided that the authors agree."*

*In their reply, the authors state that they "do not agree with the reviewer's opinion that MIROC3.2 CCM shows the same issue". They have added information and discussion on the role of HOCl, which I consider an important improvement. However, if I look at the HCl values at 87.9 in figure 15, I see that HCl remains constant (and above 0.5 ppb) for days 140-160 (this airmass is in darkness as can be seen from Cl2, panel c). Isn't this the issue discussed by Grooß et al. (2018)? HCl is only close to zero at day ~200 (I would argue due to HOx production in sunlight); after this day HOCl starts increasing (see also below).*

We agree with the reviewer that HCl remains constant for days 140-160 in our MIROC3.2 CCM, which might be the same phenomena as shown for three models in Grooß et al. (2018).   However, HCl starts to decrease from day 160, and reached almost zero around day 210 (29 July) at 87.9°S in Figure 15.   At this point, sun light is not yet available at 87.9°S, and HOCl has not yet started increasing.   This is the main difference of our MIROC3.2 CCM result with other three models in Grooß et al. (2018).

*If the authors really do not agree that they see the "HCl discrepancy", this is only possible if they have a*

*process in the model which removes the remaining HCl in darkness. Which process could there be in the model? Does the model contain some process producing NOx in polar night? The latter point would be very interesting. Another possibility would be that MIROC3.2 does not see this issue because of a too coarse spatial resolution, but this would be a model artefact.*

Our MIROC3.2 CCT doesn't have a process which removes the remaining HCl in darkness, as is now shown by the list of chemical reactions in Appendix Table A2.   As is described in the revised text, attached Figure R5 in this reply, and reply to the reviewer for the point "(8) Continuous loss of HCl in the core of the polar vortex", we consider the continuous loss of HCl occurs by the mixing of ClONO$_2$ rich airmass with remaining HCl by the heterogeneous reaction (R1) and with HOCl by reaction (R4).

**(7) Overestimated transport across the vortex edge in models**
*The paper originally stated that transport across the vortex edge might be overestimated in the MIROC3.2 model results. This is perhaps no surprise given the relatively coarse (T42) horizontal spatial resolution. In the reply, the authors now state that they "looked at the differences between observed and modeled N2O gradients at the vortex edge, but could not find apparent differences between them". Thus they do not seem to find the same result as Hoppe et al. (2014) who found that models might overestimate mixing across the vortex edge.*

*Such a point should not be covered only in passing in the reply. I suggest to include the relevant plots in the reply (which will remain accessible at ACP) or, even better, include this information as an appendix to the paper.*

We now prepared the comparison plots for N$_2$O field between MLS measurements and MIROC3.2 CCM outputs, which are attached at the end of this reply as Figures R1, R2, R3, and R4.   For those Figures, upper panels show southern polar plots of temperature and N$_2$O field by MLS measurements at 490 K (~56 hPa), while lower panels show N$_2$O field by MIROC3.2 CCM output at 50 hPa, for June 24 (Fig. R1), June 29 (Fig. R2), July 4 (Fig. R3), and July 9 (Fig. R4).   The color scale are chosen to be similar values between MLS and MIROC3.2 CCM.   The most apparent difference is that the low N$_2$O values seen in MLS measurements at the vortex core were not reproduced by MIROC3.2 CCM.   For the N$_2$O gradients around the vortex edge, it is rather difficult to tell the difference between MLS measurements and MIROC3.2 CCM outputs.   However, if you see the difference in N$_2$O values between the core of polar vortex and vortex edge on July 4 in Fig. R3, it was about 100 ppbv (100 and 200 ppbv for core and edge, respectively) for MLS measurement.   On the other hand, the difference in N$_2$O values between the core of the polar vortex and vortex edge for MIROC3.2 CCM was about 80 ppbv (150 and 230 ppbv).   Therefore, it is supposed that there may exists more mixing of airmass by MIROC3.2 CCM compared with MLS measurements, due to coarse resolution of the model calculation.

**(8) Continuous loss of HCl in the core of the polar vortex**

*In response to the review comments, the authors have substantially revised the discussion on continuous loss of HCl in the core of the polar vortex. Nonetheless, I would like to come back on a few points of which I think that they could be better represented in the manuscript.*

*Note that the HCl loss processes described by Grooß et al. (2018) and Solomon et al. (2015) are very different: Grooß et al. (2018) describe a polar night process, whereas Solomon et al. (2015) describe a dynamical process (acting later in the course of the existence of the polar vortex). The dynamical process suggested by Solomon et al. (2015) requires light as formation of ClONO2 is involved. This is also true for the explanation for the continuing decline of HCl under sunlit conditions through the formation of HOCl (Grooß et al., 2011; Müller et al., 2018).*

*Finally, ClONO2 is not enhanced at the vortex edge in June or May (see Figs. 13 and 14). Also transport of ClONO2 from the vortex edge to the vortex core does not occur in isolation but in mid-winter likely mixes in air with enhanced HCl and ozone (see below and first review). A very different issue again is the mixing of 'out-of-vortex' air; first, this is less likely and, second, it would certainly bring in non-activated air with substantial concentrations of HCl.*

We agree to reviewer that Solomon et al. (2015) and Jaeglé et al. (1997) are describing rather different dynamical process on mixing of airmass after August (day 210-), which is different from the continuous loss of HCl at the core of the polar vortex in June and July (day 160-210).   Therefore, we deleted the citation of Solomon et al. (2015) and Jaeglé et al. (1997) here.   Nevertheless, we believe that the continuous loss of HCl in June and July occurs due to the mixing of $ClONO_2$ containing air with remaining HCl at the vortex core.   In order to show our idea of mixing of $ClONO_2$ rich air with remaining HCl at the core of the polar vortex during the polar night time (June to July), we now created a Figure R5 (attached at the end of this reply) which shows equivalent-latitude-mean distribution of HCl and $ClONO_2$ from MIROC3.2 CCM at 50 hPa from early June to late July in 2007. At the beginning of June (Fig. R5(a)), the shape of distributions of HCl and $ClONO_2$ were quite similar. Gradually, the amount of HCl and $ClONO_2$ in the polar vortex decreased in mid-June, (Fig. R5(b)), then the amount of $ClONO_2$ at the vortex edge (equivalent latitude of ~60°S) showed an increase while HCl continued to decrease there (Fig. R5(c)).   Afterwards, there existed an apparent difference in the distribution of $ClONO_2$ and HCl at the vortex boundary region (equivalent latitude of 62-70°S; Figs. R5(d)-(f)).   $ClONO_2$ existed about 5-8 degrees toward poleward compared with HCl throughout this period.   Consequently, HCl near the vortex edge reacted with the $ClONO_2$ due to the mixing of airmass by heterogeneous reaction (R1) until HCl was totally depleted (See HCl values at ~65°S equivalent latitude on June 29 and July 9 in Figure R5 (d) and (e)).   Such reaction continues toward poleward during the polar night period between June and July, and gradually HCl value in the polar vortex decreased as is shown in Figure R5 (f).   Such differences in distribution of HCl and $ClONO_2$ can be also seen if you look at the distribution of these species carefully at the edge of polar vortex on June 24, 2007 in Figure 13.

*(9) Model description*

*In response to my first review, the authors have substantially extended the model description. However, the detailed description of the heterogeneous reactions is in Japanese and the paper still does not contain a list of reactions (e.g. add a list of reactions as an electronic appendix). Perhaps a bit more could be done regarding model description. As discussed below, some model behaviour raises the questions, exactly which reactions are taken into account in the model formulation.*

In response to the reviewer's comment, we now added a new appendix Table A2 which shows all the 42 photolysis reactions, 140 gas phase reactions, and 13 heterogeneous reactions with their reaction coefficients.

***(10) Data Availability***
*A data availability statement has now been added to this paper. I think in particular making the FTIR data available is very helpful. You might want to add statements regarding MLS, MIPAS and MIROC3.2.*

The data availability for MIROC3.2 CCM outputs were already stated in the previous manuscript.   Now, we added data availability statement for MLS, MIPAS, and CALIOP satellite data at the same place.

***Some details regarding the revised version***

*(11) p 1, l 21: ClONO2 was almost depleted; but what about HCl?*

HCl was also almost depleted in our observations.   We now modified the text to: "both HCl and ClONO$_2$ were almost depleted ...".

*(12) p 1, l 25 "comprehensive behaviour" is unclear, what is meant here?*

We reworded to: "... to see the behavior of whole chlorine and related species ...".

*(13) p. 3, l 6: perhaps you want to add a reference here for R 12 also? Santee et al. (2008) would be a possibility.*

We added Santee et al. (2008) for the reference here.

*(14) p. 3, l 22: replace 'formation' by 'existence'*

It was replaced as suggested.

*(15) p. 8, l 31: This relation (Eq. 2) is only true for conditions of 1997. Assuming that N2O is constant, the Cly calculated from this equation is only valid for 1997. Correct? So one needs to take into account the temporal trend of Cly between 1997 and 2007 or 2011. In reality there are also temporal changes of N2O that should be*

*taken into account. Only thereafter the adjustments suggested by Strahan et al. (2014) should be taken into account.*

Thank you for your comment.    We now considered the temporal trends of $Cl_y$ and $N_2O$ values, and calculated $Cl_y*$ values by considering the change of $Cl_y$ and $N_2O$ trends.    Accordingly, plot values of $Cl_y*$ and ratios to $Cl_y*$ in Figs. 4-9, 11, and 12 are also modified.

*(16) p 9, l 5: state here to which figure the 'light shaded region' etc belongs.*

It was for Figures 4-7.    It is now stated in the text.

*(17) p 9, l 16: "was occurred" ---> "occurred"*

It was modified as suggested.

*(18) p 10, l 20: 'ratio of HCl' etc – this is unclear. I think you mean mixing ratio here? Or HCl/Cly ratio? This should be clarified. (similar in l. 29).*

In this section, we meant "ratio of HCl" to "$HCl/Cl_y*$ ratio".    However, it might be a bit unclear for readers. Instead, we now reworded to use the term "$HCl/Cl_y*$" etc. throughout this section.

*(19) p. 11, l 32: taken up by PSCs (or similar)*

It was modified as suggested.

*(20) p 12, l 2 (R18): there is also the reaction HNO3 + OH ---> NO3 + H2O*

We now added the suggested new reaction (R19) here.

*(21) p 12, l. 27: this is indeed not unlikely, however the change of stratospheric Cly between 1997 and 2007 as well as between 1997 and 2011 can be quantified and should not be ignored.*

We agree.    We deleted this explanation here.    Also, the change of stratospheric $Cl_y$ among 1997, 2007, and 2011 is now considered in the draft.

*(22) 13, l 8: note that enhanced ClONO2 is not seen in June!*

As stated in the response (8), there is a $ClONO_2$ enhanced area at equivalent latitude 62-70°S in June and July where the HCl is very low (see attached Figure R5).

*(23) p 13, l 11: not only CLaMS but also SD-WACCM and TOMCAT/SLIMCAT*

Now, all three models are stated in the draft.

*(24) p 14, l 21: "the decrease of HCl stopped . . . " – I think the authors have an important point here. But is this not the model behaviour reported by Grooß et al. (2018)? And a model behaviour which is not in agreement with HCl observations. I suggest more discussion here.*

I think at this point, HCl sudden decrease stopped because the counter-part of reaction (R1); $ClONO_2$ was gone at this moment.   This is the same as the model behavior reported by Grooß et al. (2018).   However, the decrease of HCl continued from mid-June to late July (days 160-210) in the core of the vortex (87.9°S), which is the different feature as Grooß et al. (2018).   This point is now discussed in Section 4.6 in more detail.

*(25) p. 15, l 2: note that there is also HOCl production in the gas-phase (ClO + HO2; the details might depend on the HO2 production in the photolysis of CH2O, see e.g. Müller et al. (2018). I would be interesting to analyse here in detail the assumptions in the MIROC3.2 model.*

We checked the list of all the reactions in MIROC3.2 CCM in new Table A2 and confirmed that all the reactions which consist of C1, C2, C3, and C4 in Müller et al. (2018) was included in our model calculation.   Therefore, the formation of HOCl at the vortex core was suggested to be due to chemical cycles C1, C2, C3, or C4 if the airmass was travelled at the sunlit area a few days earlier due to the distortion of the polar vortex in this period. Such a discussion is now added in the new draft.

*(26) p. 15., l 7: I think cycles C3 and C4 are not defined in the manuscript. I suggest to include a brief explanation here not just a reference.*

The explanation of cycles C1, C2, C3, and C4 are now added in Appendix C, including all related reactions.

*(27) p. 15, l 19: For this interpretation, it is important to take into account that the poleward transport of ClONO2 by mixing (this is the point here if I understand the text correctly) does not occur in isolation. The impact of such mixing depends on the time period considered. In deep winter mixing would not transport only ClONO2, but also HCl and ozone. On the other hand, in June (Fig. 13) there is no enhancement of ClONO2 at the vortex edge, so what could be the effect of mixing of vortex edge air in June?*

As is now explicitly shown in attached Figure R5, there is an airmass at equivalent latitude of ~62-70°S where there was substantial amount of $ClONO_2$, but almost no HCl between the middle of June and the end of July. If such an airmass mixes with more poleward airmass, it can destroy the remaining HCl at the vortex core without sunlight.   If you carefully look at the $ClONO_2$ distribution on June 24, 2007 in Figure 13, you can

see some enhancement of ClONO$_2$ around the polar vortex edge where there is no HCl there.

*(28) p 16, l 10: PSCs do not form at NAT saturation temperature a stated in this sentence. In general; I suggest using the same wording throughout the paper for this issue. Be clear if you mean onset of heterogeneous chemistry, PSC formation/existence etc. For example it is NAT saturation temperature, not NAT PSC saturation temperature.*

Thank you for your comments. We checked the PSC-related description throughout the draft, and tried to more clearly define the PSC formation/existence etc. Now, it was stated "When the stratospheric temperature over Syowa Station fell ~4K below NAT saturation temperature, PSCs started to form ...".

*(29) p 16, l 15: 'comprehensive behaviour'?*

It was reworded as: "To see the behavior of whole chlorine and related species ...".

*(30) Fig 13: It is difficult to see two white circles here – I can see only one.*

There is only one circle in each panel, because only one of outer or inner edges were defined for these days (please see Figures A1 and A2). In order to more clearly state this, the term "(outer) edge" was added in the caption.

*(31) Fig 14: It is difficult to see two white circles here for August and October – I can see only one. The last sentence of the caption is difficult to understand. The dotted line for 5 July 2011 is confusing – is it correct? It looks rather different than the various chemical species. Is the point here that on 5 July 2011 only an inner vortex edge is defined but no actual vortex edge?*

We are sorry for the confusion. In all these four days, only either outer edge (Aug. 19, 21, Oct. 9) or inner edge (July 5) was defined at 450 K (see Figure A2 (a)). Since the locations of inner and outer edge were different, we plotted the inner edge with different dotted circle. We modified the captions to more clearly show the feature of defined vortex edges in Figures 13 and 14.

[Figure]

[Figure]

Fig. R1. SH N$_2$O field on 2007/06/24 (175 days)

Fig. R3. SH N$_2$O field on 2007/07/04 (185 days)

[Figure]

Fig. R2. SH N$_2$O field on 2007/06/29 (180 days)

Fig. R4. SH N$_2$O field on 2007/07/09 (190 days)

**(a)**

[Figure]

**(b)**

[Figure]

**(c)**

[Figure]

**(d)**

[Figure]

**(e)**

[Figure]

**(f)**

[Figure]

Figure R5.   Equivalent-latitude-mean distribution of HCl and ClONO$_2$ from MIROC3.2 CCM at 50 hPa on (a) June 1 (day 152), (b) June 11 (day 162), (c) June 21 (day 172), (d) June 29 (day 180), (e) July 9 (day 190), and (f) July 19 (day 200), 2007.   Dotted line shows the defined (outer) polar vortex edge at 450 K.

[revised manuscript text omitted]